# Charge-Compensated Derivatives of Nido-Carborane

**Marina Yu. Stogniy [1,2,*], Sergey A. Anufriev [1]** and **Igor B. Sivaev [1,3]**

1.  A.N. Nesmeyanov Institute of Organoelement Compounds, Russian Academy of Sciences, 28 Vavilov Str., 119334 Moscow, Russia
2.  M.V. Lomonosov Institute of Fine Chemical Technology, MIREA—Russian Technological University, 86 Vernadsky Av., 119571 Moscow, Russia
3.  Basic Department of Chemistry of Innovative Materials and Technologies, G.V. Plekhanov Russian University of Economics, 36 Stremyannyi Line, 117997 Moscow, Russia
*   Correspondence: stogniy@ineos.ac.ru

**Abstract:** This review summarizes data on the main types of charge-compensated nido-carborane derivatives. Compared with organic analogs, onium derivatives of nido-carborane have increased stability due to the stabilizing electron-donor action of the boron cage. Charge-compensated derivatives are considered according to the type of heteroatom bonded to a boron atom.

**Keywords:** nido-carborane; charge compensation; onium derivatives; synthesis; reactivity

## 1. Introduction

The synthesis of the first polyhedral boranes, carboranes, and metallacarboranes in the early 1960s was one of the major highlights in the development of inorganic chemistry over the last century [1]. The first reports on the synthesis of icosahedral carboranes appeared almost sixty years ago, at the end of 1963 when both the United States and the Soviet Union almost simultaneously declassified documents about their boron fuel projects [2–6]. A few months later, the nucleophile-promoted removal of one boron atom from the icosahedral ortho-carborane cage to form the 11-vertex nido-carborane cage species (Figure 1) was reported [7,8]. It was one of the most significant findings in the early years of the development of carborane chemistry, and now, more than five decades later, it remains indispensable for the synthesis of numerous metallacarboranes [9–17] and hydrophilic functionalized carboranes for medical [18–27] and other [28–40] applications.

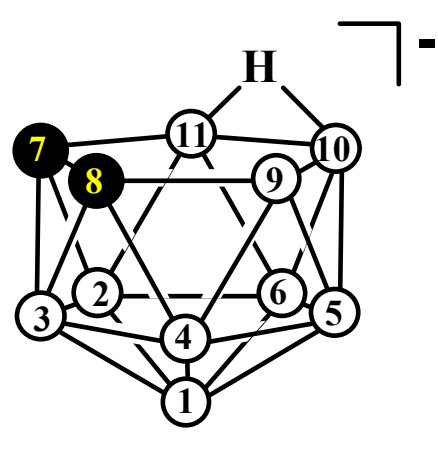

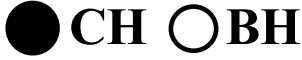

**Figure 1.** The idealized structure and atom numbering of nido-carborane $[7,8-C_2B_9H_{12}]^-$.

Metallacarboranes based on the dicarbollide ligand $[7,8\text{-}C_2B_9H_{11}]^{2-}$, which is formed upon the deprotonation of nido-carborane with strong bases, resemble the well-known transition metal cyclopentadienyl complexes. However, the dicarbollide ligand differs from the cyclopentadienyl ligand in a number of ways. In addition to its 3D character, the dicarbollide ligand is a significantly stronger donor than the cyclopentadienyl one and has a double charge. The donor nature of the dicarbollide ligand can be largely tuned via the introduction of substituents of various natures. At the same time, the charge of the ligand can be partially compensated by introducing into the dicarbollide ligand the so-called charge-compensating substituents of an onium nature (ammonium, phosphonium, sulfonium, etc.). This significantly brings the properties of the dicarbollide and cyclopentadienyl complexes closer together and causes a high interest in metallacarboranes based on charge-compensated dicarbollide ligands [41–50].

In this study, we review the synthesis and properties of charge-compensated nido-carborane derivatives in which the onium center is bonded to the boron atom directly or through a short single-atom spacer and, therefore, not only reduces the ligand charge but also has a significant effect on its electron-donating properties. Therefore, derivatives containing charge-compensating substituents at the carbon atoms [51–67] or bound to the carborane basket through a longer spacer, for example, obtained by opening cyclic oxonium derivatives [68] and some others [69–72], are beyond the scope of this review.

As a rule, charge-compensating substituents are groups in which the positive charge is localized on the atoms of Group 5 (nitrogen, phosphorus, and arsenic) or Group 6 (oxygen, sulfur, and selenium) elements. In most cases, this atom is bonded directly to the boron atom of the nido-carborane basket. Therefore, the classification of charge-compensated derivatives of nido-carborane according to the type of boron–element bond is the most convenient. Another important factor is the position of the substituent, which can be located in the upper (open) belt (positions 9, 10, and 11) or lower (positions 1, 2, 3, 4, 5, and 6) of nido-carborane. Substituents located in the upper belt of nido-carborane in some cases can affect the coordination environment of the metal in metallacarboranes based on them both due to steric factors and in the presence of additional donor groups. In addition, for the synthesis of derivatives with substituents in the upper belt, substitution reactions in the nido-carborane itself are mainly used, while the preparation of derivatives with substituents in the lower belt is based on the decapitation of the corresponding ortho-carborane derivatives. It should also be kept in mind that positions 1, 3, and 10 are in the plane of symmetry of the nido-carborane cage, and, therefore, the substitution of hydrogen atoms in these positions leads to symmetrically substituted derivatives. At the same time, the substitution of hydrogen atoms in positions 2, 4, 5, 6, 9, and 11 leads to asymmetrically substituted derivatives, which are racemic mixtures of the corresponding isomers.

## 2. Charge-Compensated Derivatives of Nido-Carborane with Boron–Nitrogen Bond

Due to the great diversity of nitrogen chemistry, the charge-compensated derivatives of nido-carborane with the B-N bond are characterized by the greatest variety of forms. The first example of the synthesis of charge-compensated derivatives of nido-carborane with a B-N bond was the reaction of the parent nido-carborane with pyridine in benzene in the presence of anhydrous $FeCl_3$, leading to the asymmetrically substituted pyridinium derivative $9\text{-}Py\text{-}7,8\text{-}C_2B_9H_{11}$ (Scheme 1) [73], the structure of which was later supported via a single-crystal X-ray diffraction study (Figure 2) [74]. When $FeCl_3 \cdot 6H_2O$ was used instead of anhydrous $FeCl_3$, the by-product of the reaction was the disubstituted pyridinium derivative $9,11\text{-}Py_2\text{-}7,8\text{-}C_2B_9H_9$ [74]. The reaction with 7,8-dimethyl-nido-carborane proceeds in a similar way, giving $9\text{-}Py\text{-}7,8\text{-}Me_2\text{-}7,8\text{-}C_2B_9H_9$ [73]. In a similar way, the reaction of nido-carborane with methyl isonicotinate in the presence of $FeCl_3$ in refluxing benzene results in $9\text{-}(4'\text{-}MeO(O)CC_5H_3N)\text{-}7,8\text{-}C_2B_9H_{11}$ [75].

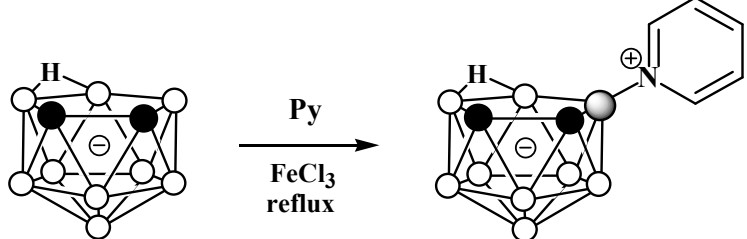

**Scheme 1.** The synthetic process to obtain 9-Py-7,8-C$_2$B$_9$H$_{11}$.

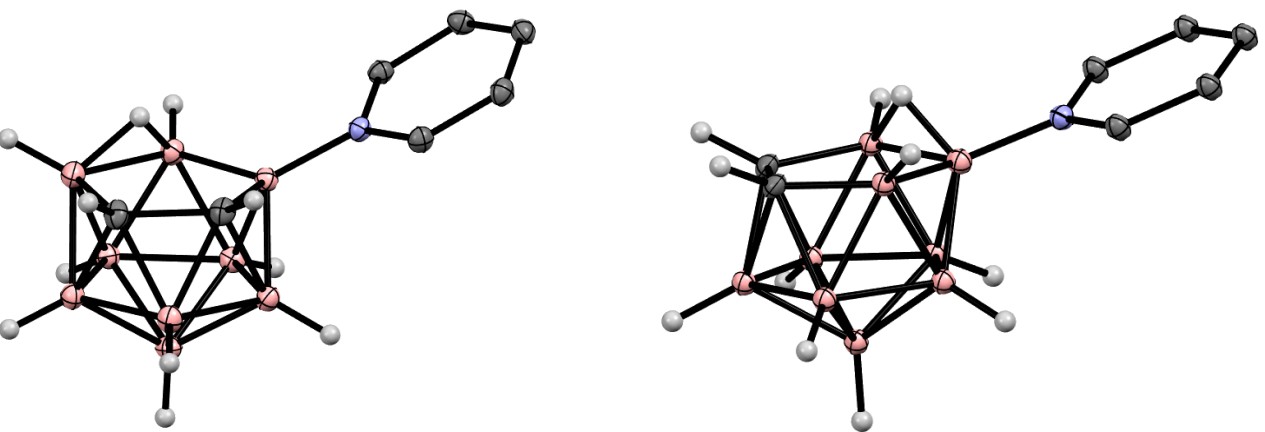

**Figure 2.** Crystal molecular structures of 9-Py-7,8-C$_2$B$_9$H$_{11}$ (**left**) and 10-Py-7,8-C$_2$B$_9$H$_{11}$ (**right**). Hydrogen atoms of organic substituents are omitted for clarity.

The asymmetrically substituted 9-pyridinium derivative of *nido*-carborane was also prepared via the reaction of the parent *ortho*-carborane with pyridine in the presence of copper acetate and water. Similar reactions with *C*-monosubstituted *ortho*-carboranes give a mixture of the corresponding isomeric pyridinium derivatives 9-Py-7-R-7,8-C$_2$B$_9$H$_{10}$ and 11-Py-7-R-7,8-C$_2$B$_9$H$_{10}$ (R = Me, Ph). In the case of 1-XCH$_2$ derivatives of *ortho*-carborane (X = Cl, Br, OH), in addition to a mixture of the corresponding 9- and 11-pyridinium derivatives of *nido*-carborane, the reaction gives the pyridinium methyl derivative 7-PyCH$_2$-7,8-C$_2$B$_9$H$_{11}$ [76].

The reaction of *nido*-carborane with pyridine in the presence of HgCl$_2$ in refluxing benzene gives a mixture of the symmetrically and asymmetrically substituted pyridinium derivatives 10-Py-7,8-C$_2$B$_9$H$_{11}$ and 9-Py-7,8-C$_2$B$_9$H$_{11}$ in a ratio of 2:1 (Scheme 2) [50,77]. The reaction of 7,8-dimethyl-*nido*-carborane [7,8-Me$_2$-7,8-C$_2$B$_9$H$_{10}$]$^-$ with pyridine proceeds in a similar way [77].

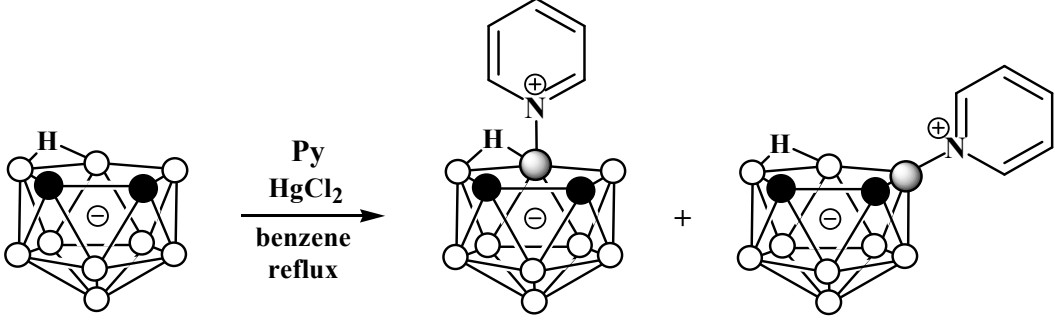

**Scheme 2.** Synthesis of 10-Py-7,8-C$_2$B$_9$H$_{11}$ and 9-Py-7,8-C$_2$B$_9$H$_{11}$ via the reaction of *nido*-carborane with pyridine in the presence of HgCl$_2$.

The symmetrically substituted pyridinium derivative 10-Py-7,8-C$_2$B$_9$H$_{11}$ was prepared via the reaction of the 10-diphenylsulfonium derivative 10-Ph$_2$S-7,8-C$_2$B$_9$H$_{11}$ with pyridine in refluxing chloroform (Scheme 3, Figure 2) [78].

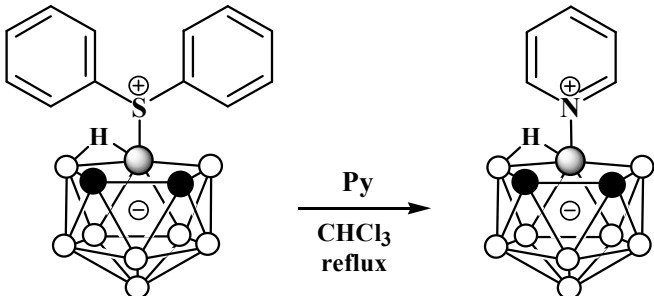

**Scheme 3.** The synthetic process to obtain 10-Py-7,8-C$_2$B$_9$H$_{11}$.

The reaction of *nido*-carborane with 4-phenylpyridine in 1,2-dimethoxyethane in the presence of 2,3-dichloro-5,6-icyanobenzoquinone (DDQ) as an oxidizing agent leads to the corresponding asymmetrically substituted pyridinium derivative [9-(4′-PhC$_5$H$_4$N)-7,8-C$_2$B$_9$H$_{11}$] (Scheme 4) [79].

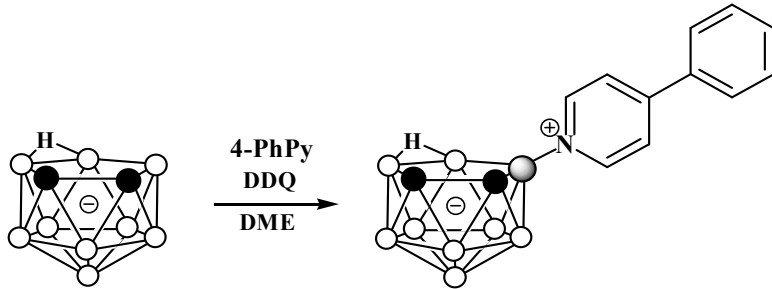

**Scheme 4.** The synthesis of [9-(4′-PhC$_5$H$_4$N)-7,8-C$_2$B$_9$H$_{11}$].

This approach is also applicable to various pyridine derivatives and *C,C′*-disubstituted *nido*-carboranes. The reaction was shown to be tolerant to the halogen, methoxy, methylcarboxy, amino, and vinyl substituents (Scheme 5, Figure 3) [79]. The same products can be prepared via reagent-free electrocatalyzed direct B-N oxidative couplings of *nido*-carboranes with pyridines (Scheme 4, Figure 3) [80,81]. In the case of *C*-monosubstituted *nido*-carboranes, the reaction leads to mixtures of the 9- and 11-pyridinium derivatives [79,81].

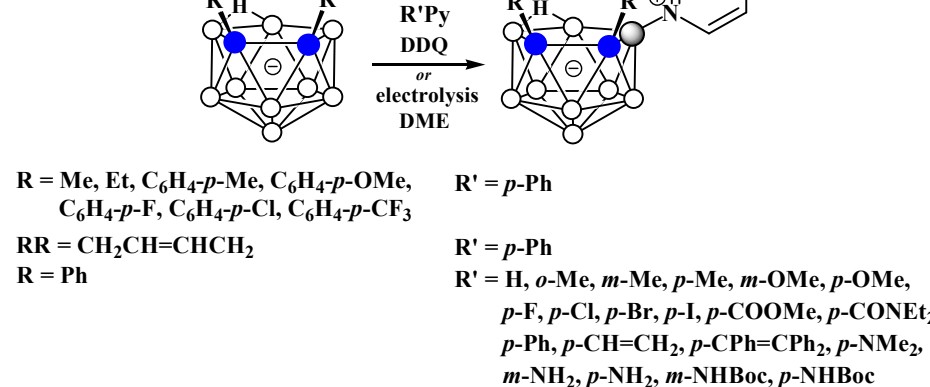

R = Me, Et, C$_6$H$_4$-*p*-Me, C$_6$H$_4$-*p*-OMe,
   C$_6$H$_4$-*p*-F, C$_6$H$_4$-*p*-Cl, C$_6$H$_4$-*p*-CF$_3$        R′ = *p*-Ph

RR = CH$_2$CH=CHCH$_2$                  R′ = *p*-Ph
R = Ph                                R′ = H, *o*-Me, *m*-Me, *p*-Me, *m*-OMe, *p*-OMe,
                                        *p*-F, *p*-Cl, *p*-Br, *p*-I, *p*-COOMe, *p*-CONEt$_2$,
                                        *p*-Ph, *p*-CH=CH$_2$, *p*-CPh=CPh$_2$, *p*-NMe$_2$,
                                        *m*-NH$_2$, *p*-NH$_2$, *m*-NHBoc, *p*-NHBoc

**Scheme 5.** The metal-free or electrocatalyzed direct B-N oxidative couplings of *nido*-carboranes with pyridines.

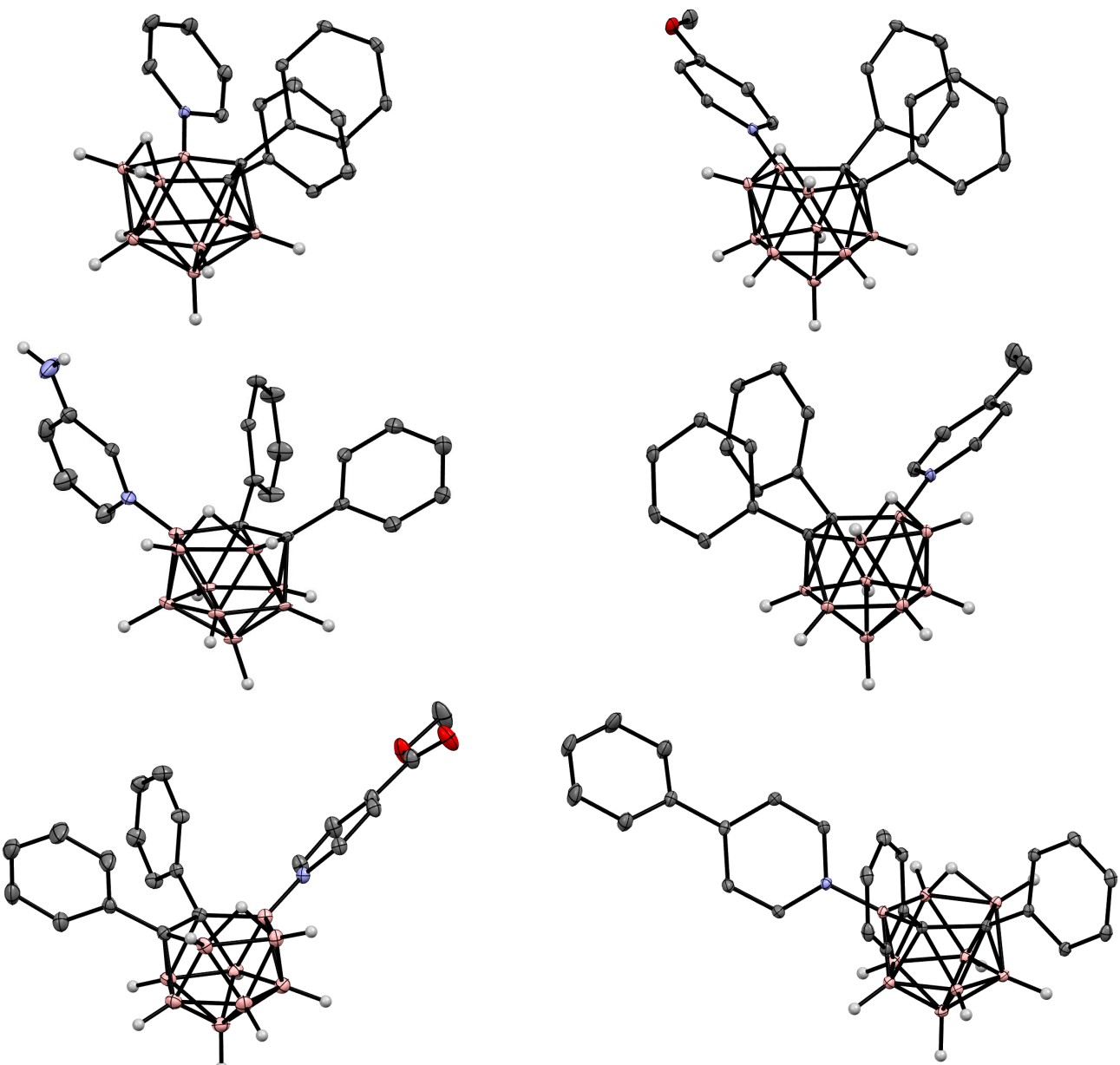

**Figure 3.** Crystal molecular structures of 9-$C_5H_5N$-7,8-$Ph_2$-7,8-$C_2B_9H_9$ (**top left**), 9-(4′-MeOC$_5$H$_4$N)-7,8-$Ph_2$-7,8-$C_2B_9H_9$ (**top right**), 9-(3′-H$_2$NC$_5$H$_4$N)-7,8-$Ph_2$-7,8-$C_2B_9H_9$ (**middle left**), 9-(4′-CH$_2$ = CHC$_5$H$_4$N)-7,8-$Ph_2$-7,8-$C_2B_9H_9$ (**middle right**), 9-(4′-MeO(O)CC$_5$H$_4$N)-7,8-$Ph_2$-7,8-$C_2B_9H_9$ (**bottom left**), and 9-(4′-PhC$_5$H$_4$N)-7,8-$Ph_2$-7,8-$C_2B_9H_9$ (**bottom right**). Hydrogen atoms of organic substituents are omitted for clarity.

The reaction of 7,8-diphenyl-*nido*-carborane with 4,4′-vinylenedipyridine in the presence of DDQ in 1,2-dimethoxyethane gives the corresponding pyridinium derivative containing two *nido*-carboranyl units (Figure 4) [79].

The electrocatalyzed B-N oxidative couplings of *nido*-carboranes with pyridines were used for the synthesis of a *nido*-carborane-based amino acid and 4,4-difluoro-4-bora-3a,4a-diaza-*s*-indacene (BODIPY) derivatives (Figures 5 and 6) [80].

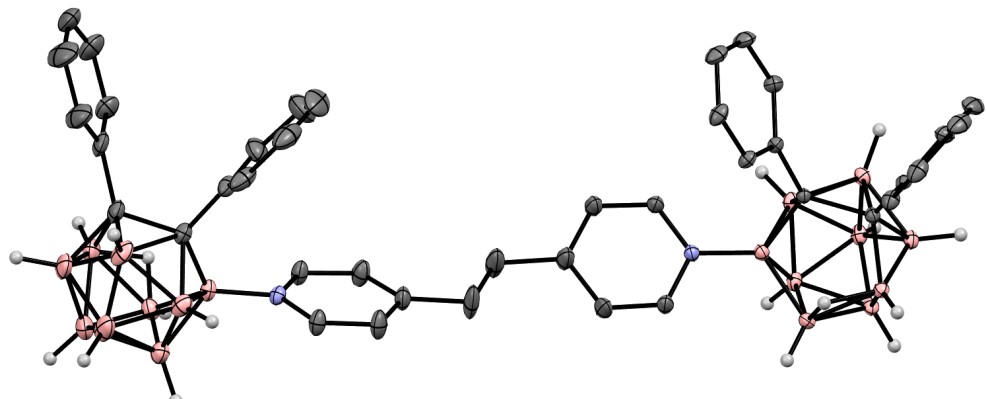

**Figure 4.** Crystal molecular structure of 9,9′-μ-(NC$_5$H$_4$-*p*-CH=CH-*p*-C$_5$H$_4$N)(7,8-Ph$_2$-7,8-C$_2$B$_9$H$_9$)$_2$. Hydrogen atoms of organic substituents are omitted for clarity.

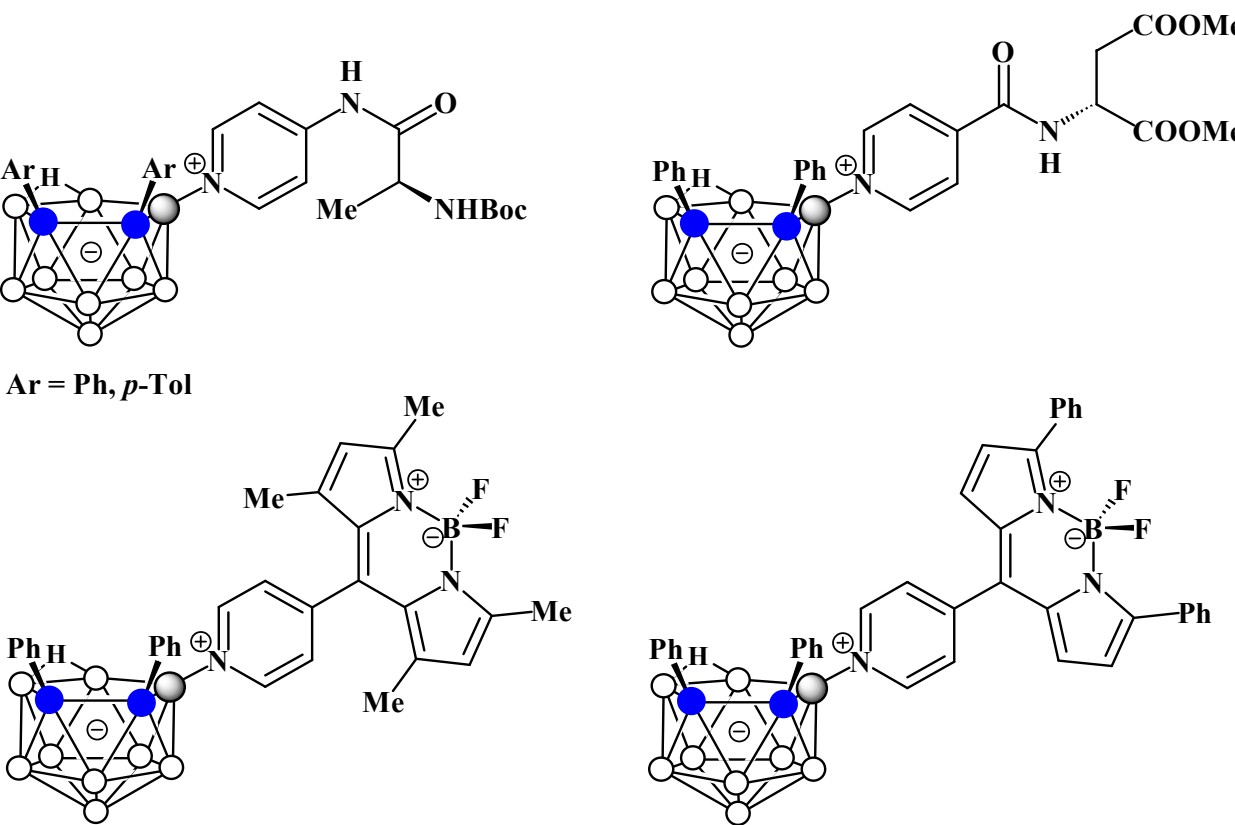

**Figure 5.** The *nido*-carborane-based amino acid and BODIPY derivatives.

The use of quinoline and isoquinoline and their derivatives instead of pyridine leads to the corresponding quinolinium and isoquinolinium derivatives of *nido*-carborane (Scheme 6, Figure 7) [79,80].

The oxidation of *C*-substituted *nido*-carboranes containing a pendant pyridine or quinoline fragment leads to intramolecular cyclization with the formation of the corresponding pyridinium and quinolinium derivatives (Scheme 7, Figure 8) [79–81].

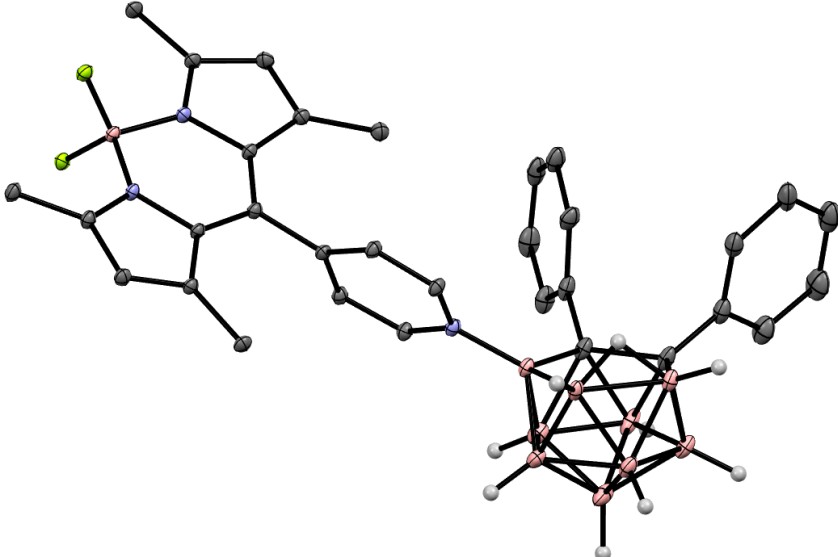

**Figure 6.** Crystal molecular structure of the BODIPY-labeled *nido*-carborane. Hydrogen atoms of organic substituents are omitted for clarity.

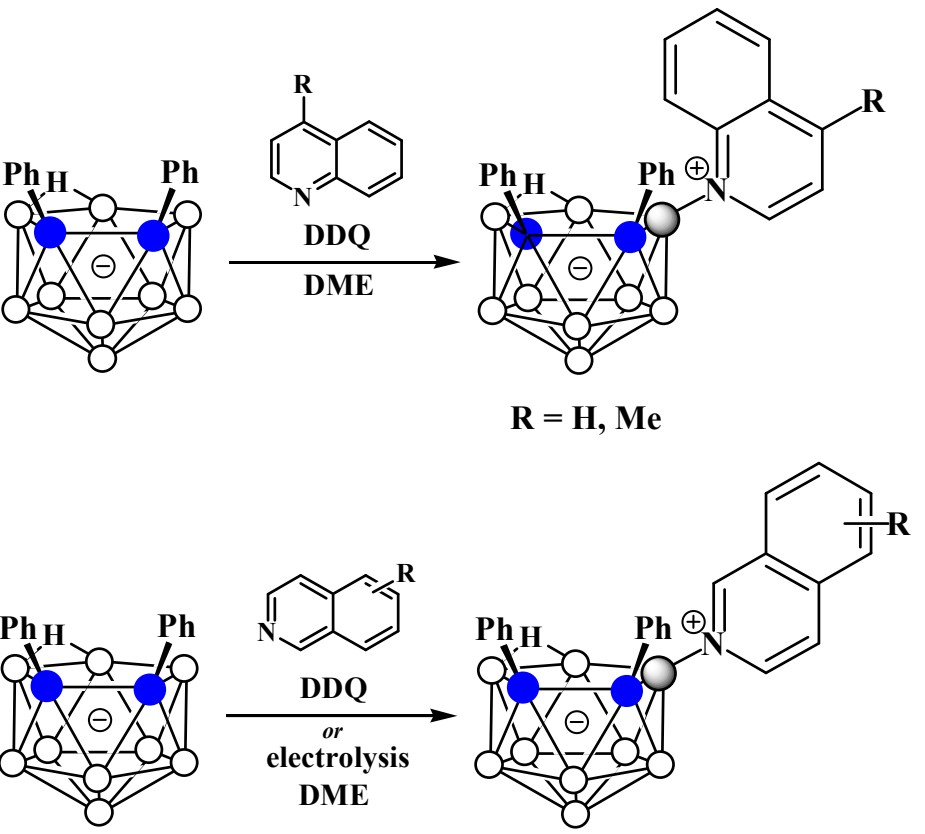

**Scheme 6.** The synthetic process to obtain quinolinium and isoquinolinium derivatives of *nido*-carborane.

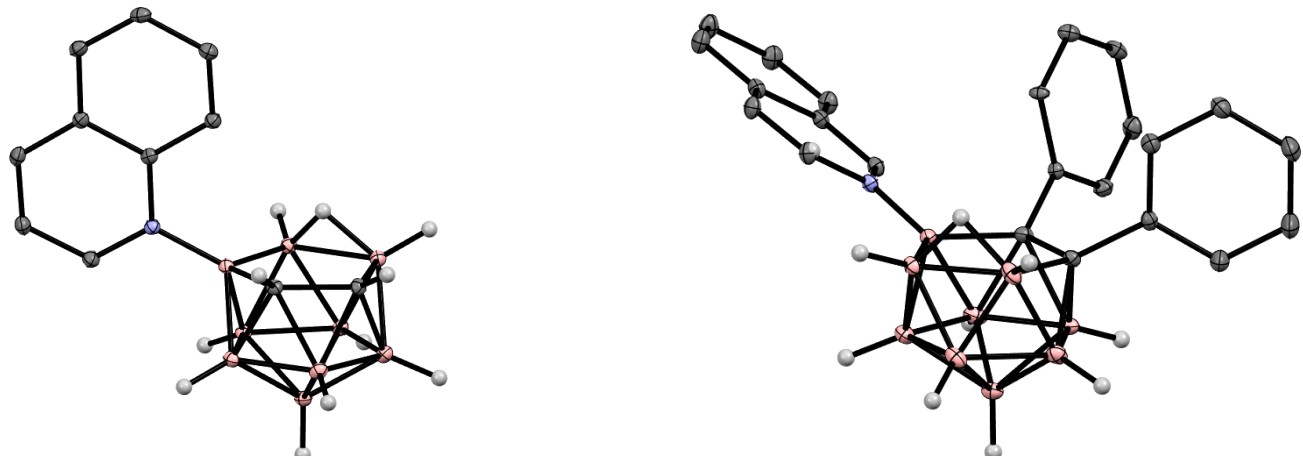

**Figure 7.** Crystal molecular structures of 9-quinoline-1-yl-7,8-$C_2B_9H_{11}$ (**left**) and 9-isoquinoline-2-yl-7,8-$Ph_2$-7,8-$C_2B_9H_9$ (**right**). Hydrogen atoms of organic substituents are omitted for clarity.

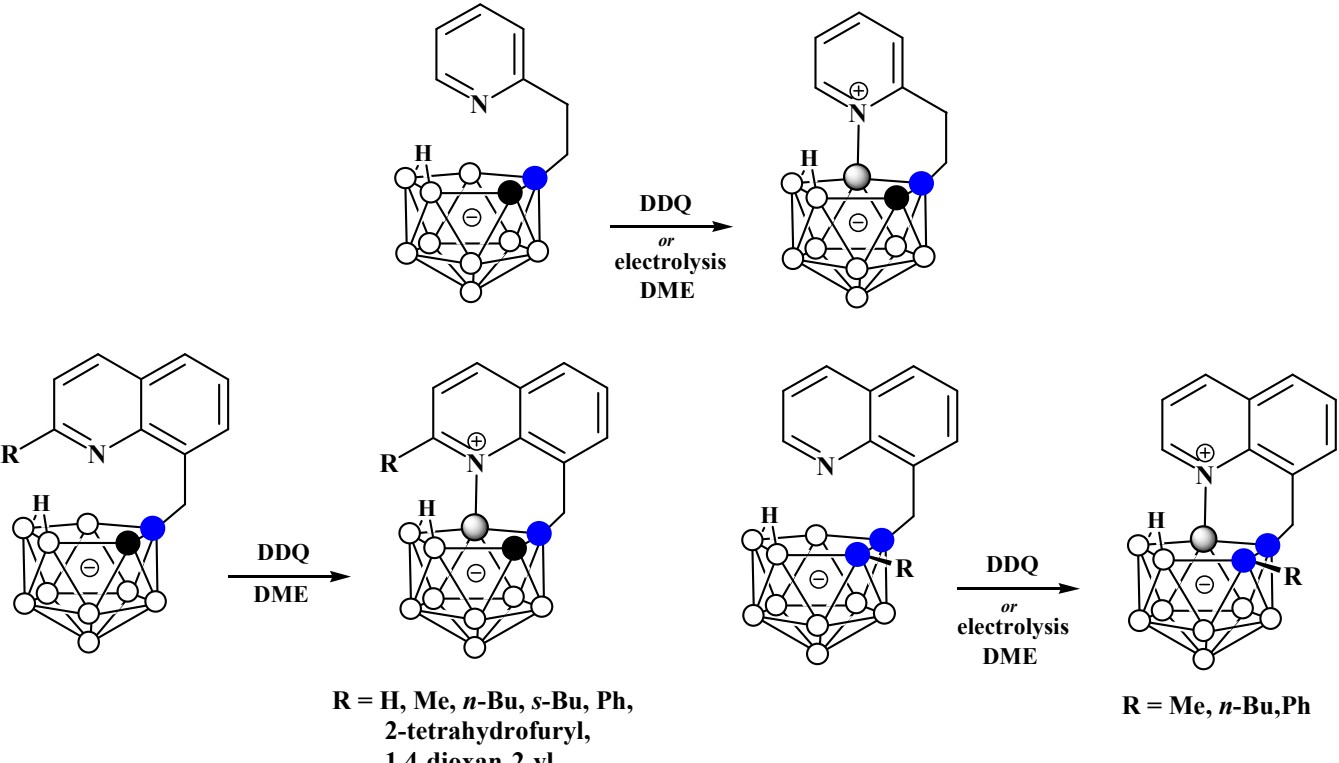

**Scheme 7.** The synthesis of pyridinium and quinolinium derivatives of *nido*-carborane.

Bromination and iodination of 9-Py-7,8-$C_2B_9H_{11}$ with bromine and iodine in acetic acid leads to 11-Br-9-PyN-7,8-$C_2B_9H_{10}$ and 11-I-9-PyN-7,8-$C_2B_9H_{10}$, correspondently (Figure 9) [82,83].

The oxidative coupling can be also applied for the synthesis of *nido*-carboranyl derivatives of other azaheterocycles, including pyrazole, imidazole, oxazole, thiazole, pyrimidine, and azaindoles (Figures 10 and 11) [79–81].

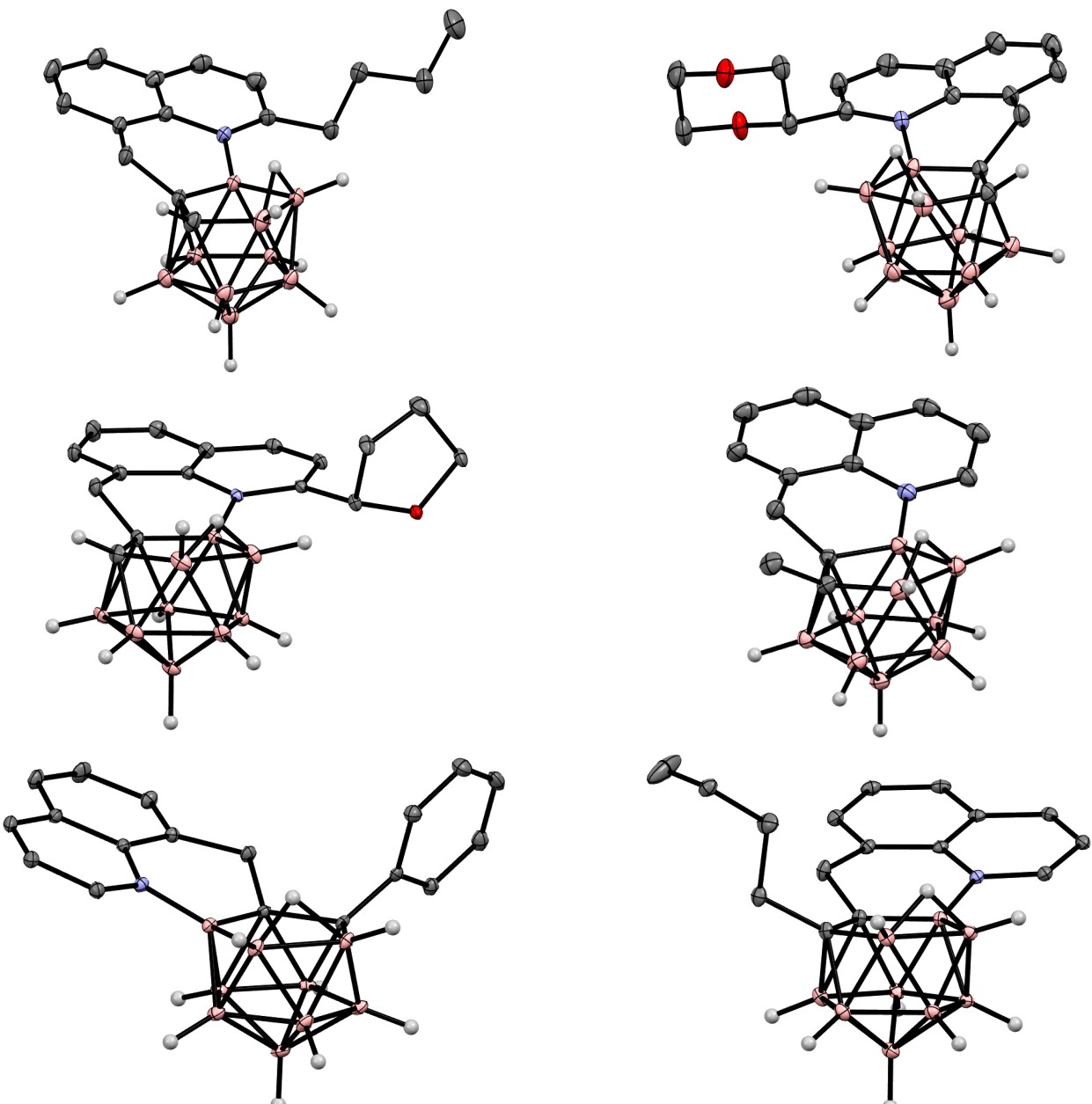

**Figure 8.** Crystal molecular structures of μ-7,11-NC$_7$H$_6$(2'-Bu)CH$_2$-7,8-C$_2$B$_9$H$_{10}$ (**top left**), μ-7,11-NC$_7$H$_6$(1'',4''-dioxan-2''-yl)CH$_2$-7,8-C$_2$B$_9$H$_{10}$ (**top right**), μ-7,11-NC$_7$H$_6$(2''-tetrahydrofuryl)CH$_2$-7,8-C$_2$B$_9$H$_{10}$ (**middle left**), μ-7,11-NC$_7$H$_7$CH$_2$-8-Me-7,8-C$_2$B$_9$H$_9$ (**middle right**), μ-7,11-NC$_7$H$_7$CH$_2$-8-Ph-7,8-C$_2$B$_9$H$_9$ (**bottom left**), and μ-7,11-NC$_7$H$_7$CH$_2$-8-Bu-7,8-C$_2$B$_9$H$_9$ (**bottom right**). Hydrogen atoms of organic substituents are omitted for clarity.

The 9-trimethylammonium derivative of *nido*-carborane 9-Me$_3$N-7,8-C$_2$B$_9$H$_{11}$ was prepared via the reaction of the potassium salt of the parent *nido*-carborane with copper(II) sulfate in the presence of aqueous ammonia and trimethylammonium chloride (Scheme 8) [74]. This approach was also used for the synthesis of a series of 9-alkyldimethylamino derivatives of *nido*-carborane, 9-RMe$_2$N-7,8-C$_2$B$_9$H$_{11}$ (R = CH$_2$Ph, CH$_2$C≡N, CH$_2$C≡CH, (CH$_2$)$_3$Cl, (CH$_2$)$_2$OH, (CH$_2$)$_3$OH, (CH$_2$)$_2$NMe$_2$) (Scheme 8, Figure 12) [84].

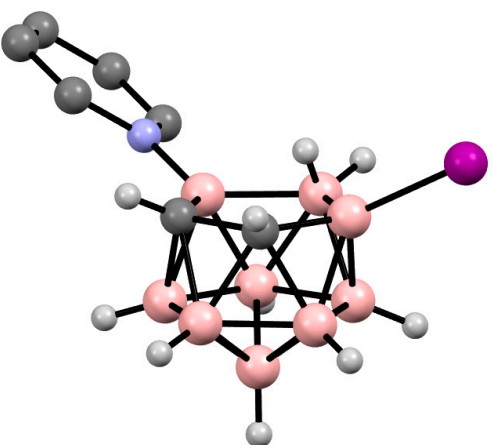

**Figure 9.** Crystal molecular structure of 11-I-9-Py-7,8-C$_2$B$_9$H$_{10}$. Hydrogen atoms of pyridyl substituent are omitted for clarity.

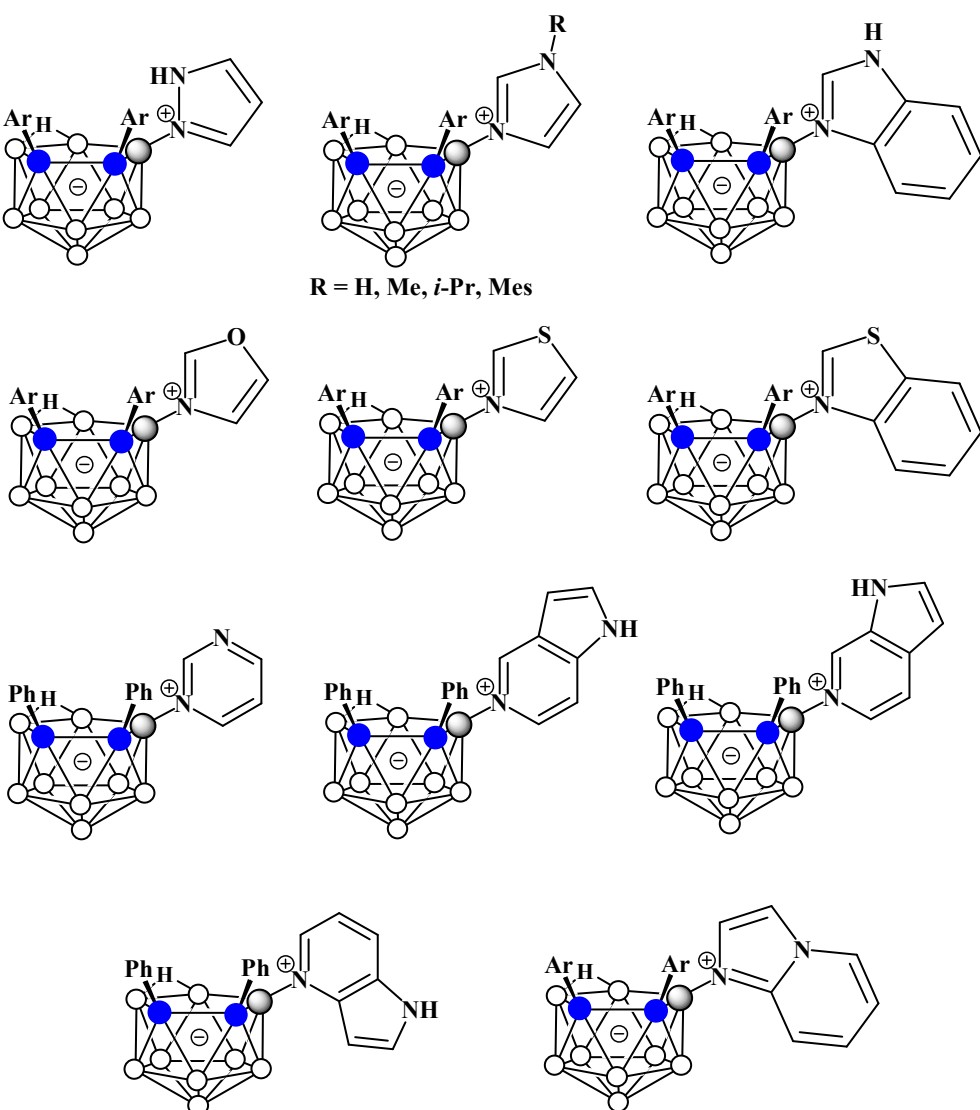

**Figure 10.** The *N*-heterocyclic charge-compensated *nido*-carborane derivatives.

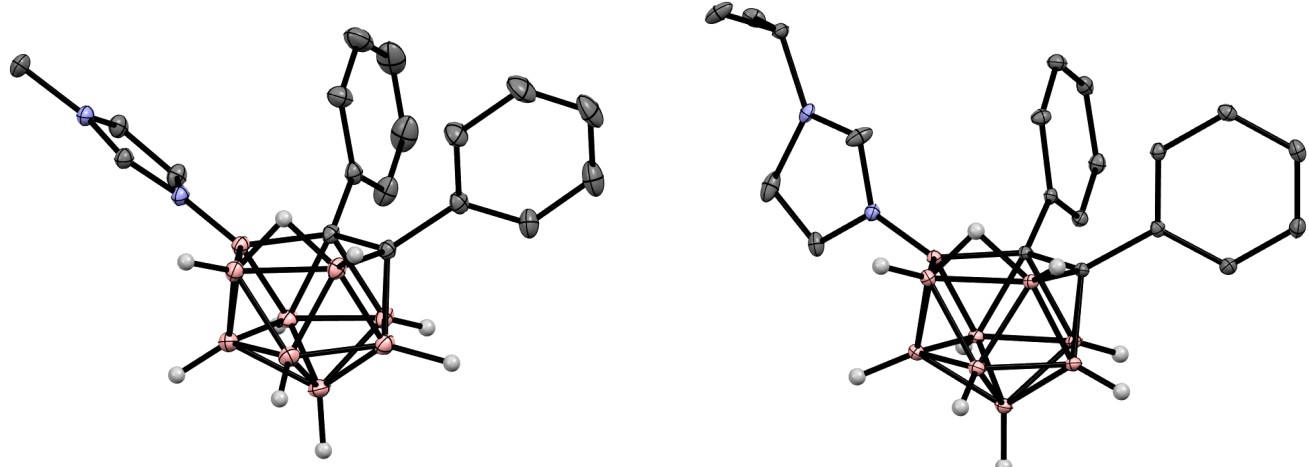

**Figure 11.** Crystal molecular structures of 9-*N*-methylimidazolium-7,8-Ph$_2$-7,8-C$_2$B$_9$H$_9$ (**left**) and 9-*N*-isopropylimidazolium-7,8-Ph$_2$-7,8-C$_2$B$_9$H$_{10}$ (**right**). Hydrogen atoms of organic substituents are omitted for clarity.

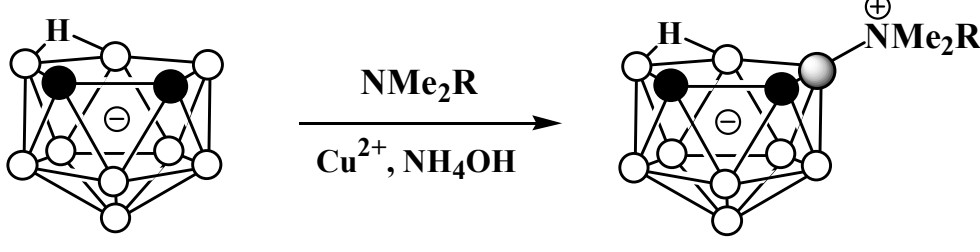

**R = Me, Bn, CH$_2$CN, CH$_2$CCH, (CH$_2$)$_3$Cl,**
**(CH$_2$)$_n$OH (n = 2, 3), (CH$_2$)$_2$NMe$_2$**

**Scheme 8.** The synthetic route to obtain 9-trimethylammonium and 9-alkyldimethylamino derivatives of *nido*-carborane.

An attempt to crystallize 9-HO(CH$_2$)$_3$Me$_2$N-7,8-C$_2$B$_9$H$_{11}$ from acetone led to intramolecular BH-activation with the formation of μ-9,4-Me$_2$N(CH$_2$)$_3$O-7,8-C$_2$B$_9$H$_{10}$ (Figure 13) [84].

The derivative with the *N,N,N′,N′*-tetramethylethylenediamine substituent 9-Me$_2$N(CH$_2$)$_2$Me$_2$N-7,8-C$_2$B$_9$H$_{11}$ can be also obtained via the nucleophilic substitution of iodine in the 9-iodo derivative [9-I-7,8-C$_2$B$_9$H$_{11}$]$^-$ with tetramethylethylenediamine (TMEDA) in the presence of *t*-BuOK (Scheme 9) [85].

Alkylation of 9-Me$_2$N(CH$_2$)$_2$Me$_2$N-7,8-C$_2$B$_9$H$_{11}$ with allyl chloride or propargyl bromide leads to the corresponding cationic derivatives of *nido*-carborane (Scheme 10) [84].

The azido derivative of 9-N$_3$(CH$_2$)$_3$Me$_2$N-7,8-C$_2$B$_9$H$_{11}$ (Figure 12) was prepared by heating the corresponding chloride with sodium azide in DMF in the presence of NaI. The copper(I)-catalyzed azide-alkyne cycloaddition reactions of 9-N$_3$(CH$_2$)$_3$Me$_2$N-7,8-C$_2$B$_9$H$_{11}$ with various terminal alkynes, including phenylacetylene and alkyne derivatives of cholesterol and cobalt and iron bis(dicarbollides), results in the corresponding 1,2,3-triazoles (Scheme 11) [86]. The zwitterionic *nido*-carborane–cholesterol conjugate was also prepared via the Cu(I)-catalyzed cycloaddition of the alkyne derivative of *nido*-carborane 9-HC≡CCH$_2$Me$_2$N-7,8-C$_2$B$_9$H$_{11}$ with 3β-(2-azidoethoxy)-5-cholestene (Scheme 12) [87].

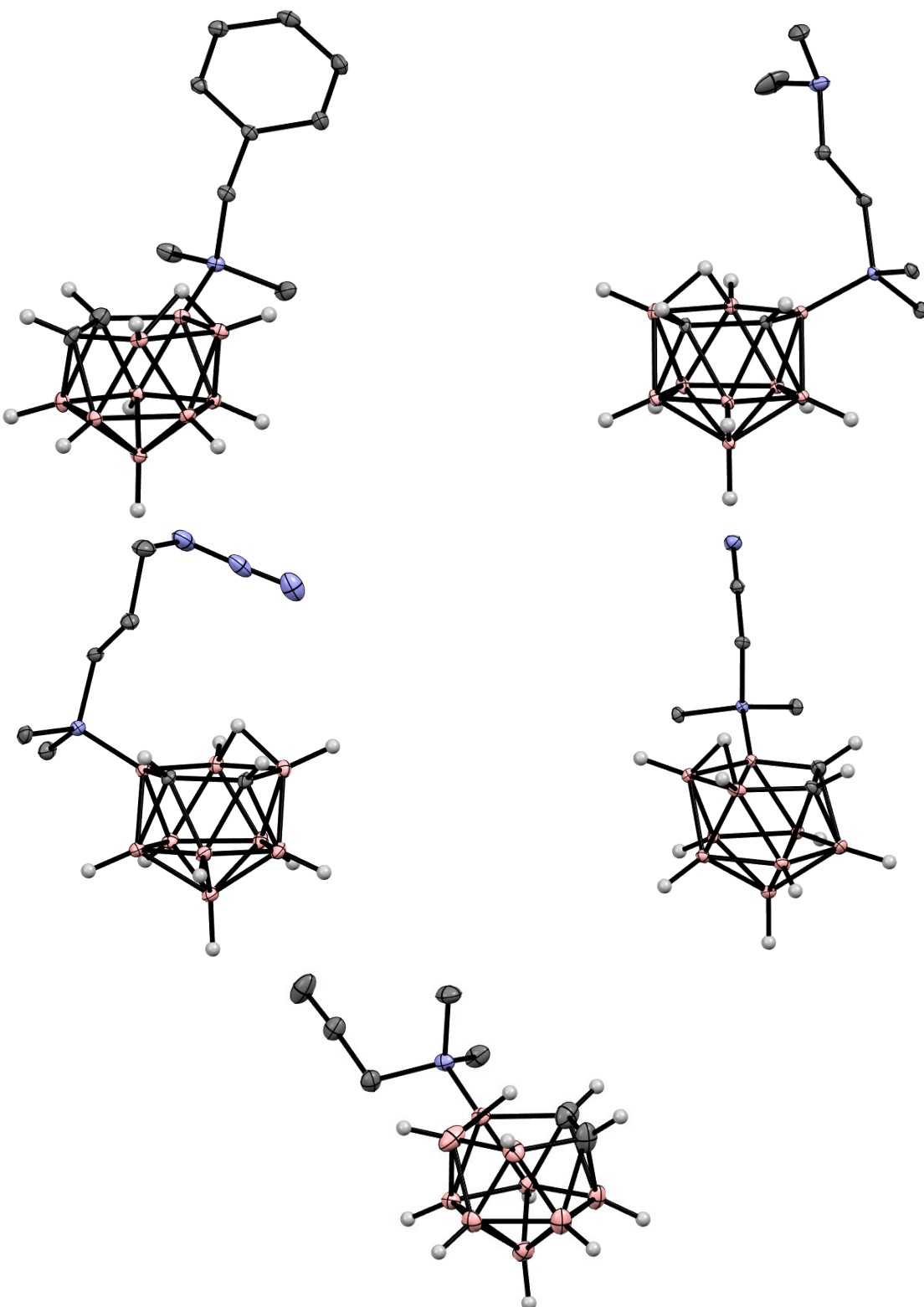

**Figure 12.** Crystal molecular structures of 9-$C_6H_5CH_2Me_2N$-7,8-$C_2B_9H_{11}$ (**top left**), 9-$Me_2NCH_2CH_2Me_2N$-7,8-$C_2B_9H_{11}$ (**top right**), 9-$N_3CH_2CH_2CH_2Me_2N$-7,8-$C_2B_9H_{11}$ (**middle left**), 9-$N{\equiv}CCH_2Me_2N$-7,8-$C_2B_9H_{11}$ (**middle right**), and 9-$HC{\equiv}CCH_2Me_2N$-7,8-$C_2B_9H_{11}$ (**bottom**). Hydrogen atoms of organic substituents are omitted for clarity.

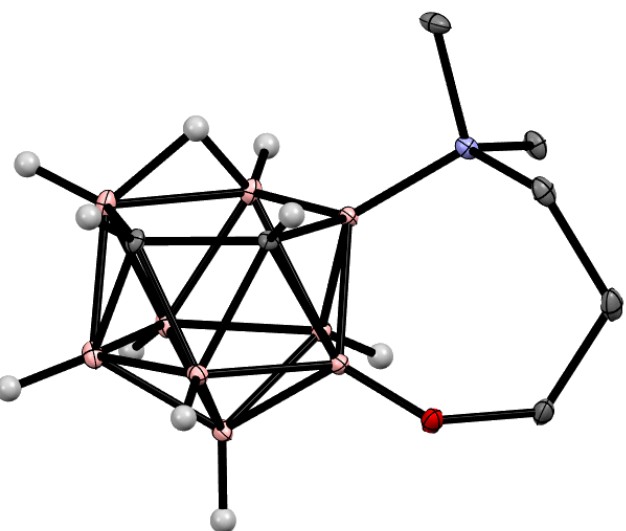

**Figure 13.** Crystal molecular structure of μ-9,4-Me$_2$N(CH$_2$)$_3$O-7,8-C$_2$B$_9$H$_{10}$. Hydrogen atoms of organic substituent are omitted for clarity.

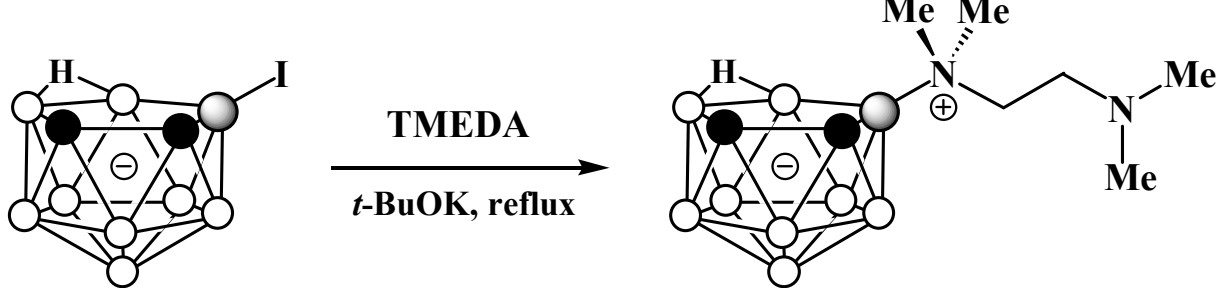

**Scheme 9.** The synthesis of 9-Me$_2$N(CH$_2$)$_2$Me$_2$N-7,8-C$_2$B$_9$H$_{11}$.

**R = CH$_2$-CH=CH$_2$, CH$_2$-C≡CH**

**Scheme 10.** The alkylation of 9-Me$_2$N(CH$_2$)$_2$Me$_2$N-7,8-C$_2$B$_9$H$_{11}$.

The same approach can be used for the preparation of trimethylammonium derivatives of *C*-substituted *nido*-carboranes. The reaction of 7,8-bis(methylthio)-*nido*-carborane with trimethylammonium chloride in the presence of copper(II) sulfate in an aqueous ammonia solution results in 9-Me$_3$N-7,8-(MeS)$_2$-7,8-C$_2$B$_9$H$_9$, whereas the similar reaction of 7-methylthio-*nido*-carborane gives a mixture of 9-Me$_3$N-7-MeS-7,8-C$_2$B$_9$H$_{10}$ (major isomer) and 11-Me$_3$N-7-MeS-7,8-C$_2$B$_9$H$_{10}$ (minor isomer) (Scheme 13, Figure 14) [88].

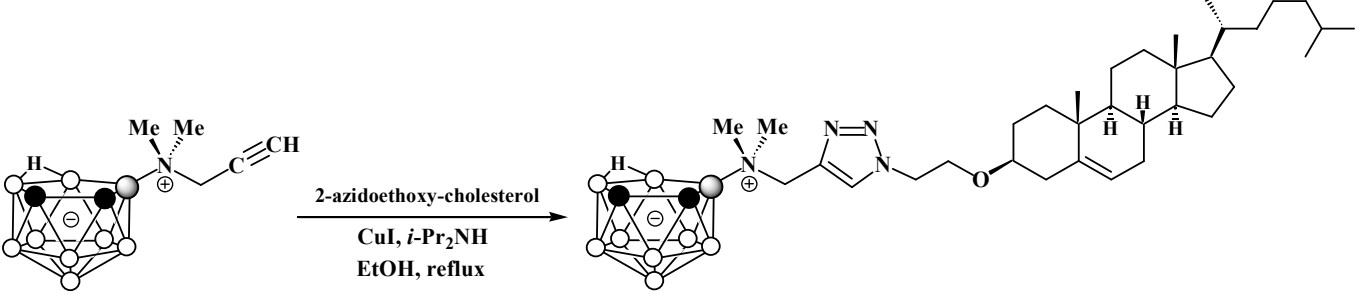

**Scheme 11.** The synthesis of azido derivative of *nido*-carborane 9-N₃(CH₂)₃Me₂N-7,8-C₂B₉H₁₁ and its copper(I)-catalyzed azide–alkyne cycloaddition reactions with various terminal alkynes.

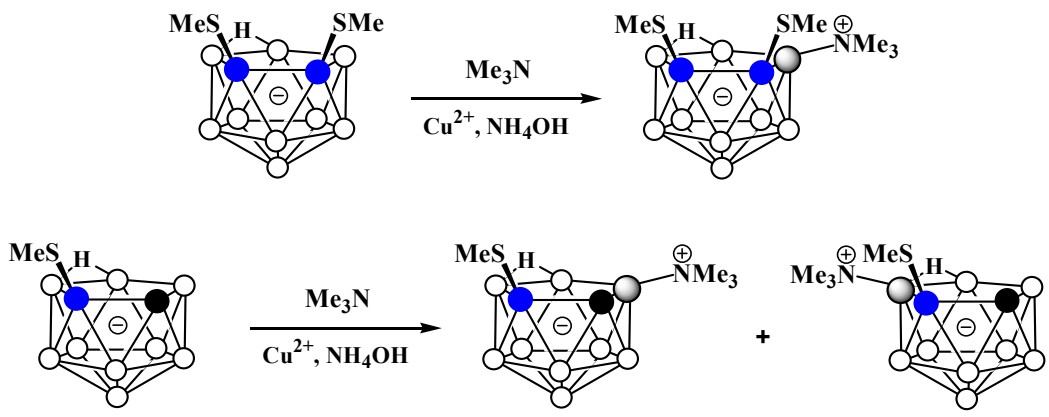

**Scheme 12.** The reaction of 9-HC≡CCH₂Me₂N-7,8-C₂B₉H₁₁ with 3β-(2-azidoethoxy)-5-cholestene.

**Scheme 13.** Synthesis of trimethylammonium derivatives of *C*-substituted *nido*-carboranes.

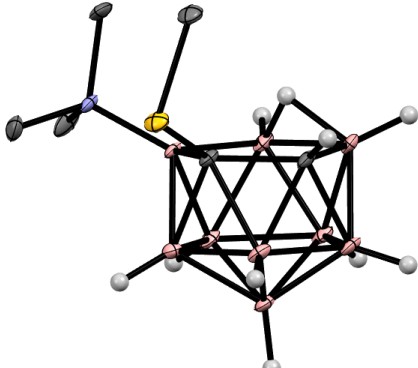

**Figure 14.** Crystal molecular structure of 11-Me$_3$N-7-MeS-7,8-C$_2$B$_9$H$_{10}$. Hydrogen atoms of methyl groups are omitted for clarity.

The symmetrically substituted triethylammonium derivative 10-Et$_3$N-7,8-C$_2$B$_9$H$_{11}$ was prepared in low yield via the reaction of the 10-diphenylsulfonium derivative 10-Ph$_2$S-7,8-C$_2$B$_9$H$_{11}$ with triethylamine in refluxing chloroform (Scheme 14, Figure 15) [78].

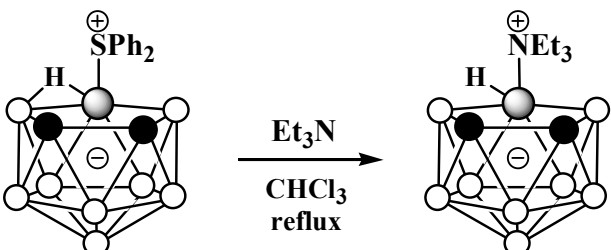

**Scheme 14.** The synthesis of 10-Et$_3$N-7,8-C$_2$B$_9$H$_{11}$ from 10-Ph$_2$S-7,8-C$_2$B$_9$H$_{11}$.

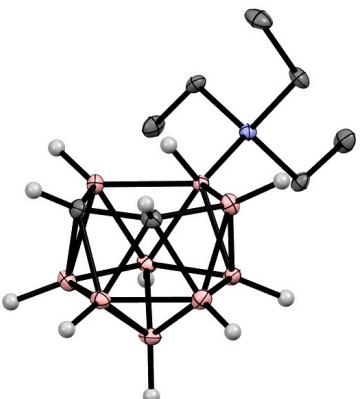

**Figure 15.** Crystal molecular structure of 10-Et$_3$N-7,8-C$_2$B$_9$H$_{11}$. Hydrogen atoms of ethyl groups are omitted for clarity.

The asymmetrically substituted triethylammonium derivative 9-Et$_3$N-7,8-C$_2$B$_9$H$_{11}$ was prepared in low yield via the reaction of the 9-iodo derivative [9-I-7,8-C$_2$B$_9$H$_{11}$]$^-$ with triethylamine in the presence of *t*-BuOK under reflux conditions [85].

A series of cyclic 9-dialkylammonium derivatives of 7,8-diphenyl-*nido*-carborane was prepared via the photoredox coupling of [7,8-Ph$_2$-7,8-C$_2$B$_9$H$_{10}$]$^-$ with secondary amines under blue LED light irradiation in the presence of 9-mesityl-10-methylacridinium perchloratee as the photocatalyst (Scheme 15, Figure 16) [89]. In a similar way, unsymmetrical and acyclic dialkylammonium derivatives of *nido*-carborane can be prepared using tetrahydroisoquinoline and methylbenzylamine, respectively (Figure 16) [89].

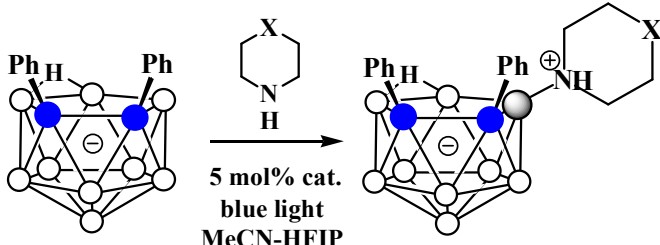

X = CH$_2$, CHPh, CHOMe, CHCOOMe, CHCN,
O, NMe, NCH$_2$C$_6$H$_4$-*p*-Cl, N-(2-pyrimidyl),
NSO$_2$Me

**Scheme 15.** Synthesis of the cyclic 9-dialkylammonium derivatives of 7,8-diphenyl-*nido*-carborane.

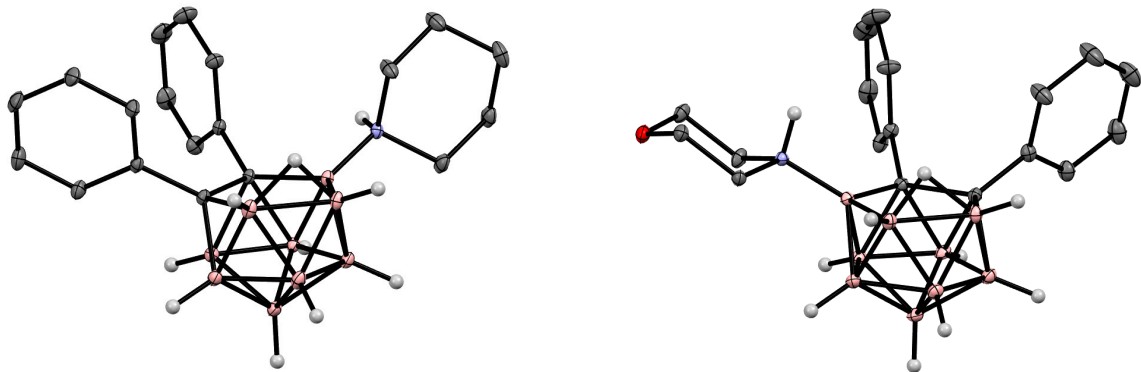

**Figure 16.** Crystal molecular structures of 9-(CH$_2$)$_5$NH-7,8-Ph$_2$-7,8-C$_2$B$_9$H$_9$ (**left**) and 9-O(CH$_2$CH$_2$)$_2$NH-7,8-Ph$_2$-7,8-C$_2$B$_9$H$_9$ (**right**). Hydrogen atoms of alkyl and aryl substituents are omitted for clarity.

This approach is also applicable to the synthesis of various primary aliphatic and heteroaromatic amines (Scheme 16, Figure 17) [89].

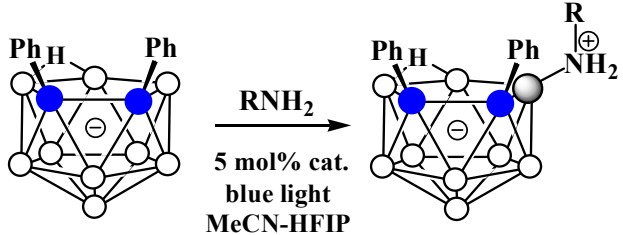

R = *n*-Pr, C$_{12}$H$_{25}$, CH$_2$CH$_2$C$_6$H$_5$, CH$_2$C$_6$H$_5$,
CH$_2$C$_6$H$_4$-*m*-OMe, CH$_2$C$_6$H$_4$-*p*-OMe,
CH$_2$C$_6$H$_4$-*m*-Cl, CH$_2$C$_6$H$_4$-*p*-Cl,
CH$_2$-2-C$_4$H$_3$S, CH$_2$-2-C$_4$H$_3$O

**Scheme 16.** The synthesis of *nido*-carborane-based primary aliphatic and heteroaromatic amines.

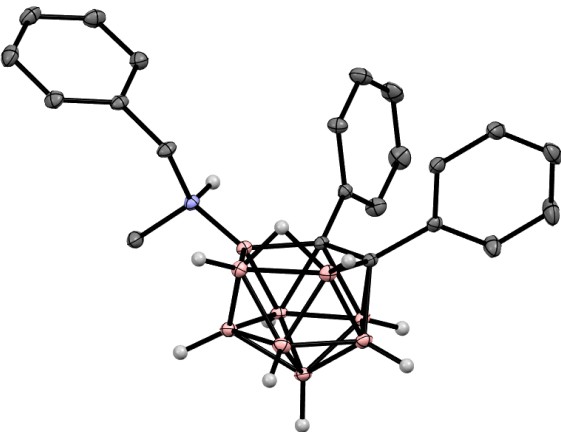

**Figure 17.** Crystal molecular structure of 9-Bn(Me)NH-7,8-Ph$_2$-7,8-C$_2$B$_9$H$_9$. Hydrogen atoms of alkyl and aryl substituents are omitted for clarity.

Halogenation of the 9-trimethylammonium derivative 9-Me$_3$N-7,8-C$_2$B$_9$H$_{11}$ was studied. The reaction with an equimolar amount of Cl$_2$ in dichloromethane at -25°C results in a mixture of 11-Cl-9-Me$_3$N-7,8-C$_2$B$_9$H$_{10}$ and 6-Cl-9-Me$_3$N-7,8-C$_2$B$_9$H$_{10}$ isolated in 14% and 45% yields, respectively, whereas the reaction with an excess of Cl$_2$ under similar conditions gives 6,11-Cl$_2$-9-Me$_3$N-7,8-C$_2$B$_9$H$_9$ isolated in a 26% yield [90]. The reaction of 9-Me$_3$N-7,8-C$_2$B$_9$H$_{11}$ with an equimolar amount of Br$_2$ in dichloromethane at -25°C results in a mixture of 11-Br-9-Me$_3$N-7,8-C$_2$B$_9$H$_{10}$ and 6-Br-9-Me$_3$N-7,8-C$_2$B$_9$H$_{10}$ isolated in 81% and 11% yields, respectively [90]. The reaction with an excess of Br$_2$ in dichloromethane under reflux gave a mixture of 6,11-Br$_2$-9-Me$_3$N-7,8-C$_2$B$_9$H$_9$] and 1,6,11-Br$_3$-9-Me$_3$N-7,8-C$_2$B$_9$H$_8$, both isolated with a yield of 12% (Figure 18) [90]. The reaction of 9-Me$_3$N-7,8-C$_2$B$_9$H$_{11}$ with I$_2$ in acetic acid under reflux leads to 11-I-9-Me$_3$N-7,8-C$_2$B$_9$H$_{10}$ (Figure 18) as a single product [90].

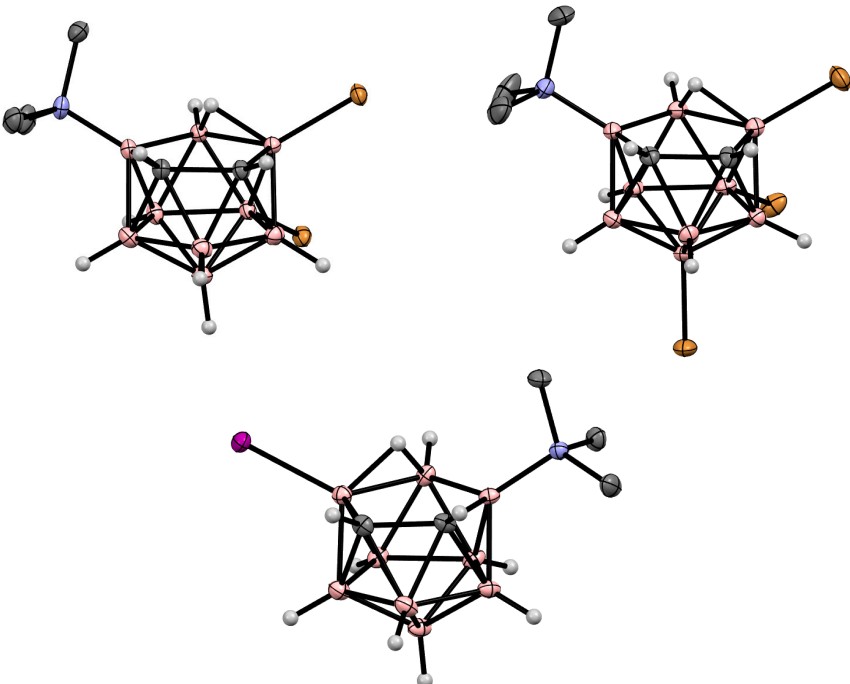

**Figure 18.** Crystal molecular structures of [6,11-Br$_2$-9-Me$_3$N-7,8-C$_2$B$_9$H$_9$] (**top left**), [1,6,11-Br$_3$-9-Me$_3$N-7,8-C$_2$B$_9$H$_8$] (**top right**), and [11-I-9-Me$_3$N-7,8-C$_2$B$_9$H$_{10}$] (**bottom**). Hydrogen atoms of methyl groups are omitted for clarity.

A convenient method for the functionalization of *nido*-carborane, leading to the formation of symmetrically substituted derivatives with a B-N bond, is the synthesis and subsequent modification of its nitrilium derivatives. The first nitrilium derivatives of *nido*-carboranes were synthesized via reactions of the potassium salts of the parent *nido*-carborane and its 7,8-dimethyl derivative with acetonitrile in the presence of FeCl$_3$. In both cases, the nitrilium derivatives were obtained as mixtures of asymmetrically and symmetrically substituted isomers 9-MeC≡N-7,8-R$_2$-7,8-C$_2$B$_9$H$_9$ (R = H, Me) [73]. Later, it was found that the reaction of (Me$_4$N)[7,8-C$_2$B$_9$H$_{12}$] with AlCl$_3$ in acetonitrile in the presence of acetone led solely to the symmetric product [10-MeC≡N-7,8-C$_2$B$_9$H$_{11}$]; however, the product yield was rather low [78]. Recently it was found that the reaction of the potassium salt of *nido*-caborane K [7,8-C$_2$B$_9$H$_{12}$] with HgCl$_2$ in a mixture of refluxing acetonitrile or propionitrile and benzene also leads selectively to the corresponding symmetrically substituted nitrilium derivatives 10-RC≡N-7,8-C$_2$B$_9$H$_{11}$ (R = Me, Et) in close to quantitative yields. Hydrolysis of 10-EtC≡N-7,8-C$_2$B$_9$H$_{11}$ leads to the protonated iminol 10-EtC(OH)=HN-7,8-C$_2$B$_9$H$_{11}$, which upon treatment with triethylamine, gives the corresponding amide (Et$_3$NH)[10-EtC(O)HN-7,8-C$_2$B$_9$H$_{11}$] (Scheme 17) [91].

**Scheme 17.** The synthesis of 10-RC≡N-7,8-C$_2$B$_9$H$_{11}$ (R = Me, Et) and some of their reactions.

A similar approach can also be applied to *C*-substituted derivatives of *nido*-carborane. The reaction of K[7-BnOOCCH$_2$-7,8-C$_2$B$_9$H$_{11}$] with acetonitrile in refluxing benzene in the presence of HgCl$_2$, followed by hydrolysis of the resulting nitrilium derivative and acid hydrolysis of the amide, produces the corresponding ammonium derivative of *nido*-carborane [7-HOOCCH$_2$-10-NH$_3$-7,8-C$_2$B$_9$H$_{10}$] [92].

The reactions of 10-EtC≡N-7,8-C$_2$B$_9$H$_{11}$ with alcohols (MeOH, EtOH, *i*-PrOH, and *n*-BuOH) and thiols (EtSH, *n*-BuSH, and *n*-HxSH) result in the corresponding imidates 10-EtC(OR)=HN-7,8-C$_2$B$_9$H$_{11}$ and thioimidates 10-EtC(SR)=HN-7,8-C$_2$B$_9$H$_{11}$ as mixtures of *E*- and *Z*-isomers, which can be separated using column chromatography on silica (Scheme 18, Figure 19) [91].

**R = Me, Et, *i*-Pr, *n*-Bu**            **R = Et, *n*-Bu, *n*-Hx**

**Scheme 18.** The reactions of 10-EtC≡N-7,8-C$_2$B$_9$H$_{11}$ with alcohols and thiols.

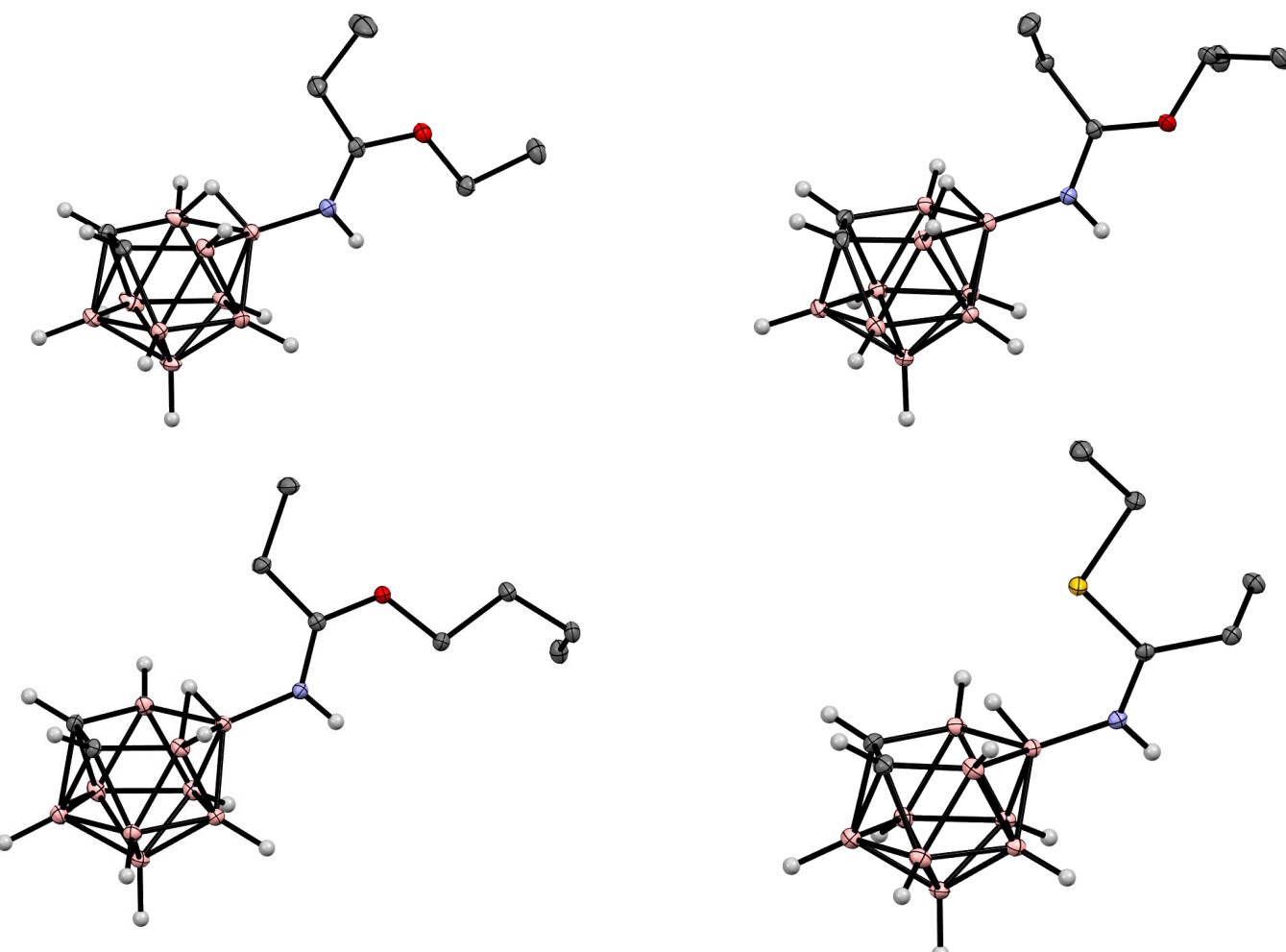

**Figure 19.** Crystal molecular structures of 10-(*E*)-EtC(OEt)=HN-7,8-$C_2B_9H_{11}$ (**top left**), 10-(*E*)-EtC(O$^i$Pr)=HN-7,8-$C_2B_9H_{11}$ (**top right**), 10-(*E*)-EtC(OBu)=HN-7,8-$C_2B_9H_{11}$ (**bottom left**), and 10-(*Z*)-EtC(SEt)=HN-7,8-$C_2B_9H_{11}$ (**bottom right**). Hydrogen atoms of alkyl substituents are omitted for clarity.

In a similar way, the reactions of 10-EtC≡N-7,8-$C_2B_9H_{11}$ with primary amines (MeNH$_2$, EtNH$_2$, *n*-PrNH$_2$, *i*-BuNH$_2$, BnNH$_2$, PhNH$_2$, HOCH$_2$CH$_2$NH$_2$, HOCH$_2$CH$_2$CH$_2$NH$_2$, MeOCH$_2$CH$_2$NH$_2$, and Me$_2$NCH$_2$CH$_2$NH$_2$) lead to the corresponding amidines 10-EtC(NHR)=HN-7,8-$C_2B_9H_{11}$ as mixtures of *E*- and *Z*-isomers (Scheme 19) [93,94]. The obtained carboranyl amidines were shown to be promising ligands for the synthesis of various metallacarboranes [94–96]. The reaction of 10-EtC≡N-7,8-$C_2B_9H_{11}$ with ethylenediamine proceeds with the elimination of imidazoline, resulting in the 10-ammonium derivative 10-H$_3$N-7,8-$C_2B_9H_{11}$ (Scheme 19) [94]. The reactions of 10-EtC≡N-7,8-$C_2B_9H_{11}$ with secondary amines (Me$_2$NH, Et$_2$NH, piperidine, and morpholine) produce the corresponding amidines 10-EtC(NR$_2$)=HN-7,8-$C_2B_9H_{11}$ as single *E*-isomers (Scheme 19, Figure 20) [93].

The reaction of the potassium salt of *nido*-caborane K[7,8-$C_2B_9H_{12}$] with bis(2-cyanoethyl) ether in the presence of HgCl$_2$ results in the corresponding nitrilium derivative 10-NCCH$_2$CH$_2$OCH$_2$CH$_2$C≡N-7,8-$C_2B_9H_{11}$; however, treatment of the iminol formed after its hydrolysis with triethylamine unexpectedly leads to side-chain shortening with the formation of the acrylamide derivative (Et$_3$NH)[10-CH$_2$=CHC(O)HN-7,8-$C_2B_9H_{11}$] (Scheme 20) [97].

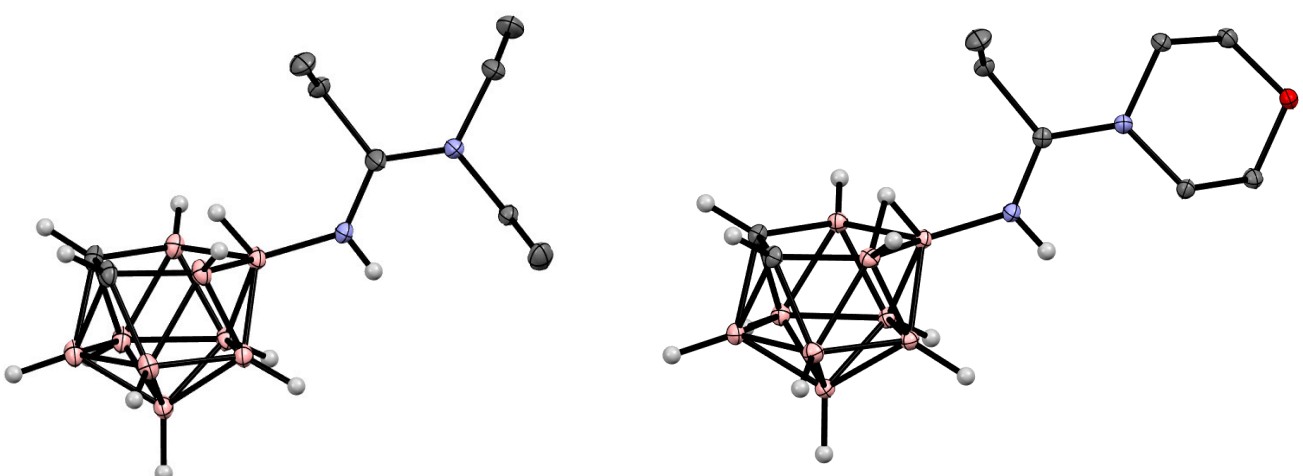

**Scheme 19.** The reactions of 10-EtC≡N-7,8-C$_2$B$_9$H$_{11}$ with primary and secondary amines and ethylenediamine.

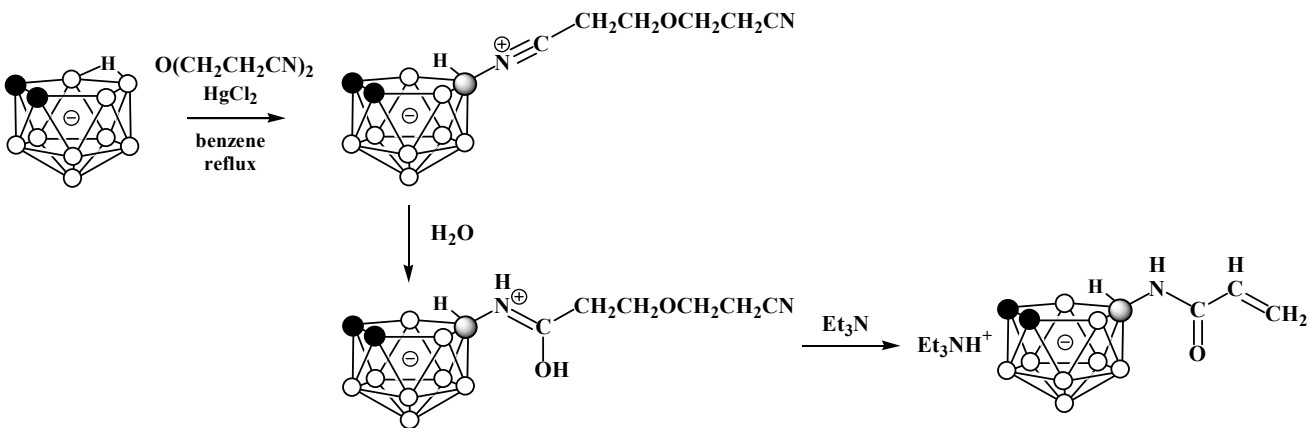

**Figure 20.** Crystal molecular structures of 10-(*E*)-EtC(NEt$_2$)=HN-7,8-C$_2$B$_9$H$_{11}$ (**left**) and 10-(*E*)-EtC(N(CH$_2$CH$_2$)$_2$O)=HN-7,8-C$_2$B$_9$H$_{11}$ (**right**). Hydrogen atoms of alkyl substituents are omitted for clarity.

**Scheme 20.** The synthesis of 10-NCCH$_2$CH$_2$OCH$_2$CH$_2$C≡N-7,8-C$_2$B$_9$H$_{11}$ and some of its reactions.

Surprisingly, the reactions of 10-NCCH$_2$CH$_2$OCH$_2$CH$_2$C≡N-7,8-C$_2$B$_9$H$_{11}$ with alcohols and thiols proceed in a different manner: the reactions with thiols lead to the expected thioimidates, 10-NCCH$_2$CH$_2$OCH$_2$CH$_2$C(SR)=HN-7,8-C$_2$B$_9$H$_{11}$ (R = Et, Bu), as mixtures of *E*- and *Z*-isomers, which can be separated using column chromatography on silica, whereas the reactions with alcohols result in side-chain shortening with the formation of a mixture of two imidates, 10-CH$_2$=CHC(OR)=HN-7,8-C$_2$B$_9$H$_{11}$ and 10-HOCH$_2$CH$_2$C(OR)=HN-7,8-C$_2$B$_9$H$_{11}$ (R = Me, Et, *i*-Pr, *n*-Bu) (Scheme 21, Figure 21). The reaction with diethylamine gives amidine 10-HOCH$_2$CH$_2$C(NEt$_2$)=HN-7,8-C$_2$B$_9$H$_{11}$ (Scheme 21) [97].

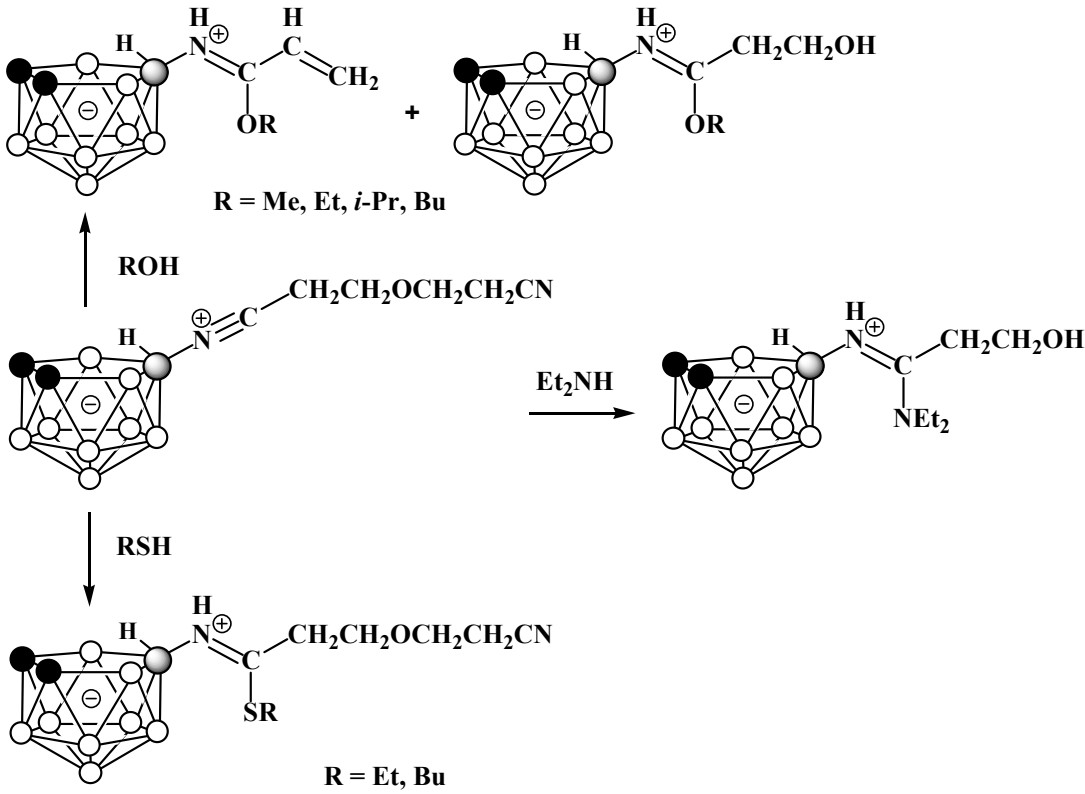

R = Me, Et, *i*-Pr, Bu

R = Et, Bu

**Scheme 21.** Reactions of 10-NCCH$_2$CH$_2$OCH$_2$CH$_2$C≡N-7,8-C$_2$B$_9$H$_{11}$ with alcohols, thiols, and diethylamine.

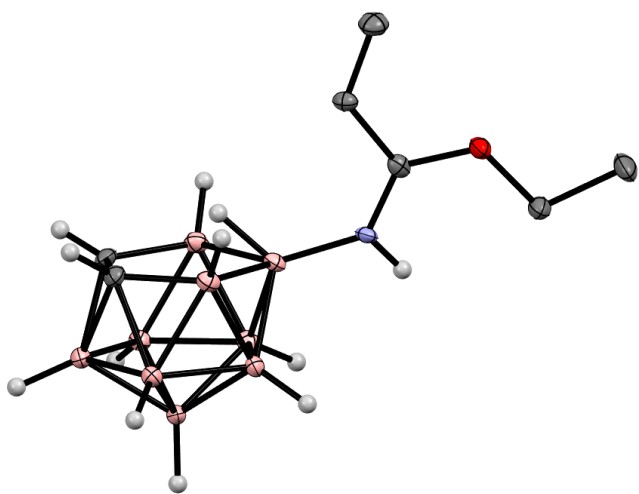

**Figure 21.** Crystal molecular structure of 10-(*E*)-H$_2$C=CHC(OEt)=HN-7,8-C$_2$B$_9$H$_{11}$. Hydrogen atoms of alkyl substituents are omitted for clarity.

Amidine 10-CH$_3$C(NCPh$_2$)=HN-7,8-C$_2$B$_9$H$_{11}$ (Figure 22) was obtained from the reaction of the tetramethylammonium salt of *nido*-carborane with benzophenone imine in acetonitrile in the presence of acetyl chloride. It is assumed that the reaction proceeds through the formation of the acetonitrilium derivative of *nido*-carborane, followed by the addition of the imine to the activated C≡N triple bond [98].

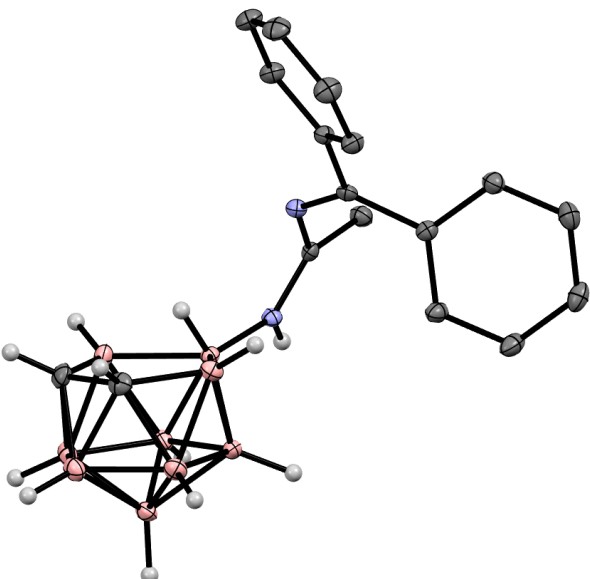

**Figure 22.** Crystal molecular structure of 10-CH$_3$C(NCPh$_2$)=HN-7,8-C$_2$B$_9$H$_{11}$. Hydrogen atoms of methyl and phenyl groups are omitted for clarity.

The 3-Ammonium derivative of *nido*-carborane 3-H$_3$N-7,8-C$_2$B$_9$H$_{11}$ was prepared via the deboronation of 3-amino-*ortho*-carborane 3-H$_2$N-1,2-C$_2$B$_{10}$H$_{11}$ with an alkali in refluxing ethanol (Scheme 22, Figure 23) [99,100]. The same approach can be used for the synthesis of the 3-ammonium derivatives of *C*-substituted *nido*-carboranes 3-H$_3$N-7-R-7,8-C$_2$B$_9$H$_{12}$ (R = *i*-Pr, CH$_2$COOH, CH$_2$COOBn) (Figure 23) [99,101]. The 3-dimethylammonium and 3-benzylammonium derivatives of *nido*-carborane were prepared via the deboronation of the corresponding derivatives of *ortho*-carborane (Scheme 22) [99]. The 3-trimethylammonium derivative of *nido*-carborane 3-Me$_3$N-7,8-C$_2$B$_9$H$_{11}$ was obtained via the treatment of the 3-ammonium derivative with methyl iodide in the presence of K$_2$CO$_3$ (Scheme 22) [99].

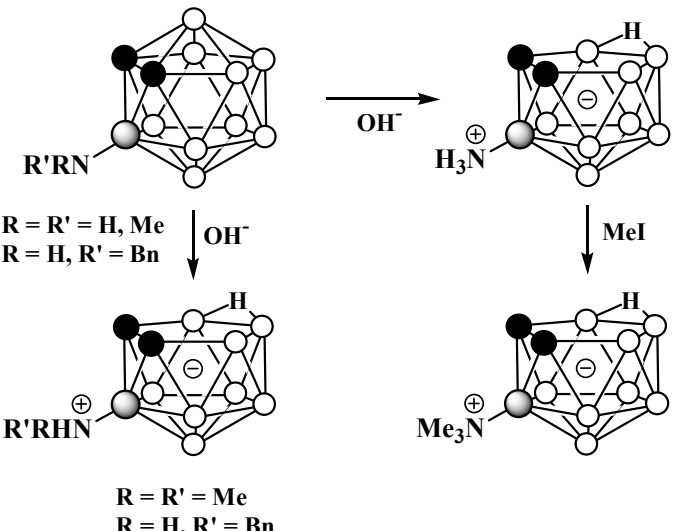

**Scheme 22.** The synthetic routes to ammonium derivatives of *nido*-carborane.

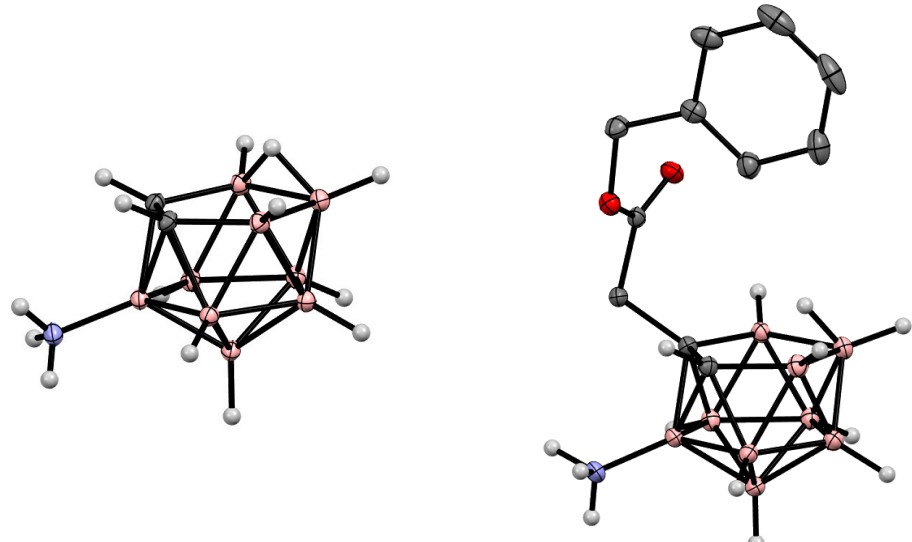

**Figure 23.** Crystal molecular structures of 3-H$_3$N-7,8-C$_2$B$_9$H$_{11}$ (**left**) and 3-H$_3$N-7-BnOOCCH$_2$-7,8-C$_2$B$_9$H$_{10}$ (**right**). Hydrogen atoms of alkyl substituents are omitted for clarity.

Heating 9-amido-*ortho*-carboranes 9-RCONH-1,2-C$_2$B$_{10}$H$_{11}$ (R = H, Alk, or Ar) with 10 mol.% of Pd(OAc)$_2$, 2 equiv. of AgOAc, and 2 equiv. of K$_2$CO$_3$ in 1,4-dioxane at 100°C results in successive deboronation and cyclization reactions with the formation of the corresponding *N*-protonated *nido*-carborane fused oxazoles 5,10-μ-RCNHO-7,8-C$_2$B$_9$H$_{10}$. The reaction is applicable to various *N*-carboranylamides including formamide and alkyl- and arylamides, as well as to C,C'-substituted *N*-carboranylamides (Scheme 23, Figure 24) [102]. Interestingly, in the case of 9-PhCONH-1,2-μ-C$_6$H$_4$(CH$_2$)$_2$-1,2-C$_2$B$_{10}$H$_9$, the reaction results in a mixture of the 5,10-μ-PhCNHO-7,8-μ-C$_6$H$_4$(CH$_2$)$_2$-7,8-C$_2$B$_9$H$_8$ and 5,9-μ-PhCNHO-7,8-μ-C$_6$H$_4$(CH$_2$)$_2$-7,8-C$_2$B$_9$H$_8$ isomers (Figure 24) in a 1:1 ratio [102]. It was found that the Pd catalyst plays an important role in promoting deboronation, while the presence of AgOAc is critical for the cyclization reaction [102].

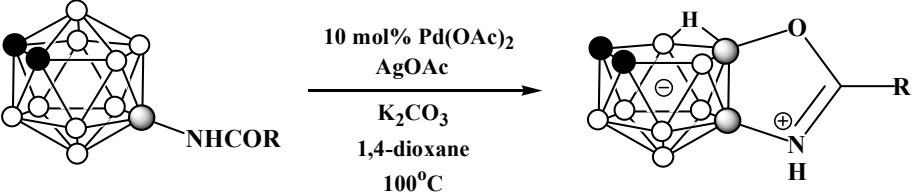

**R = H, Me, *i*-Pr, *c*-C$_3$H$_5$, *c*-C$_6$H$_1$, Bn, CH$_2$-1-Naph,**
**Ph, C$_6$H$_4$-4-Me, C$_6$H$_4$-3-Me, C$_6$H$_4$-4-OMe, C$_6$H$_4$-2-OMe,**
**C$_6$H$_3$-3,5-(OMe)$_2$, C$_6$H$_4$-4-Cl, C$_6$H$_4$-4-F, C$_6$H$_4$-4-CF$_3$**

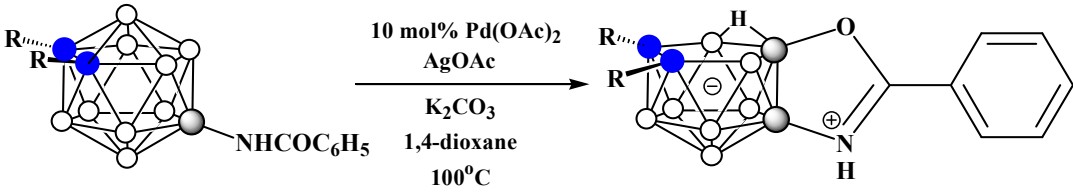

**R = R = Me, Et; RR = 1,2-(CH$_2$)$_2$C$_6$H$_4$**

**Scheme 23.** The synthetic route to obtain *N*-protonated *nido*-carborane oxazoles.

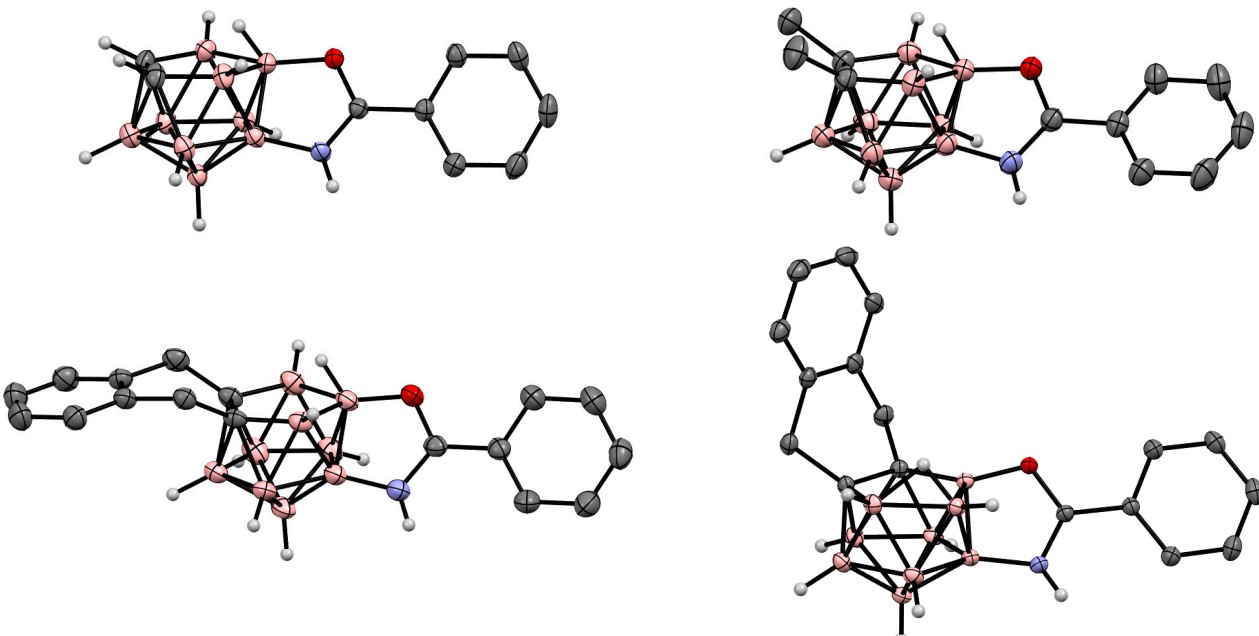

**Figure 24.** Crystal molecular structures of 5,10-μ-PhCNHO-7,8-C$_2$B$_9$H$_{10}$ (**top left**), 5,10-μ-PhCNHO-7,8-Me$_2$-7,8-C$_2$B$_9$H$_8$ (**top right**), 5,10-μ-PhCNHO-7,8-μ-C$_6$H$_4$(CH$_2$)$_2$-7,8-C$_2$B$_9$H$_8$ (**bottom left**), and 5,9-μ-PhCNHO-7,8-μ-C$_6$H$_4$(CH$_2$)$_2$-7,8-C$_2$B$_9$H$_8$ (**bottom right**). Hydrogen atoms of alkyl and aryl substituents are omitted for clarity.

## 3. Charge-Compensated Derivatives of *Nido*-Carborane with Boron–Phosphorus Bond

Unlike derivatives with a boron–nitrogen bond, derivatives of *nido*-carboranes with a boron–phosphorus bond can be prepared using electrophilic substitution reactions. Heating the potassium salt of *nido*-carborane K[7,8-C$_2$B$_9$H$_{12}$] with Ph$_2$PCl in tetrahydrofuran at reflux leads to the *P*-protonated diphenylphosphonium derivative 9-Ph$_2$PH-7,8-C$_2$B$_9$H$_{11}$, which can be alkylated with MeI under reflux in ethanol in the presence of EtONa as a base to give the methyldiphenylphosphonium derivative 9-MePh$_2$P-7,8-C$_2$B$_9$H$_{11}$ (Scheme 24, Figure 25) [103,104].

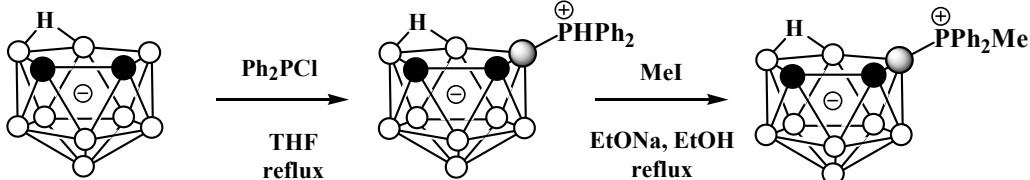

**Scheme 24.** The synthesis of 9-Ph$_2$PH-7,8-C$_2$B$_9$H$_{11}$ and its alkylation.

The symmetrically substituted phosphonium derivatives 10-Ph$_2$PH-7,8-C$_2$B$_9$H$_{11}$ and 10-MePh$_2$P-7,8-C$_2$B$_9$H$_{11}$ (Figure 25) were prepared in a similar way using the disodium dicarbollide salt Na$_2$[7,8-C$_2$B$_9$H$_{11}$] as a starting material (Scheme 25) [104].

Phosphonium derivatives of *nido*-carborane also can be prepared via Lewis-acid-mediated nucleophilic substitution reactions. The reaction of the potassium salt of *nido*-carborane K[7,8-C$_2$B$_9$H$_{12}$] with PPh$_3$ in the presence FeCl$_3$ in benzene at 80 °C leads to a mixture of triphenylphosphonium 9-Ph$_3$P-7,8-C$_2$B$_9$H$_{11}$ and 10-Ph$_3$P-7,8-C$_2$B$_9$H$_{11}$, which were separated using column chromatography on silica (Scheme 26) [103,104].

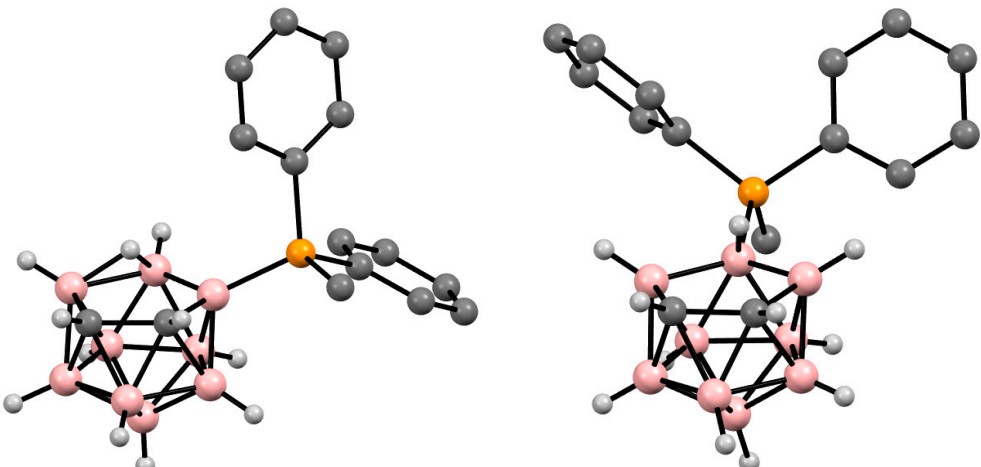

**Figure 25.** Crystal molecular structures of 9-MePh$_2$P-7,8-C$_2$B$_9$H$_{11}$ (**left**) and 10-MePh$_2$P-7,8-C$_2$B$_9$H$_{11}$ (right). Hydrogen atoms of phenyl and methyl groups are omitted for clarity.

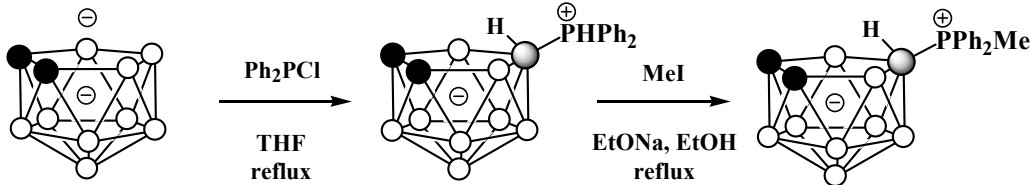

**Scheme 25.** The synthesis of 10-Ph$_2$PH-7,8-C$_2$B$_9$H$_{11}$ and its alkylation.

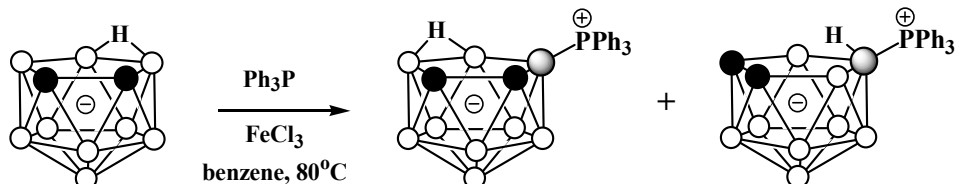

**Scheme 26.** Reaction of *nido*-carborane with triphenylphosphine in the presence of FeCl$_3$.

The asymmetrical triphenylphosphonium derivative 9-Ph$_3$P-7,8-C$_2$B$_9$H$_{11}$ was also obtained via the reaction of triphenylphosphine with the dithallium dicarbollide salt Tl$_2$[7,8-C$_2$B$_9$H$_{11}$] in dichloromethane and in the presence of AgBr at ambient temperature (Scheme 27) [105].

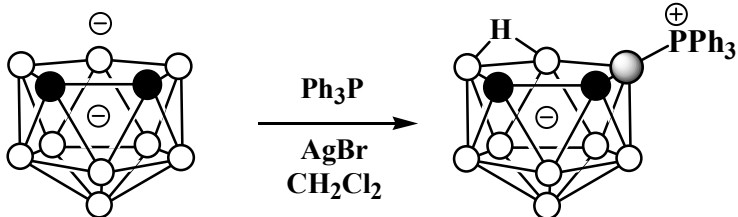

**Scheme 27.** Synthesis of 9-Ph$_3$P-7,8-C$_2$B$_9$H$_{11}$.

Similar to the pyridinium derivatives, a series of asymmetrically substituted phosphonium derivatives 9-R'R$_2$P-7,8-Ph$_2$-7,8-C$_2$B$_9$H$_9$ was prepared via electrocatalyzed B-P oxidative couplings of 7,8-diphenyl-*nido*-carborane with various phosphines and phosphites (Scheme 28, Figure 26) [106].

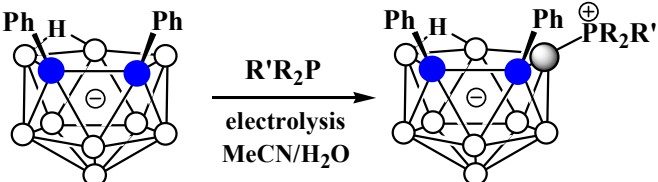

R = R' = Me, Bu, C$_6$H$_5$, C$_6$H$_4$-*p*-Me, C$_6$H$_4$-*m*-Me,
C$_6$H$_4$-*p*-F, C$_6$H$_4$-*m*-F, C$_6$H$_4$-*m*-Cl,
2-C$_4$H$_3$O, 2-C$_4$H$_3$S,
OEt, OBu, OCH$_2$CH$_2$Cl, OCH$_2$CH=CH$_2$;
R = Me, R' = Ph; R = Ph, R' = Me, CH$_2$CH=CH

**Scheme 28.** The electrocatalyzed B-P oxidative coupling of triarylphosphines, diarylalkylphosphines, and trialkylphosphines with 7,8-diphenyl-*nido*-carborane.

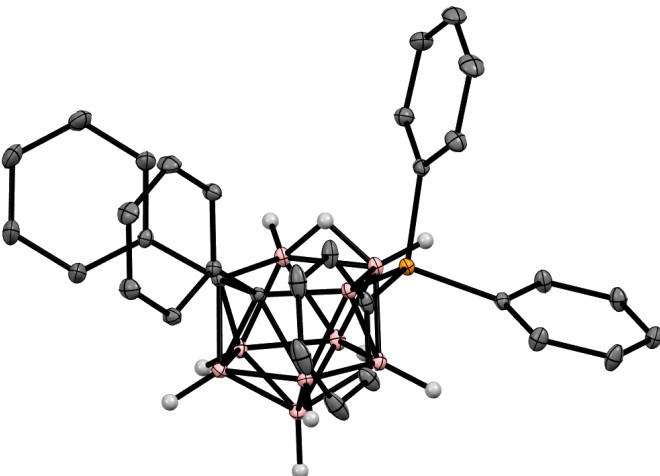

**Figure 26.** Crystal molecular structure of 9-Ph$_3$P-7,8-Ph$_2$-7,8-C$_2$B$_9$H$_9$. Hydrogen atoms of phenyl rings are omitted for clarity.

However, the largest number of phosphonium derivatives of *nido*-carborane was obtained through transition-metal-catalyzed cross-coupling reactions [107]. A series of triphenylphosphonium derivatives of *nido*-carborane, X-Ph$_3$P-7,8-C$_2$B$_9$H$_{11}$ (X = 1, 3, 5, 9), were synthesized via the reactions of the corresponding iodo derivatives with PPh$_3$ in the presence of 10 mol.% of [(Ph$_3$P)$_4$Pd] in 1,4-dioxane at 90 °C. This approach can be used to synthesize derivatives containing substituents in different positions of the *nido*-carborane cage, including the upper and lower belts, as well as the bottom of the basket (Scheme 29) [108].

The 5-triphenyl- and 5-bis(*t*-butyl)phosphonium derivatives of *nido*-carborane 5-Ph$_3$P-7,8-C$_2$B$_9$H$_{11}$ and 5-$^t$Bu$_2$PH-7,8-C$_2$B$_9$H$_{11}$ were obtained in low yields directly by heating 9-iodo-*ortho*-carborane with an excess of AgF and catalytic amounts of [(Ph$_3$P)$_4$Pd] or [($^t$Bu$_3$P)$_2$Pd] in DMF at 140 °C (Scheme 30, Figure 27) [109].

The symmetrically substituted diiodo derivatives [5,6-I$_2$-7,8-C$_2$B$_9$H$_{10}$]$^-$ and [9,11-I$_2$-7,8-C$_2$B$_9$H$_{10}$]$^-$ under the same conditions give the corresponding mono-coupling products 5-I-6-Ph$_3$P-7,8-C$_2$B$_9$H$_{10}$ and 9-I-11-Ph$_3$P-7,8-C$_2$B$_9$H$_{10}$ in good yields, whereas the reaction of the asymmetrical diiodo derivative with substituents at boron atoms in both pentagonal belts [6,9-I$_2$-7,8-C$_2$B$_9$H$_{10}$]$^-$ results in selective coupling at the open pentagonal belt to form 6-I-9-Ph$_3$P-7,8-C$_2$B$_9$H$_{10}$ in a moderate yield (Scheme 31, Figure 28) [108].

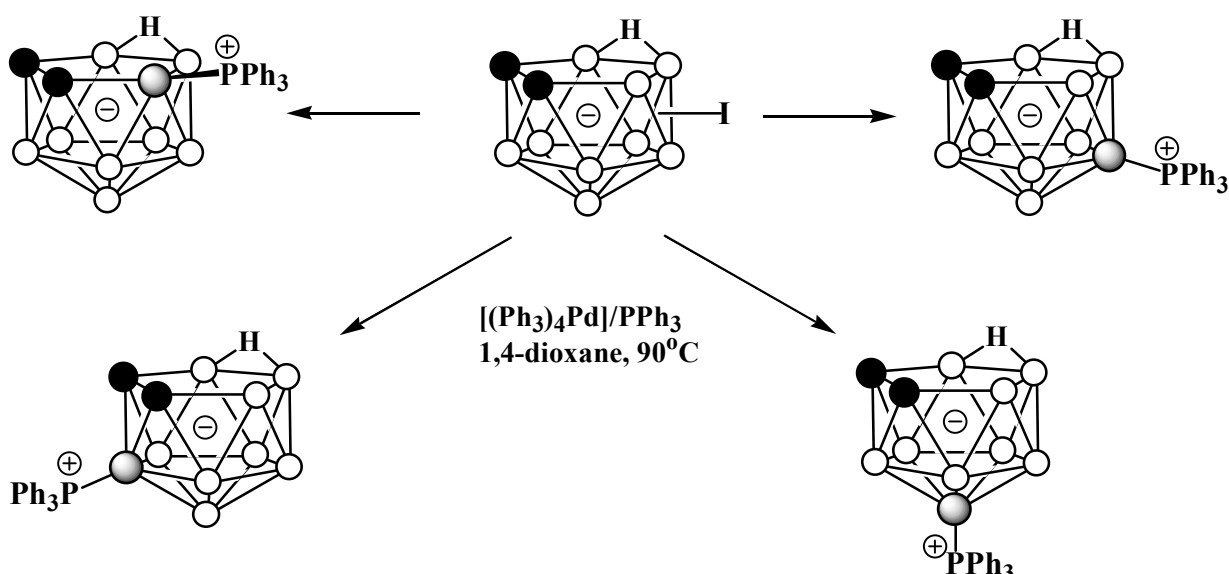

**Scheme 29.** Synthesis of triphenylphosphonium derivatives of *nido*-carborane X-Ph$_3$P-7,8-C$_2$B$_9$H$_{11}$ (X = 1, 3, 5, 9) via transition-metal-catalyzed cross-coupling reactions.

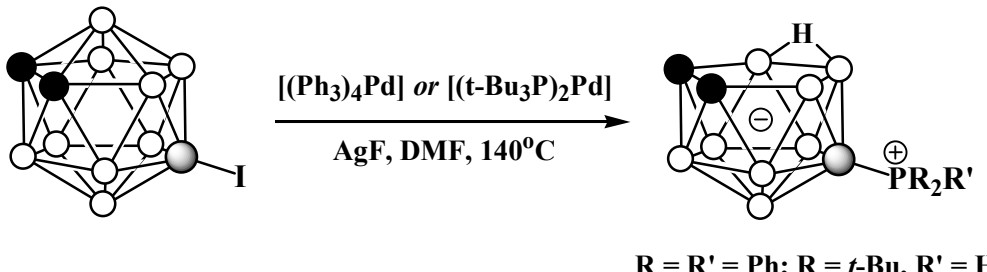

R = R' = Ph; R = *t*-Bu, R' = H

**Scheme 30.** Synthesis of 5-Ph$_3$P-7,8-C$_2$B$_9$H$_{11}$ and 5-$^t$Bu$_2$PH-7,8-C$_2$B$_9$H$_{11}$.

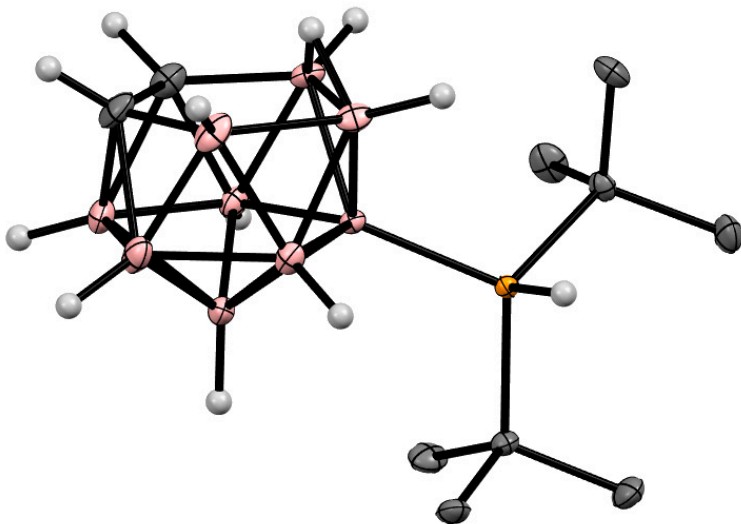

**Figure 27.** Crystal molecular structure of 5-*t*-Bu$_2$HP-7,8-C$_2$B$_9$H$_{11}$. Hydrogen atoms of alkyl groups are omitted for clarity.

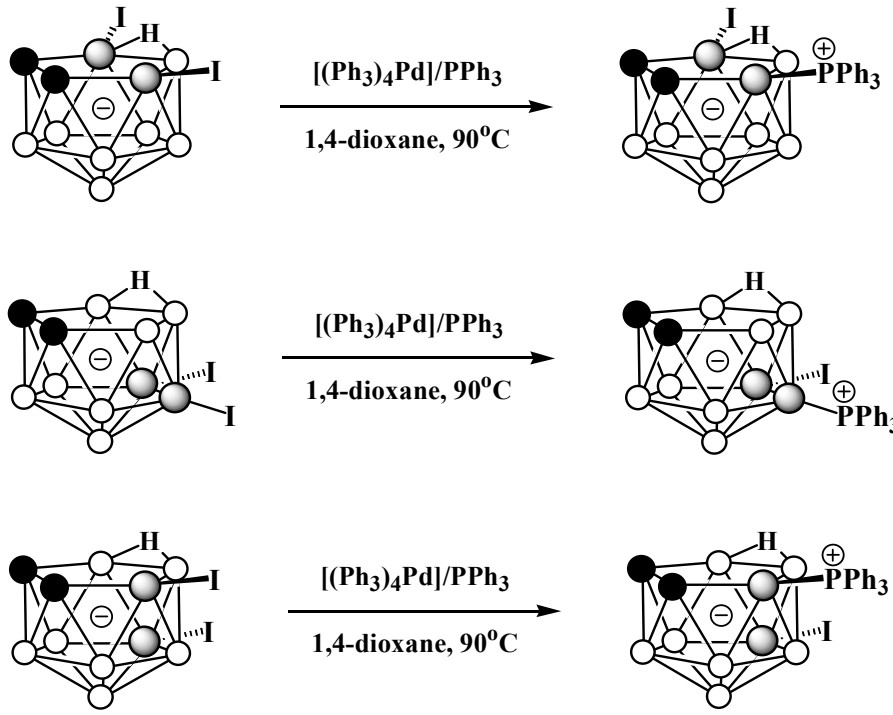

**Scheme 31.** Synthesis of 9-I-11-Ph$_3$P-7,8-C$_2$B$_9$H$_{10}$, 5-I-6-Ph$_3$P-7,8-C$_2$B$_9$H$_{10}$, and 6-I-9-Ph$_3$P-7,8-C$_2$B$_9$H$_{10}$.

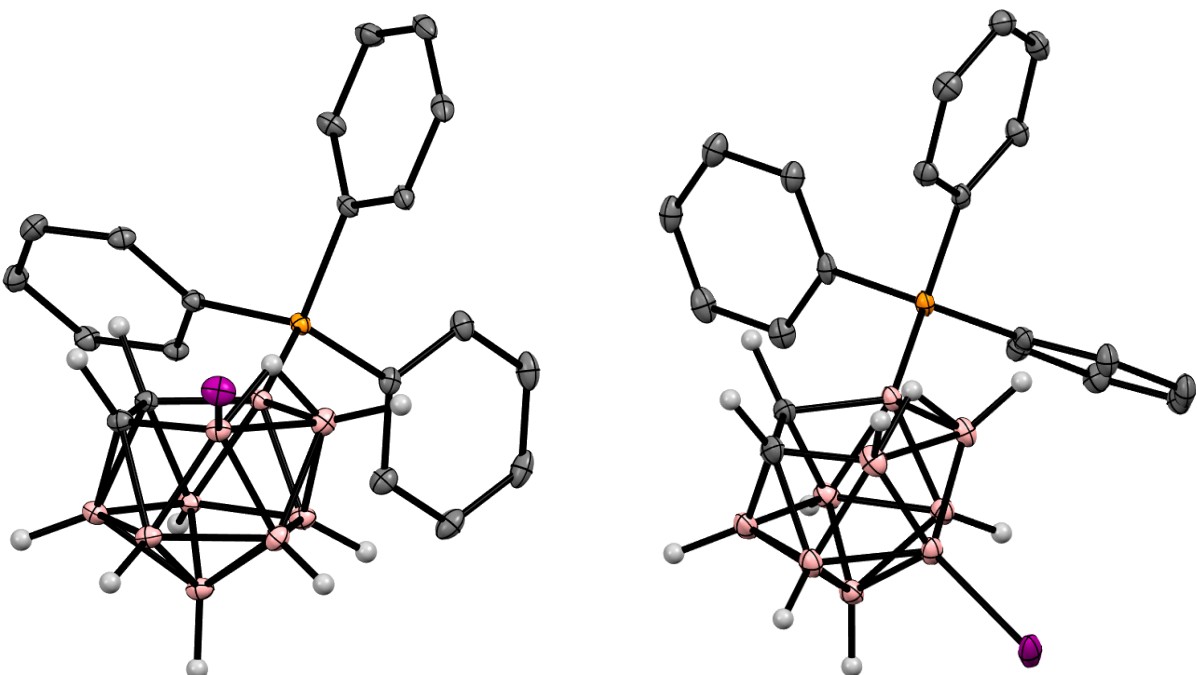

**Figure 28.** Crystal molecular structures of 9-Ph$_3$P-11-I-7,8-C$_2$B$_9$H$_{10}$ (**left**) and 9-Ph$_3$P-6-I-7,8-C$_2$B$_9$H$_{10}$ (**right**). Hydrogen atoms of phenyl groups are omitted for clarity.

The bis(triphenylphosphonium) derivatives 5,6-(Ph$_3$P)$_2$-7,8-C$_2$B$_9$H$_{10}$ and 9,11-(Ph$_3$P)$_2$-7,8-C$_2$B$_9$H$_{10}$ can be prepared in a similar way with the addition of Cs$_2$CO$_3$ as a base to remove the bridging hydrogen (Scheme 32, Figure 29) [108]. 9,11-(Ph$_3$P)$_2$-7,8-C$_2$B$_9$H$_{10}$ can also be obtained in a two-step process through 11-Ph$_3$P-9-I-7,8-C$_2$B$_9$H$_{10}$ (Scheme 32) [108].

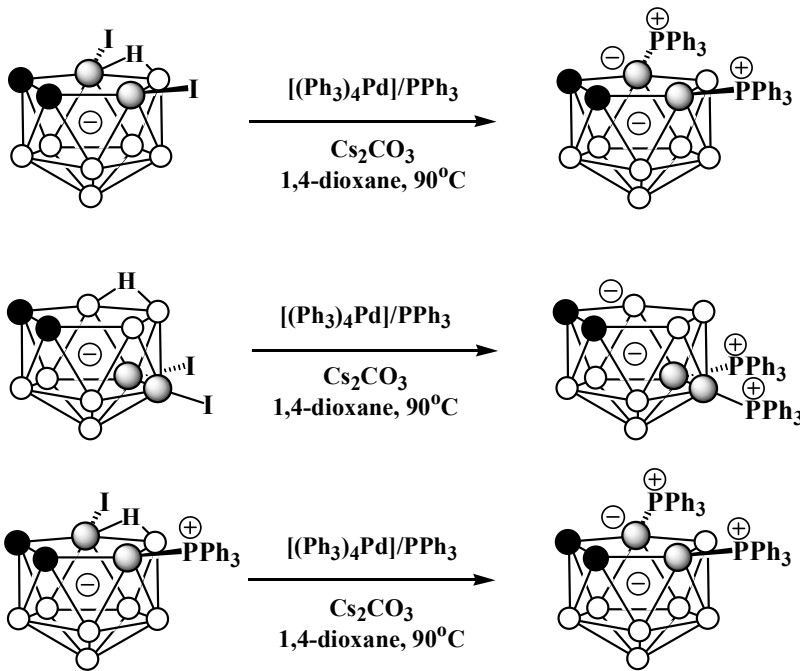

**Scheme 32.** Synthesis of 9,11-(Ph$_3$P)$_2$-7,8-C$_2$B$_9$H$_{10}$ and 5,6-(Ph$_3$P)$_2$-7,8-C$_2$B$_9$H$_{10}$.

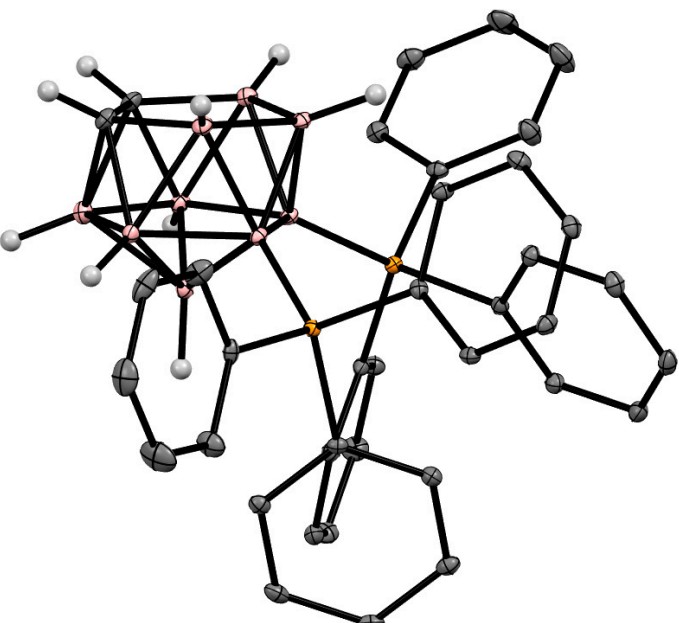

**Figure 29.** Crystal molecular structure of 5,6-(Ph$_3$P)$_2$-7,8-C$_2$B$_9$H$_9$. Hydrogen atoms of phenyl groups are omitted for clarity.

In the case of the 5,6,9-triiodo derivative of *nido*-carborane [5,6,9-I$_3$-7,8-C$_2$B$_9$H$_9$]$^-$, the substitution proceeds selectively in the open pentagonal belt and in the most distant position in the lower pentagonal belt to form 6,9-(Ph$_3$P)$_2$-5-I-7,8-C$_2$B$_9$H$_8$ (Scheme 33, Figure 30). Alternatively, this compound can be obtained via the reaction of the diiodo derivative of *nido*-carborane [5,6-I$_2$-7,8-C$_2$B$_9$H$_{10}$]$^-$ with PPh$_3$ in the presence of a sub-equimolar amount of [(Ph$_3$P)$_4$Pd] in dioxane at 90°C (Scheme 33) [108]. In this case, both BI and BH activation takes place.

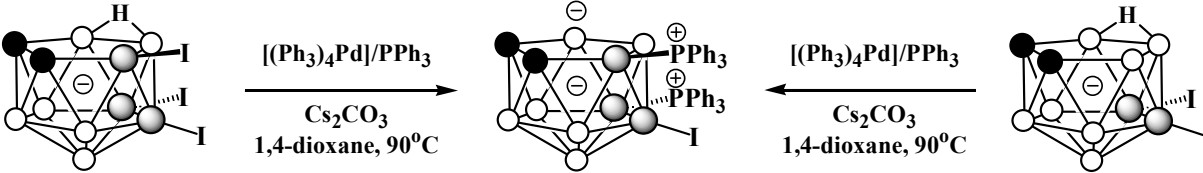

**Scheme 33.** Two synthetic processes to obtain 6,9-(Ph$_3$P)$_2$-5-I-7,8-C$_2$B$_9$H$_8$.

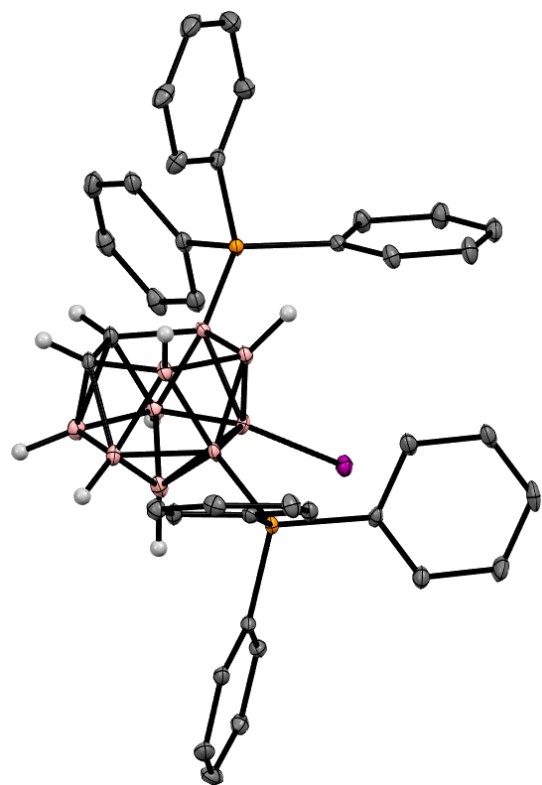

**Figure 30.** Crystal molecular structure of 6,9-(Ph$_3$P)$_2$-5-I-7,8-C$_2$B$_9$H$_8$. Hydrogen atoms of phenyl groups are omitted for clarity.

The reaction of *nido*-carborane with 0.5 equiv. of [(Me$_2$PhP)$_2$PdCl$_2$] in dichloromethane at room temperature followed by heating with NaBH$_4$ in benzene leads to the *BH*-activation of *nido*-carborane with the formation of a mixture of isomeric phosphonium derivatives 9-Me$_2$PhP-7,8-C$_2$B$_9$H$_{11}$ and 10-Me$_2$PhP-7,8-C$_2$B$_9$H$_{11}$ (Scheme 34, Figure 31) [110].

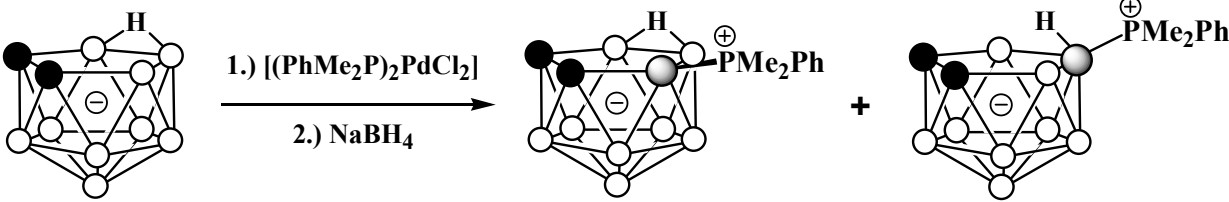

**Scheme 34.** Synthesis of 9-Me$_2$PhP-7,8-C$_2$B$_9$H$_{11}$ and 10-Me$_2$PhP-7,8-C$_2$B$_9$H$_{11}$.

The reaction of *ortho*-carborane with triphenylphosphine in the presence of catalytic amounts of PdCl$_2$ in benzene at 80 °C directly results in the 5-triphenylphosphonium derivative of *nido*-carborane 5-Ph$_3$P-7,8-C$_2$B$_9$H$_{11}$ in a moderate yield (Scheme 35, Figure 32) [111].

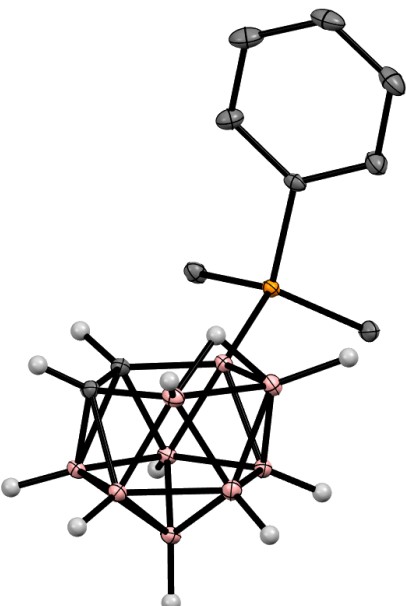

**Figure 31.** Crystal molecular structure of 9-PhMe$_2$P-7,8-C$_2$B$_9$H$_{11}$. Hydrogen atoms of substituents are omitted for clarity.

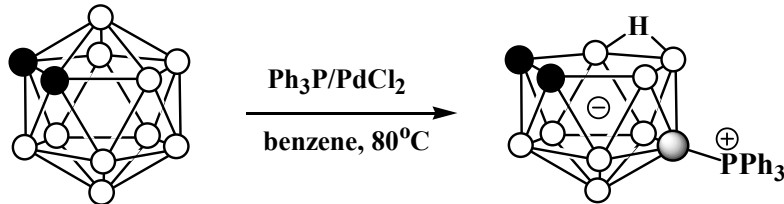

**Scheme 35.** Synthesis of 5-Ph$_3$P-7,8-C$_2$B$_9$H$_{11}$.

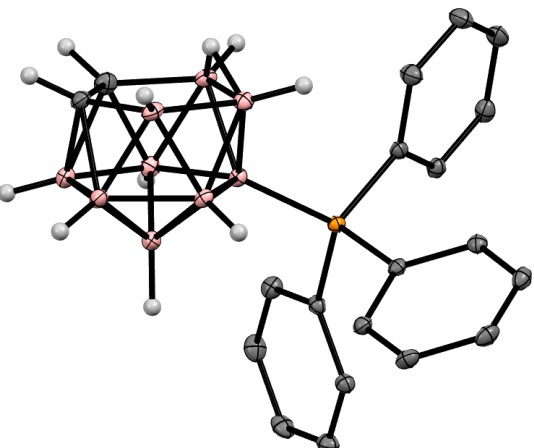

**Figure 32.** Crystal molecular structure of 5-Ph$_3$P-7,8-C$_2$B$_9$H$_{11}$. Hydrogen atoms of phenyl groups are omitted for clarity.

The reactions of sulfide [7,8-μ-S(CH$_2$)$_3$-7,8-C$_2$B$_9$H$_{10}$]$^-$ with 1 equiv. of palladium(II) complexes [L$_2$PdCl$_2$] (L = PPh$_3$, PPh$_2$Me) in boiling ethanol led to selective *BH*-activation at the nearest-t-sulfur boron atom of the upper pentagonal belt with the formation of 11-R′R$_2$P-7,8-μ-(S(CH$_2$)$_3$)-7,8-C$_2$B$_9$H$_9$ (Scheme 36, Figure 33) [112]. It is assumed that the alkyl sulfide substituent plays the role of a directing group.

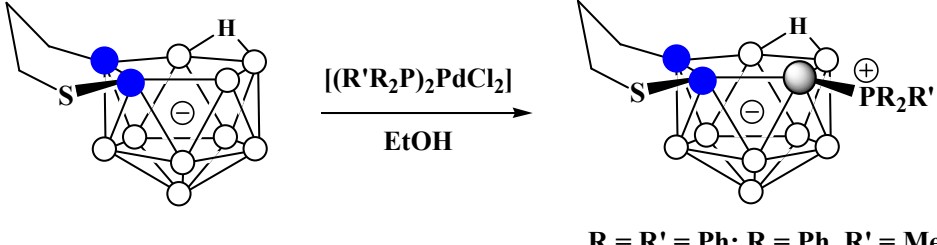

**Scheme 36.** The synthesis of 11-R′R$_2$P-7,8-μ-(S(CH$_2$)$_3$)-7,8-C$_2$B$_9$H$_9$.

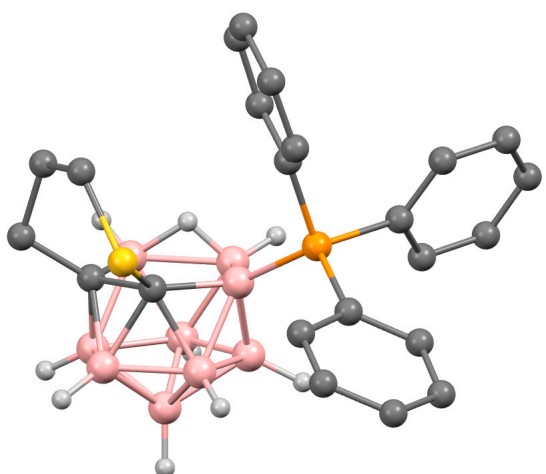

**Figure 33.** Crystal molecular structure of 9-Ph$_3$P-7,8-(μ-(CH$_2$)$_3$S)-7,8-C$_2$B$_9$H$_9$. Hydrogen atoms of organic substituents are omitted for clarity.

The 2-pyridyl substituent can also act as a directing group. The reactions 2-pyridyl-substituted *nido*-carborane [7-(2′-Py)-7,8-C$_2$B$_9$H$_{11}$]$^-$, formed by heating the corresponding *ortho*-carborane 1-(2′-Py)-1,2-C$_2$B$_9$H$_{12}$ in aqueous acetonitrile, with various phosphines in the presence of catalytic amounts of PdCl$_2$ in a mixture of toluene, water, and acetonitrile at 120°C lead to the corresponding phosphonium derivatives 11-R′R$_2$P-7-(2′-Py)-7,8-C$_2$B$_9$H$_{10}$ (Scheme 37, Figure 34). The reaction is tolerant to the presence of alkyl and aryl substituents at the second carbon atom of the *nido*-carborane cage (Scheme 37) [113].

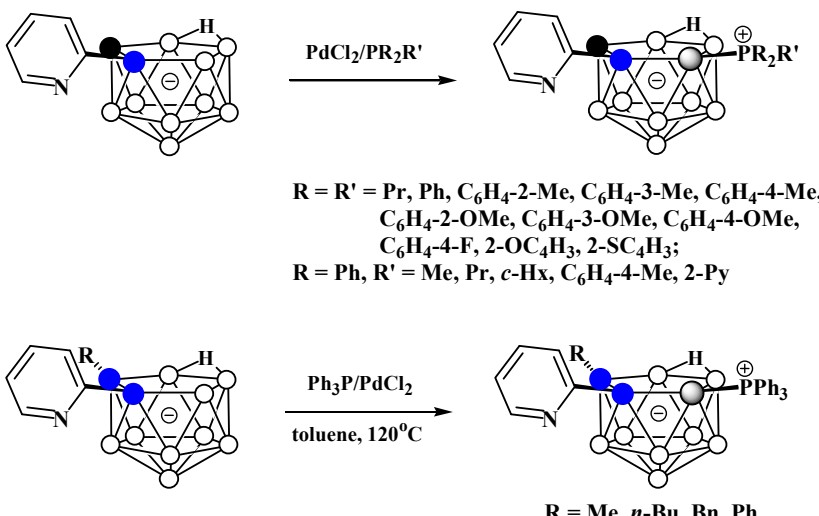

**Scheme 37.** Synthesis of triarylphosphonium, diarylalkylphosphonium, and trialkylphosphonium derivatives 11-R′R$_2$P-7-(2′-Py)-7,8-C$_2$B$_9$H$_{10}$ starting from 2-pyridyl-substituted *nido*-carborane.

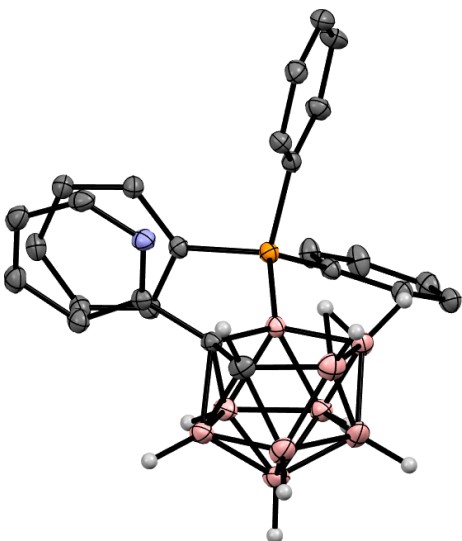

**Figure 34.** Crystal molecular structure of 11-Ph$_3$P-7-(NC$_5$H$_4$-2′-)-7,8-C$_2$B$_9$H$_{10}$. Hydrogen atoms of organic substituents are omitted for clarity.

Various substituted pyridines (5-, 6-methyl-, and 4-trifluoromethylpyridine), as well as 2-benzoxazolyl and diphenylphosphine groups, can be used as guide groups as well (Scheme 38) [113].

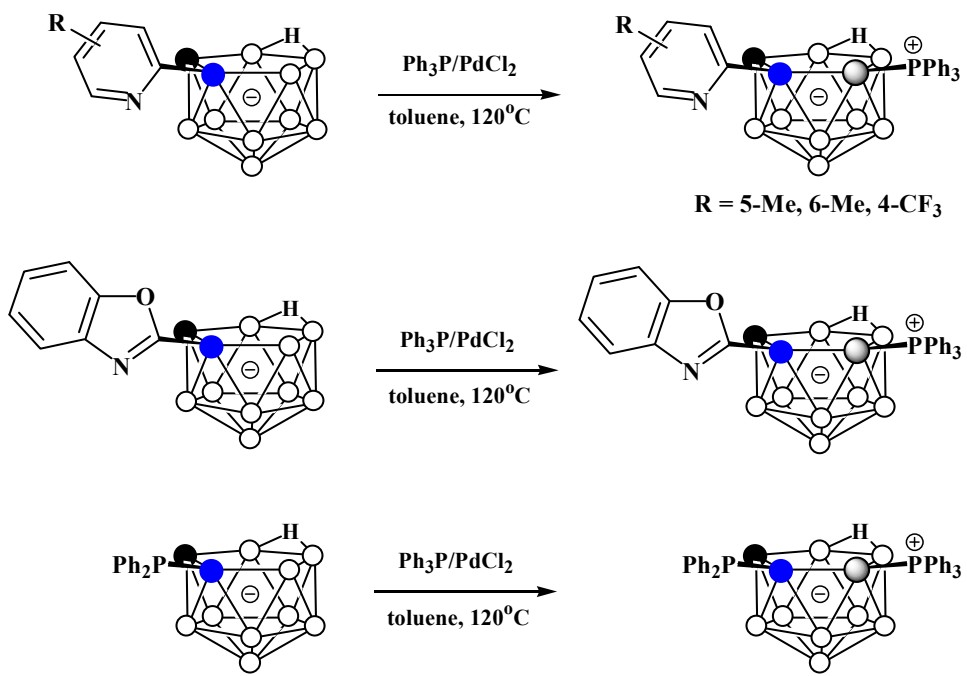

**Scheme 38.** Synthesis of triphenylphosphonium derivatives starting from different *C*-substituted *nido*-carborane pyridines as well as from the *C*-2-benzoxazolyl and *C*-diphenylphosphine derivatives.

The reaction of [7-(NC$_5$H$_3$-3′-Me-2′-)-7,8-C$_2$B$_9$H$_{11}$]$^-$ with triphenylphosphine in the presence of 10 mol. % of PdCl$_2$ unexpectedly led to a mixture of 11- and 2-triphenylphosphonium derivatives (Scheme 39, Figure 35) isolated in 31% and 25% yields, respectively. It was found that the addition of 20 mol. % of 2-amino-5-methyl-pyridine as a ligand leads to a change in the product ratio, with an increase in the 2-isomer yield of up to 64% (the yield of the 11-isomer is 14% in this case) [113,114].

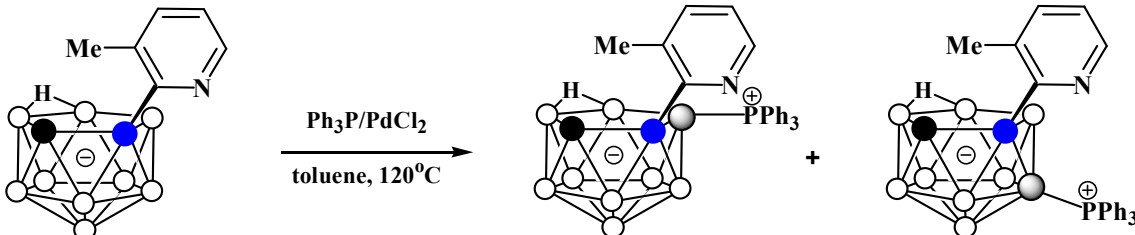

**Scheme 39.** Reaction of $[7-(NC_5H_3-3'-Me-2'-)-7,8-C_2B_9H_{11}]^-$ with triphenylphosphine in the presence of $PdCl_2$.

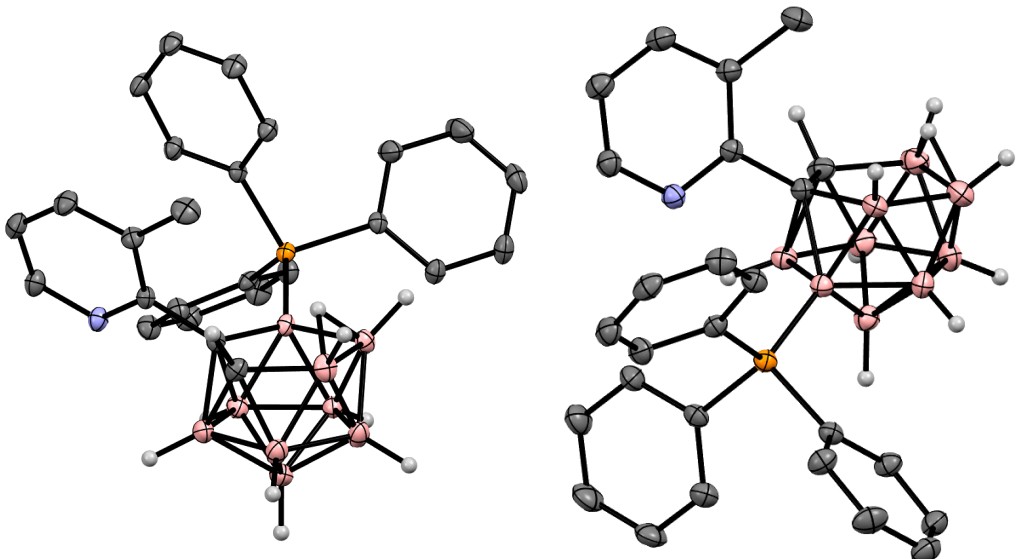

**Figure 35.** Crystal molecular structures of $11-Ph_3P-7-(NC_5H_3-3'-Me-2'-)-7,8-C_2B_9H_{10}$ (**left**) and $2-Ph_3P-7-(NC_5H_3-3'-Me-2'-)-7,8-C_2B_9H_{10}$ (**right**). Hydrogen atoms of organic substituents are omitted for clarity.

This reaction is applicable to a wide variety of phosphines; however, the expected 2-phosphonium derivatives are formed in rather low yields varying from 12 to 45% (Scheme 40) [114].

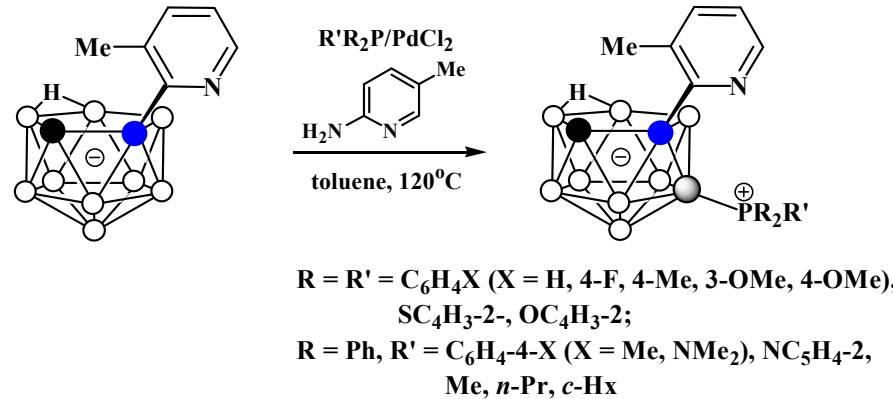

$R = R' = C_6H_4X$ (X = H, 4-F, 4-Me, 3-OMe, 4-OMe),
$SC_4H_3$-2-, $OC_4H_3$-2;
R = Ph, $R' = C_6H_4$-4-X (X = Me, $NMe_2$), $NC_5H_4$-2,
Me, *n*-Pr, *c*-Hx

**Scheme 40.** Reaction of $[7-(NC_5H_3-3'-Me-2'-)-7,8-C_2B_9H_{11}]^-$ with triarylphosphines and diarylalkylphosphines in the presence of $PdCl_2$.

Another group that directs the substitution to position two of the *nido*-carborane cage under similar conditions is the isoquinolin-1-yl group (Scheme 41). The reaction is tolerant

to the presence of a substituent at the second carbon atom of the *nido*-carborane cage (Scheme 42) [114].

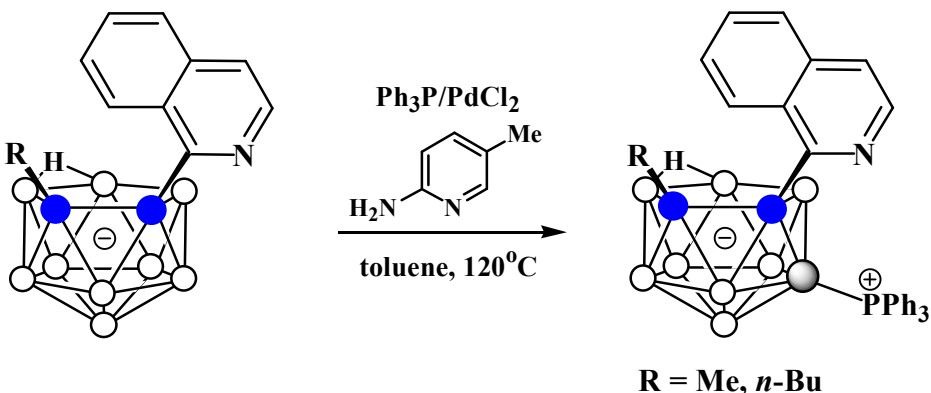

**R = R' = C₆H₄-4-X (X = H, F, Me, OMe);**
**R = Ph, R' = *n*-Pr**

**Scheme 41.** Reaction of *C*-substituted isoquinolin-1-yl derivative of *nido*-carborane with triarylphosphines and diarylalkylphosphines in the presence of PdCl₂.

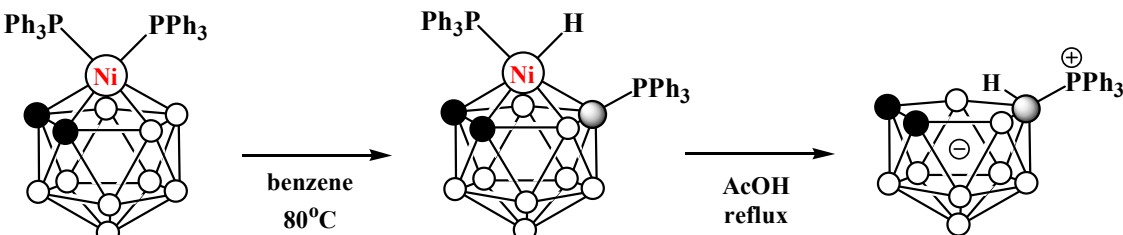

**R = Me, *n*-Bu**

**Scheme 42.** Reactions of *C*-isoquinolin-1-yl derivatives of *nido*-carborane with triphenylphosphine in the presence of PdCl₂.

It is also worth mentioning the formation of a symmetrically substituted triphenylphosphonium derivative of *nido*-carborane during the migration of the phosphine ligand from the metal atom to the dicarbollide ligand in nickelacarborane 3,3-(Ph₃P)₂-3,1,2-NiC₂B₉H₁₁ (Scheme 43) [115,116].

**Scheme 43.** Formation of 10-Ph₃P-7,8-C₂B₉H₁₁ from nickelacarborane 3,3-(Ph₃P)₂-3,1,2-NiC₂B₉H₁₁.

## 4. Charge-Compensated Derivatives of *Nido*-Carborane with Boron–Arsenic and Boron–Antimony Bonds

The charge-compensated derivatives of *nido*-carborane with boron–arsenic and boron–antimony bonds are rare and are limited to a few examples. Similar to the triphenylphosphonium derivative, the asymmetrically substituted triphenylarsonium and tetraphenyl-

stilbonium derivatives 9-Ph$_3$X-7,8-Ph$_2$-7,8-C$_2$B$_9$H$_9$ (X = As, Sb) were prepared via electrocatalyzed oxidative couplings of 7,8-diphenyl-*nido*-carborane with Ph$_3$As and Ph$_3$Sb, respectively (Scheme 44, Figure 36) [106].

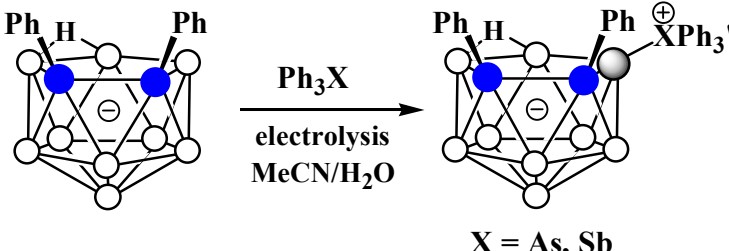

**Scheme 44.** Synthesis of 9-Ph$_3$As-7,8-Ph$_2$-7,8-C$_2$B$_9$H$_9$ and 9-Ph$_3$As-7,8-Ph$_2$-7,8-C$_2$B$_9$H$_9$.

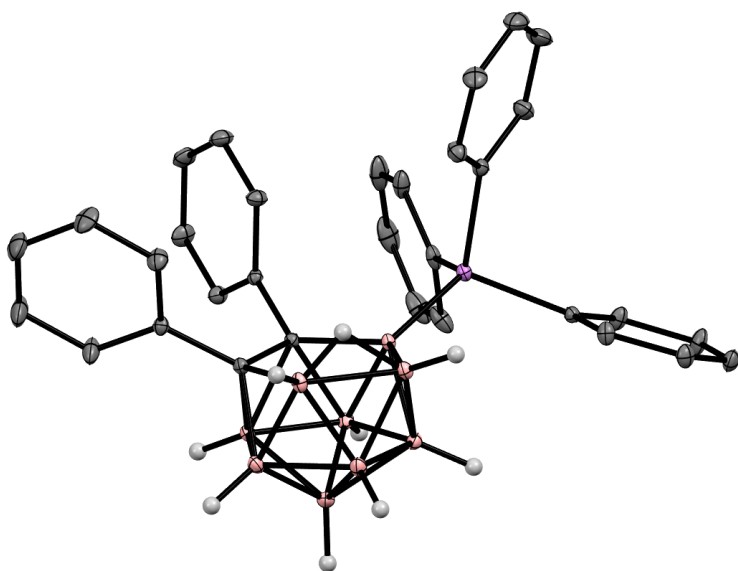

**Figure 36.** Crystal molecular structure of 9-Ph$_3$As-7,8-Ph$_2$-7,8-C$_2$B$_9$H$_9$. Hydrogen atoms of phenyl groups are omitted for clarity.

The reaction of 2-pyridyl-substituted *nido*-carborane [7-(2′-Py)-7,8-C$_2$B$_9$H$_{11}$]$^-$ with triphenylarsine in the presence of catalytic amounts of PdCl$_2$ in a mixture of toluene, water, and acetonitrile at 120°C results in the corresponding triphenylarsonium derivative 11-Ph$_3$As-7-(2′-Py)-7,8-C$_2$B$_9$H$_{10}$ (Scheme 45) [113].

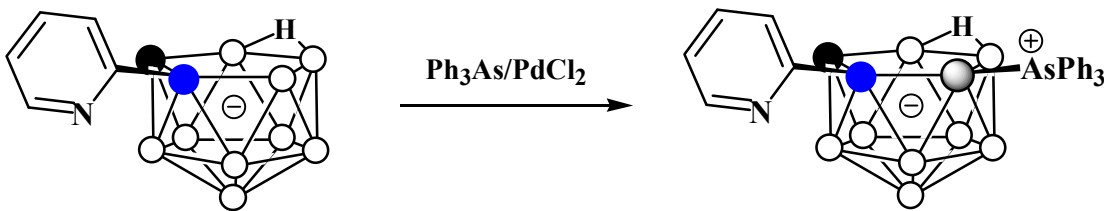

**Scheme 45.** Synthesis of 11-Ph$_3$As-7-(2′-Py)-7,8-C$_2$B$_9$H$_{10}$.

## 5. Charge-Compensated Derivatives of *Nido*-Carborane with Boron–Oxygen Bond

Alkyloxonium salts are much less stable than ammonium and phosphonium salts, and some of them are used in organic chemistry as strong alkylating agents. Nevertheless, strong electron-withdrawing of polyhedral boron hydride clusters substituted at boron atoms and, in particular, *nido*-carborane [117], is capable of stabilizing their oxonium derivatives [68,118]. The first example of such a derivative was obtained very soon after

the discovery of *nido*-carborane via the reaction of the potassium salt of *nido*-carborane with tetrahydrofuran in the presence of FeCl$_3$ in benzene. As a result, a mixture of two isomeric tetrahydrofuran derivatives of *nido*-carborane was obtained. The reaction with the *C,C′*-dimethyl derivative of *nido*-carborane proceeds in a similar way (Scheme 46) [73].

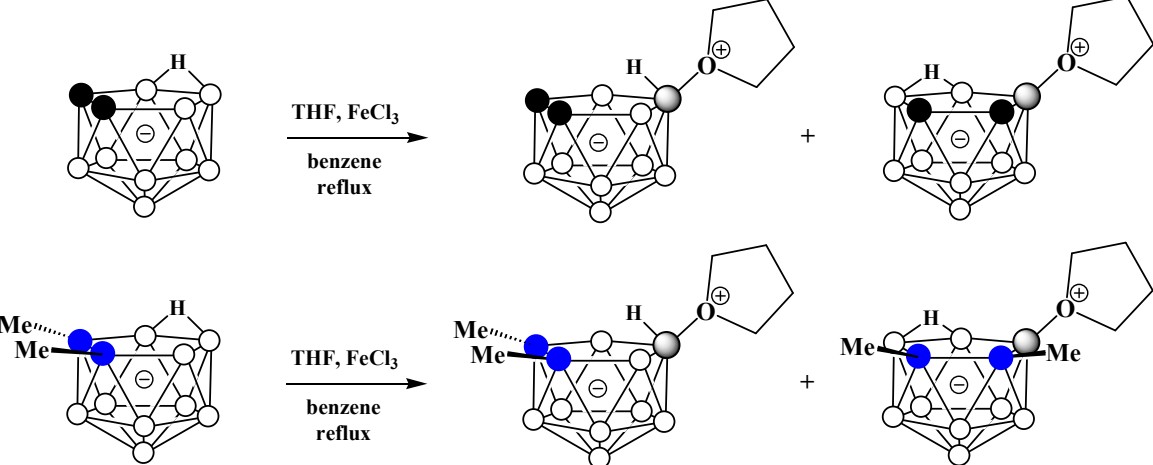

**Scheme 46.** Synthesis of 9-(CH$_2$)$_4$O-7,8-C$_2$B$_9$H$_{11}$ and 10-(CH$_2$)$_4$O-7,8-C$_2$B$_9$H$_{11}$ via the reaction of *nido*-carborane with FeCl$_3$ in THF–benzene mixture.

It was later found that the replacement of FeCl$_3$ with HgCl$_2$ in this reaction leads to the selective formation of the symmetrically substituted tetrahydrofuran derivative 10-(CH$_2$)$_4$O-7,8-C$_2$B$_9$H$_{11}$ [77,119]. The symmetrically substituted derivative was also obtained via the reaction of the tetramethylammonium salt of *nido*-carborane with AlCl$_3$ in a mixture of tetrahydrofuran and acetone [78] and by the treatment of the potassium salt of *nido*-carborane with tetrahydrofuran in the presence of acetaldehyde or formaldehyde and hydrochloric acid in a mixture of water and toluene [120] (Scheme 47).

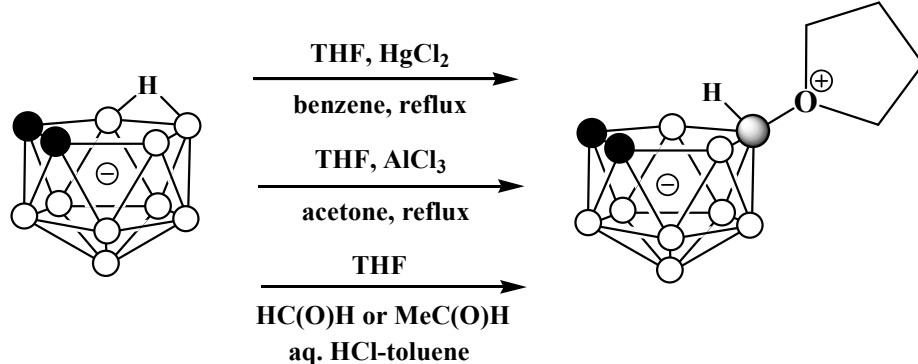

**Scheme 47.** Different pathways for synthesis of 10-(CH$_2$)$_4$O-7,8-C$_2$B$_9$H$_{11}$.

Oxonium derivatives of *nido*-carborane with other cyclic ethers were synthesized as well. The symmetrically substituted 1,4-dioxane derivative 10-O(CH$_2$CH$_2$)$_2$O-7,8-C$_2$B$_9$H$_{11}$ can be prepared via the reaction of the potassium salt of *nido*-carborane with 1,4-dioxane in the presence of HgCl$_2$ in benzene [119] or in the presence of acetaldehyde and hydrochloric acid in a water–toluene mixture [120]. The 1,4-dioxane derivative can also be synthesized via the heating of the protonated form of *nido*-carborane 7,8-C$_2$B$_9$H$_{13}$ with 1,4-dioxane [121] (Scheme 48). The molecular structure of the 1,4-dioxane derivative of *nido*-carborane was determined using single-crystal X-ray diffraction (Figure 37) [122].

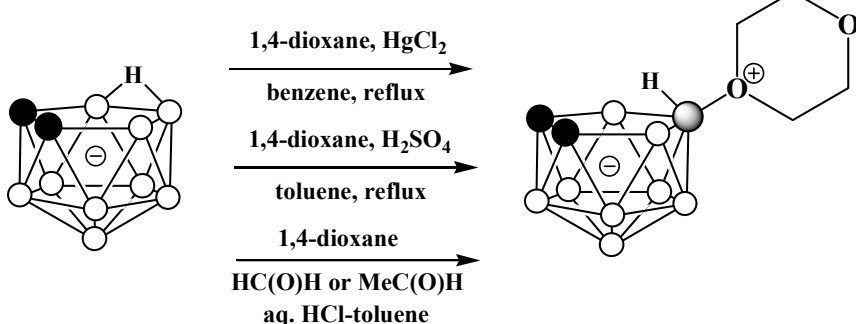

**Scheme 48.** Different pathways for synthesis of 10-O(CH$_2$CH$_2$)$_2$O-7,8-C$_2$B$_9$H$_{11}$.

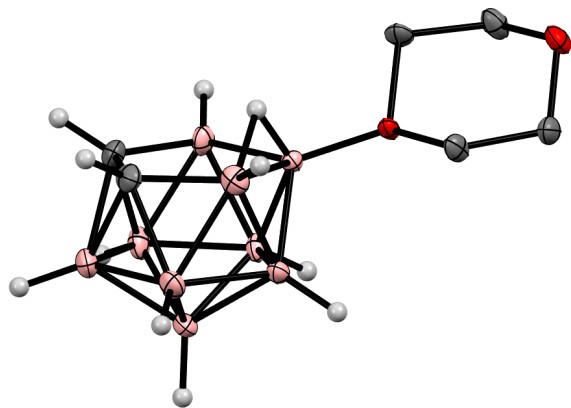

**Figure 37.** Crystal molecular structure of 10-O(CH$_2$CH$_2$)$_2$O-7,8-C$_2$B$_9$H$_{11}$. Hydrogen atoms of organic substituent are omitted for clarity.

The reaction of the potassium salt of *nido*-carborane with tetrahydropyran in the presence of mercury(II) chloride in benzene results in the tetrahydropyran derivative 10-(CH$_2$)$_5$O-7,8-C$_2$B$_9$H$_{11}$ (Scheme 49) [123,124].

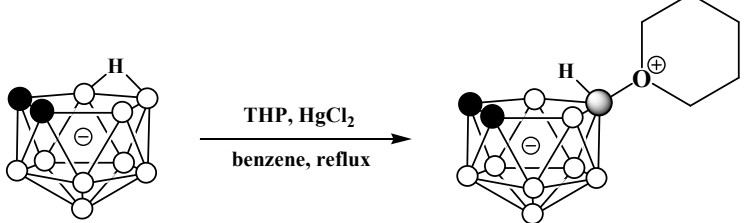

**Scheme 49.** Different pathways for synthesis of 10-(CH$_2$)$_5$O-7,8-C$_2$B$_9$H$_{11}$.

The most important property of the cyclic oxonium derivatives of *nido*-carborane is their tendency for ring-opening reactions under the action of nucleophiles. This makes it possible to modify the *nido*-carborane cluster and introduce various terminal groups including functional groups as side substituents. At the same time, depending on the oxonium derivative used, it is possible to obtain compounds with different spacer lengths between the terminal group and the cluster (Scheme 50). In this way, *nido*-carborane-based carboxylic acids [119,125], azides [119,126,127], calixarenes [122], and coumarins [128], as well as hydroxy [122,129,130], halogen [122], ammonium [121,122,124], and mercapto [122,131] derivatives, and others, were prepared. The use of two equivalents of oxonium derivatives in the reaction with dinucleophiles allowed us to obtain podands, which were used for the synthesis of crown ethers [129,131] (Scheme 50).

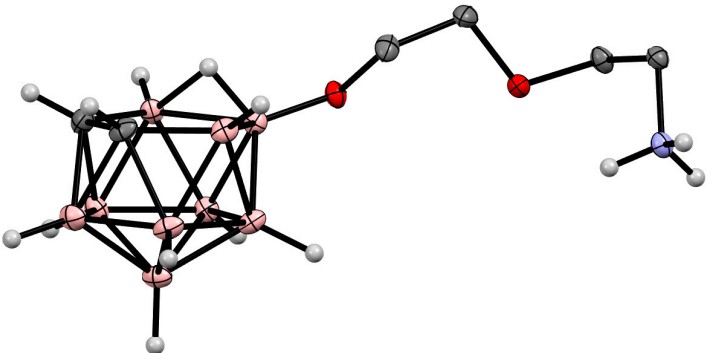

**Scheme 50.** The ring-opening reactions of the cyclic oxonium derivatives of *nido*-carborane under action of nucleophiles.

It should be noted that the use of neutral nucleophiles, such as ammonia, in ring-opening reactions leads to charge-compensated derivatives in which the charges are separated by a spacer formed during the opening of the oxonium ring (Figure 38) [122].

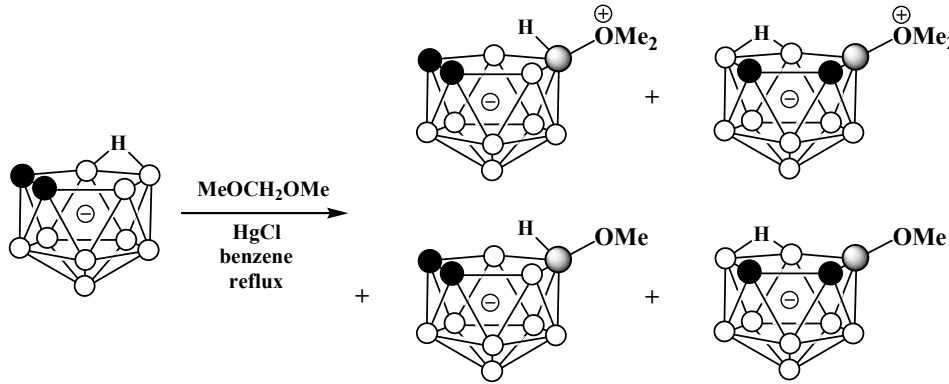

**Figure 38.** Crystal molecular structure of $10\text{-}H_3N(CH_2CH_2O)_2\text{-}7,8\text{-}C_2B_9H_{11}$. Hydrogen atoms of methylene groups are omitted for clarity.

There are several examples of acyclic oxonium derivatives of *nido*-carborane. The reaction of the potassium salt of *nido*-carborane with dimethoxymethane in the presence of mercury(II) chloride in a benzene solution leads to a mixture of the asymmetrically and symmetrically substituted dimethyloxonium derivatives $9\text{-}Me_2O\text{-}7,8\text{-}C_2B_9H_{11}$ and $10\text{-}Me_2O\text{-}7,8\text{-}C_2B_9H_{11}$, along with the corresponding methoxy derivatives $[9\text{-}MeO\text{-}7,8\text{-}C_2B_9H_{11}]^-$ and $[10\text{-}MeO\text{-}7,8\text{-}C_2B_9H_{11}]^-$ [132] (Scheme 51).

**Scheme 51.** Synthesis of the dimethyloxonium derivatives of *nido*-carborane.

The symmetrically substituted diethyloxonium derivative of *nido*-carborane can be obtained via the reaction of the potassium salt of *nido*-carborane with diethyl ether in the presence of formaldehyde or acetaldehyde and hydrochloric acid in a mixture of water and toluene [120], or via the reaction of the potassium salt of *nido*-carborane with diethyl ether in the presence of $HgCl_2$ in benzene [50] (Scheme 52).

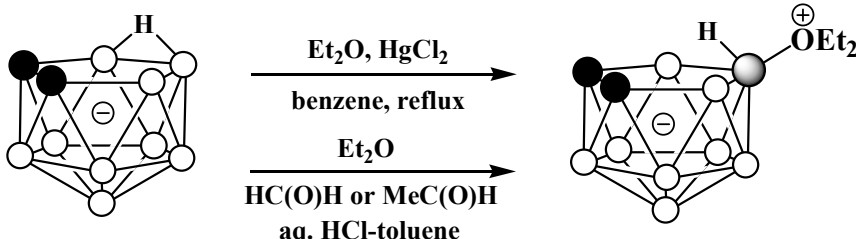

**Scheme 52.** Different pathways for synthesis of $10\text{-Et}_2\text{O-}7,8\text{-C}_2\text{B}_9\text{H}_{11}$.

The symmetrically and asymmetrically substituted diethyloxonium derivatives of *nido*-carborane $9\text{-Et}_2\text{O-}7,8\text{-C}_2\text{B}_9\text{H}_{11}$ and $10\text{-Et}_2\text{O-}7,8\text{-C}_2\text{B}_9\text{H}_{11}$ along with the 9-diethyloxonium-11-chloro derivative $[9\text{-Et}_2\text{O-}11\text{-Cl-}7,8\text{-C}_2\text{B}_9\text{H}_{10}]$ were found to form as by-products in the reaction of the dicarbollide dianion with $\text{PhBCl}_2$ in diethyl ether [133]. The structure of the 10-diethyloxonium derivative $10\text{-Et}_2\text{O-}7,8\text{-C}_2\text{B}_9\text{H}_{11}$ was determined using single-crystal X-ray diffraction (Figure 39) [133].

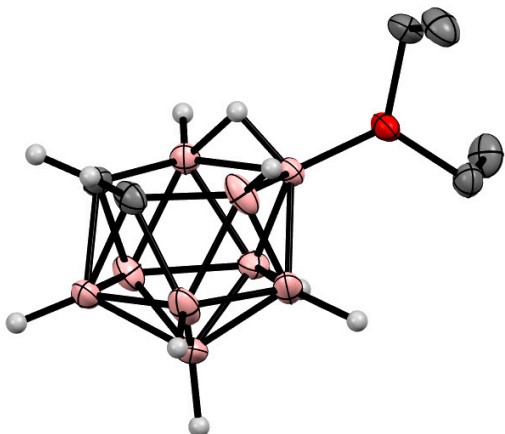

**Figure 39.** Crystal molecular structure of $10\text{-Et}_2\text{O-}7,8\text{-C}_2\text{B}_9\text{H}_{11}$. Hydrogen atoms of organic substituent are omitted for clarity.

The dialkyloxonium derivatives of *nido*-carboranes easily lose the alkyl group when they react with nucleophiles and therefore can be considered alkylating agents [132,133].

An interesting example of a charge-compensated derivative of *nido*-carborane is $10\text{-Me}_2\text{SO-}7,8\text{-}\mu\text{-(CH}_2)_3\text{-}7,8\text{-C}_2\text{B}_9\text{H}_9$, which can be obtained via the reaction of the potassium salt of $[7,8\text{-}\mu\text{-(CH}_2)_3\text{-}7,8\text{-C}_2\text{B}_9\text{H}_{10}]^-$ with a dimethyl sulfoxide/water solution in the presence of concentrated $\text{H}_2\text{SO}_4$, or via the reaction of the trimethylammonium salt with DMSO in dry 1,2-dichloroethane in the presence of triflic acid [134] (Scheme 53). The structure of $10\text{-Me}_2\text{SO-}7,8\text{-}\mu\text{-(CH}_2)_3\text{-}7,8\text{-C}_2\text{B}_9\text{H}_9$ was determined using single-crystal X-ray diffraction (Figure 40) [134].

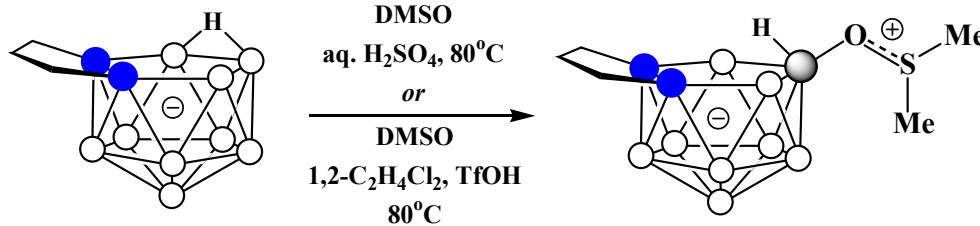

**Scheme 53.** Synthesis of $10\text{-Me}_2\text{SO-}7,8\text{-}\mu\text{-(CH}_2)_3\text{-}7,8\text{-C}_2\text{B}_9\text{H}_9$.

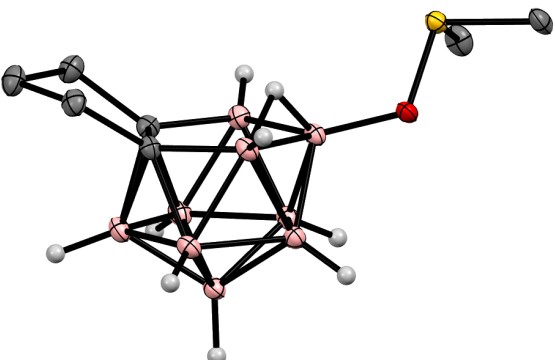

**Figure 40.** Crystal molecular structure of 10-Me$_2$SO-7,8-μ-(CH$_2$)$_3$-7,8-C$_2$B$_9$H$_9$. Hydrogen atoms of organic substituent are omitted for clarity.

## 6. Charge-Compensated Derivatives of *Nido*-Carborane with Boron–Sulfur Bond

Compared with the oxonium derivatives, the sulfonium derivatives of *nido*-carborane are represented by a wider variety of derivatives and synthetic methods for their preparation. However, the most studied of them are the dimethylsulfonium derivatives of *nido*-carborane, which are widely used for the synthesis of metallacarboranes [41,50,135].

It should be noted that symmetrically and asymmetrically substituted dimethylsulfonium derivatives of *nido*-carborane are usually obtained in different ways, which excludes the formation of mixtures of their isomers. The asymmetrically substituted 9-dimethylsulfonium derivative of *nido*-carborane 9-Me$_2$S-7,8-C$_2$B$_9$H$_{11}$ was prepared via the reaction of the parent *nido*-carborane with dimethylsulfoxide in the presence of sulfuric acid at 80°C [74,136,137]. The reactions of the *C,C′*-substituted derivatives of *nido*-carborane proceed in a similar way, leading to the corresponding dimethylsulfonium derivatives 9-Me$_2$S-7,8-R$_2$-7,8-C$_2$B$_9$H$_9$ (R = Me, Ph) [74,138] (Scheme 54). These conditions are similar to those used for the synthesis of the dimethylsulfonium derivatives of the *closo*-decaborate [139] and *closo*-dodecaborate [140] anions. The *C,C′*-substituted derivatives 9-Me$_2$S-7,8-Me$_2$-7,8-C$_2$B$_9$H$_9$ and 9-Me$_2$S-7,8-μ-(1′,2′-C$_6$H$_4$(CH$_2$)$_2$)-7,8-C$_2$B$_9$H$_9$ were prepared via the reactions of the corresponding *nido*-carboranes with dimethylsulfoxide in the presence of triflic acid in 1,2-dichloroethane [134] (Scheme 54). The structures of 9-Me$_2$S-7,8-R$_2$-7,8-C$_2$B$_9$H$_9$ (R = H, Ph) were determined using single-crystal X-ray diffraction (Figure 41) [141,142].

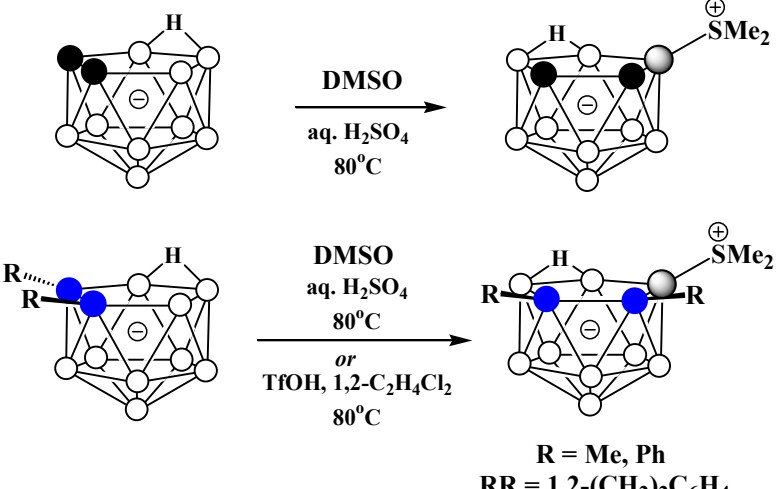

R = Me, Ph
RR = 1,2-(CH$_2$)$_2$C$_6$H$_4$

**Scheme 54.** Synthesis of the asymmetrically substituted dimethylsulfonium derivatives of *nido*-carborane 9-Me$_2$S-7,8-R$_2$-7,8-C$_2$B$_9$H$_9$ (R = H, Me, Ph; RR = μ-1′,2′-C$_6$H$_4$(CH$_2$)$_2$).

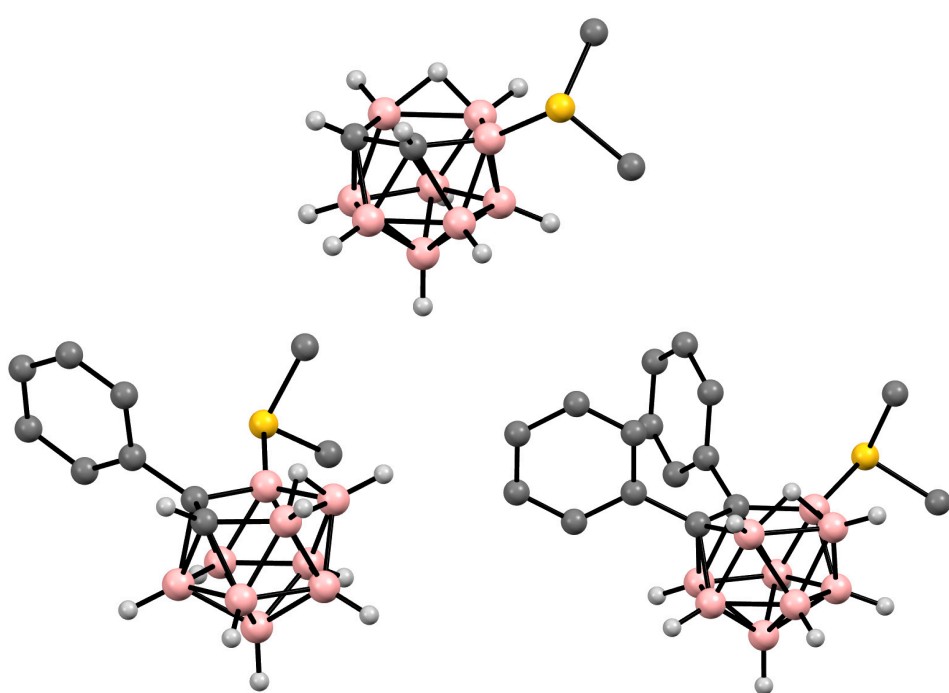

**Figure 41.** Crystal molecular structures of $9\text{-}Me_2S\text{-}7,8\text{-}C_2B_9H_{11}$ (**top**), $11\text{-}Me_2S\text{-}7\text{-}Ph\text{-}7,8\text{-}R_2\text{-}7,8\text{-}C_2B_9H_{10}$ (**bottom left**), and $9\text{-}Me_2S\text{-}7,8\text{-}Ph_2\text{-}7,8\text{-}C_2B_9H_9$ (**bottom right**). Hydrogen atoms of organic substituent are omitted for clarity.

The asymmetrically substituted dimethylsulfonium derivatives $9\text{-}Me_2S\text{-}7,8\text{-}Me_2\text{-}7,8\text{-}C_2B_9H_9$ and $9\text{-}Me_2S\text{-}7,8\text{-}\mu\text{-}(CH_2OCH_2)\text{-}7,8\text{-}C_2B_9H_9$ were prepared via the reactions of the corresponding *nido*-carboranes with dimethylsulfide in the presence of $Fe(NO_3)_3$ in aqueous ethanol [49].

In the case of *C*-substituted *nido*-carboranes, such as $K[7\text{-}Ph\text{-}7,8\text{-}C_2B_9H_{11}]$, the introduction of a $Me_2S$ group results in a mixture of $9\text{-}Me_2S\text{-}7\text{-}Ph\text{-}7,8\text{-}C_2B_9H_{10}$ and $11\text{-}Me_2S\text{-}7\text{-}Ph\text{-}7,8\text{-}C_2B_9H_{10}$ isomers, which can be separated using column chromatography [137]. The molecular structure of $11\text{-}Me_2S\text{-}7\text{-}Ph\text{-}7,8\text{-}C_2B_9H_{10}$ was determined using single-crystal X-ray diffraction (Figure 41) [138].

In a similar way, the reactions of the cesium salts of the 5-methyl or 5-bromo derivatives of *nido*-carborane $K[5\text{-}R\text{-}7,8\text{-}C_2B_9H_{11}]$ (R = Me or Br) with dimethylsulfide in the presence of iron(III) chloride $FeCl_3$ in aqueous ethanol result in mixtures of the $9\text{-}Me_2S\text{-}5\text{-}R\text{-}7,8\text{-}C_2B_9H_{10}$ and $11\text{-}Me_2S\text{-}5\text{-}R\text{-}7,8\text{-}C_2B_9H_{10}$ isomers, which were separated using column chromatography on silica (Scheme 55) [143].

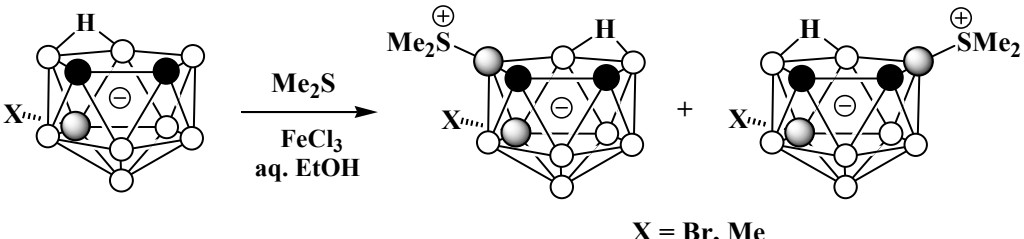

**Scheme 55.** Reaction of the 5-substituted derivatives of *nido*-carborane $5\text{-}X\text{-}7,8\text{-}C_2B_9H_{11}$ (X = Br, Me) with dimethylsulfide in the presence of aq. $FeCl_3$.

The reaction of the tetramethylammonium salt of 9-methyl-*nido*-carborane $(Me_4N)[9\text{-}Me\text{-}7,8\text{-}C_2B_9H_{11}]$ with dimethyl sulfide under the same conditions results in the introduction of a $Me_2S$ group into position nine of the *nido*-carborane cage. However, the reaction is accompanied by a rearrangement of the carborane cage, leading to a mixture of the $9\text{-}Me_2S\text{-}$

3-Me-7,8-C$_2$B$_9$H$_{10}$ (main isomer), 9-Me$_2$S-4-Me-7,8-C$_2$B$_9$H$_{10}$, 9-Me$_2$S-2-Me-7,8-C$_2$B$_9$H$_{10}$, 9-Me$_2$S-1-Me-7,8-C$_2$B$_9$H$_{10}$, and 9-Me$_2$S-10-Me-7,8-C$_2$B$_9$H$_{10}$ isomers, which were separated using column chromatography (Scheme 56) [143]. It was assumed that the reaction proceeds through the oxidative closure of the *nido*-carborane cage followed by a series of rearrangements of the resulting 11-vertex *B*-substituted *closo*-carborane with a subsequent reopening of its isomers under the action of dimethylsulfide. The structures of 9-Me$_2$S-3-Me-7,8-C$_2$B$_9$H$_{10}$ and 9-Me$_2$S-4-Me-7,8-C$_2$B$_9$H$_{10}$ were determined using single-crystal X-ray diffraction (Figure 42) [143].

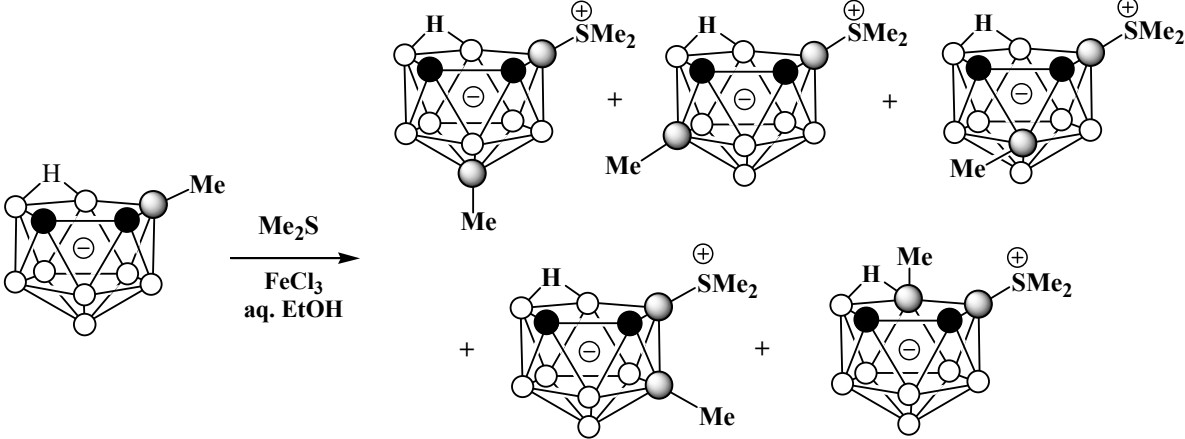

**Scheme 56.** Reaction of the 9-methyl derivative of *nido*-carborane 9-Me-7,8-C$_2$B$_9$H$_{11}$ with dimethylsulfide in the presence of aqueous iron(III) chloride.

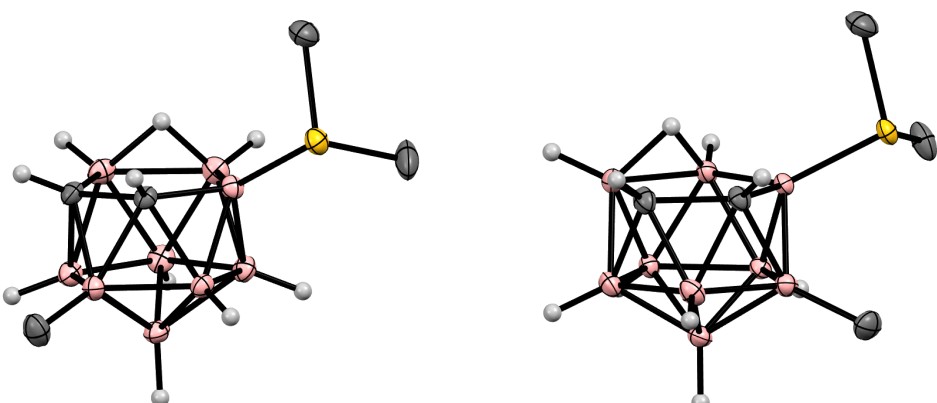

**Figure 42.** Crystal molecular structures of 9-Me$_2$S-3-Me-7,8-C$_2$B$_9$H$_{10}$ (**left**) and 9-Me$_2$S-4-Me-7,8-C$_2$B$_9$H$_{10}$ (**right**). Hydrogen atoms of methyl groups are omitted for clarity.

The 9-dimethylsulfonium derivative of *nido*-carborane can be demethylated with strong bases (sodium naphthalenide in tetrahydrofuran at room temperature [144,145]; 1,1′-bis(diphenylphosphino)ferrocene (dppf) in toluene at 80 °C [146]; sodium in liquid ammonia at −40 °C [145]; and boiling TMEDA, morpholine or triethylamine [147], and sodium amide in boiling toluene [147]) to the 9-methylthio derivative [9-MeS-7,8-C$_2$B$_9$H$_{11}$]$^-$, the subsequent alkylation of which gives a whole series of new dialkylsulfonium derivatives of *nido*-carborane 9-R(Me)S-7,8-C$_2$B$_9$H$_{11}$, including derivatives with various functional groups. The resulting *nido*-carboranyl esters, nitriles, and phthalimides can be converted into corresponding carboxylic acids and amines using acid hydrolysis and deprotection with hydrazine, respectively (Scheme 57, Figure 43) [147–149].

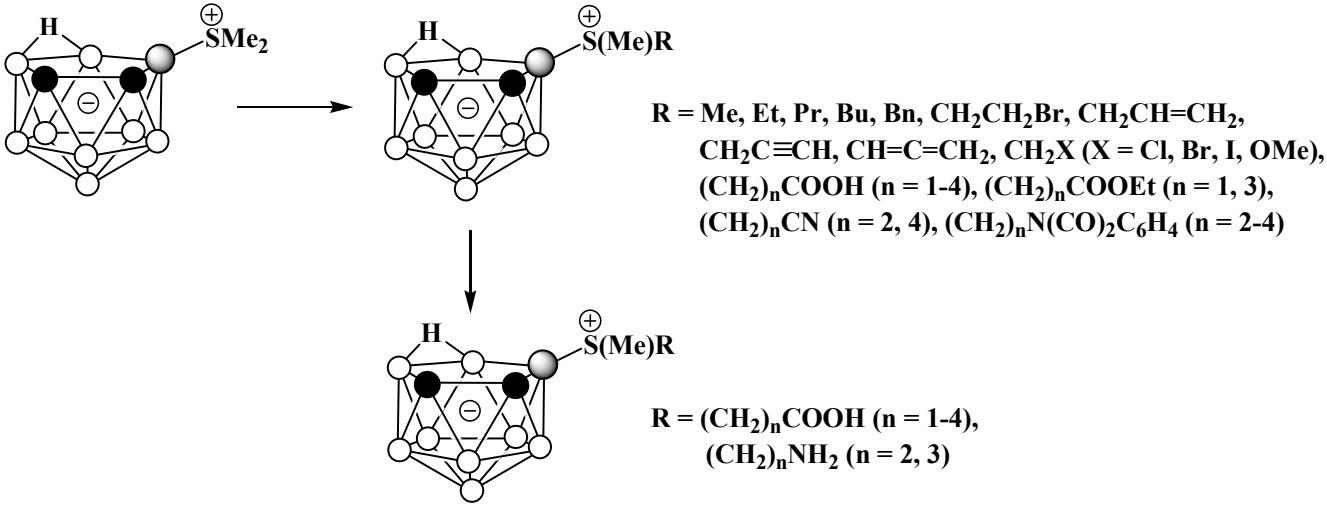

$R = Me, Et, Pr, Bu, Bn, CH_2CH_2Br, CH_2CH=CH_2,$
$CH_2C{\equiv}CH, CH=C=CH_2, CH_2X (X = Cl, Br, I, OMe),$
$(CH_2)_nCOOH (n = 1-4), (CH_2)_nCOOEt (n = 1, 3),$
$(CH_2)_nCN (n = 2, 4), (CH_2)_nN(CO)_2C_6H_4 (n = 2-4)$

$R = (CH_2)_nCOOH (n = 1-4),$
$(CH_2)_nNH_2 (n = 2, 3)$

**Scheme 57.** The synthetic pathway to obtain $9\text{-R(Me)S-7,8-}C_2B_9H_{11}$ derivatives.

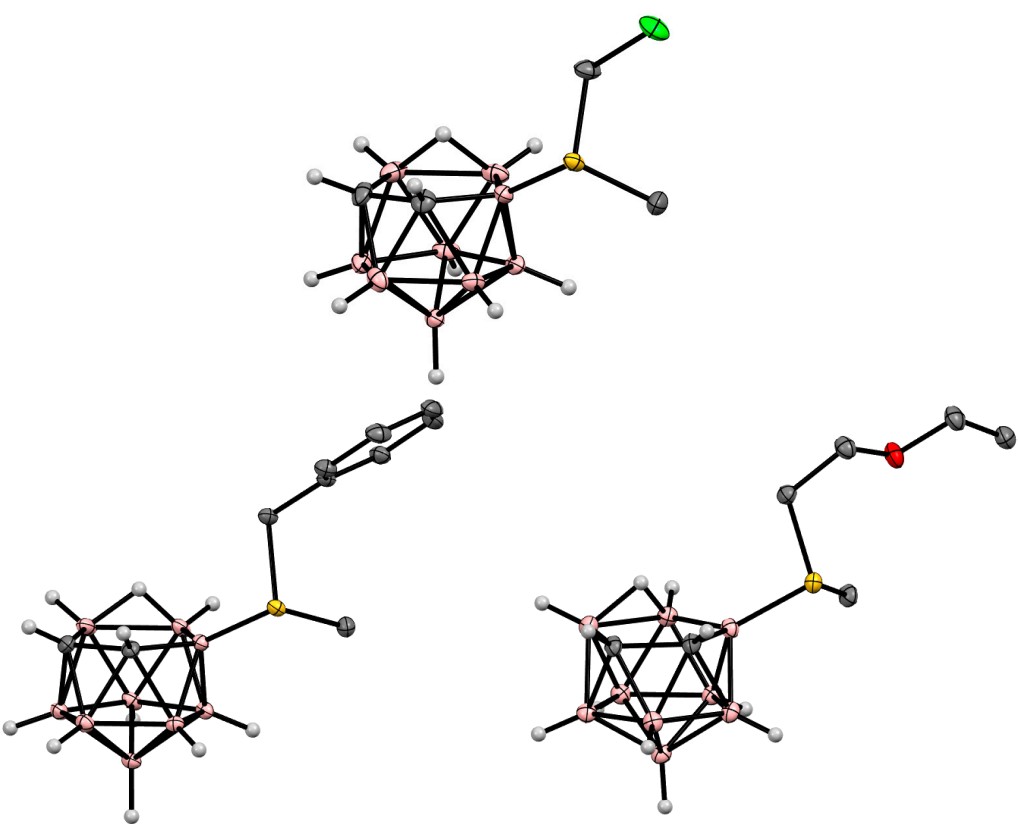

**Figure 43.** Crystal molecular structures of $9\text{-ClCH}_2(Me)S\text{-7,8-}C_2B_9H_{11}$ (**top**), $9\text{-Bn(Me)S-7,8-}C_2B_9H_{11}$ (**bottom left**), and $9\text{-EtOCH}_2CH_2(Me)S\text{-7,8-}C_2B_9H_{11}$ (**bottom right**). Hydrogen atoms of organic substituents are omitted for clarity.

The reaction of the $2'$-bromoethyl(methyl)sulfonium derivative $9\text{-BrCH}_2CH_2(Me)S\text{-}$ $7,8\text{-}C_2B_9H_{11}$ with $K_2CO_3$ in ethanol leads to the ethoxy derivative $9\text{-EtOCH}_2CH_2(Me)S\text{-}$ $7,8\text{-}C_2B_9H_{11}$, while the reaction in chloroform results in the vinylsulfonium derivative $9\text{-CH}_2=CH(Me)S\text{-7,8-}C_2B_9H_{11}$ (Scheme 58, Figure 43) [148].

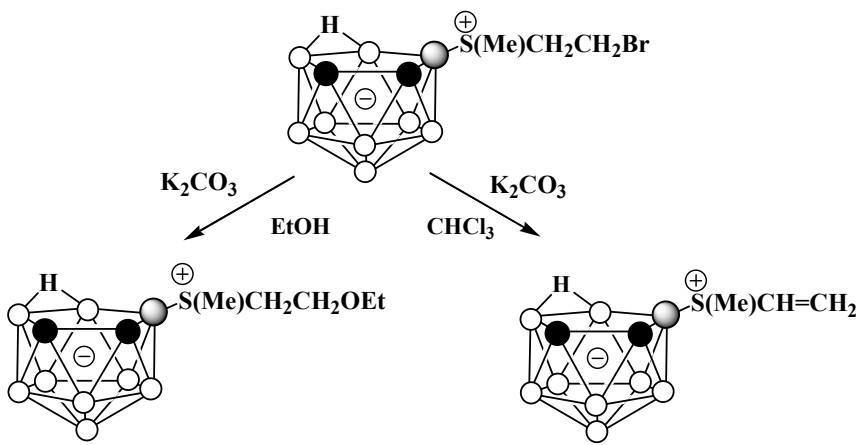

**Scheme 58.** Reaction of 9-BrCH$_2$CH$_2$(Me)S-7,8-C$_2$B$_9$H$_{11}$ with K$_2$CO$_3$.

The asymmetrically substituted diethylsulfonium derivative 9-Et$_2$S-7,8-C$_2$B$_9$H$_{11}$ was prepared via electrocatalyzed B-S oxidative couplings of the tetramethylammonium salt of *nido*-carborane with diethylsulfide (Scheme 59). The reaction is applicable to various *C,C'*-dialkyl and diaryl derivatives of *nido*-carborane (Scheme 59, Figure 44) [106]. This approach can be used for the synthesis of carboranyl sulfonium derivatives containing various alkyl and aryl groups (Scheme 60, Figure 44) [106].

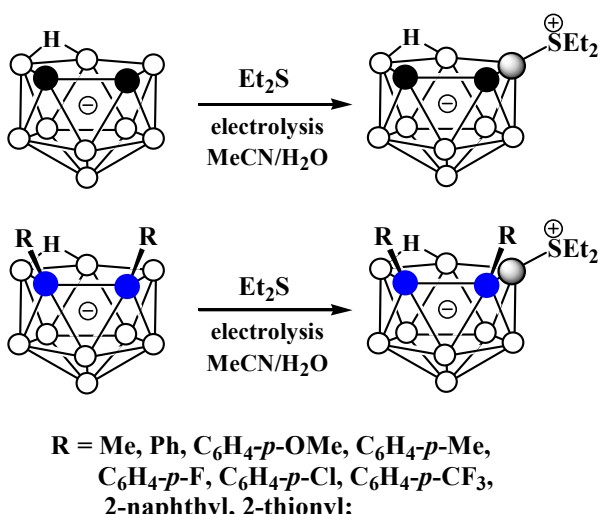

**R = Me, Ph, C$_6$H$_4$-*p*-OMe, C$_6$H$_4$-*p*-Me,**
**C$_6$H$_4$-*p*-F, C$_6$H$_4$-*p*-Cl, C$_6$H$_4$-*p*-CF$_3$,**
**2-naphthyl, 2-thionyl;**
**RR = CH$_2$CH$_2$CH$_2$, CH$_2$CH=CHCH$_2$**

**Scheme 59.** The electrocatalyzed B-S oxidative coupling of *nido*-carboranes with diethylsulfide.

Halogenation of the 9-dimethylsulfonium derivative 9-Me$_2$S-7,8-C$_2$B$_9$H$_{11}$ was studied. The reaction with an equimolar amount of *N*-chlorosuccinimide in acetonitrile produces 11-Cl-9-Me$_2$S-7,8-C$_2$B$_9$H$_{10}$ [150], while bubbling gaseous Cl$_2$ through a solution of 9-Me$_2$S-7,8-C$_2$B$_9$H$_{11}$ in dichloromethane results in the dichloro derivative 6,11-Cl$_2$-9-Me$_2$S-7,8-C$_2$B$_9$H$_9$ (Figure 45) [151,152]. The reaction of 9-Me$_2$S-7,8-C$_2$B$_9$H$_{11}$ with an equimolar amount of Br$_2$ in dichloromethane results in a mixture of 11-Br-9-Me$_2$S-7,8-C$_2$B$_9$H$_{10}$ (Figure 45) and 6-Br-9-Me$_2$S-7,8-C$_2$B$_9$H$_{10}$ isolated in 61% and 18% yields, respectively, while the reaction with an excess of Br$_2$ gives 6,11-Br$_2$-9-Me$_2$S-7,8-C$_2$B$_9$H$_9$ (Figure 45) in an 88% yield [150]. The reaction of 9-Me$_2$S-7,8-C$_2$B$_9$H$_{11}$ with I$_2$ in acetic acid under reflux leads to 11-I-9-Me$_2$S-7,8-C$_2$B$_9$H$_{10}$ (Figure 45) as a single product isolated in a 37% yield [150]. The same product was prepared in a 78% yield via the reaction of the iodo derivative of *nido*-carborane [9-I-7,8-C$_2$B$_9$H$_{11}$]$^-$ with dimethylsulfoxide in the presence of concentrated sulfuric acid [150].

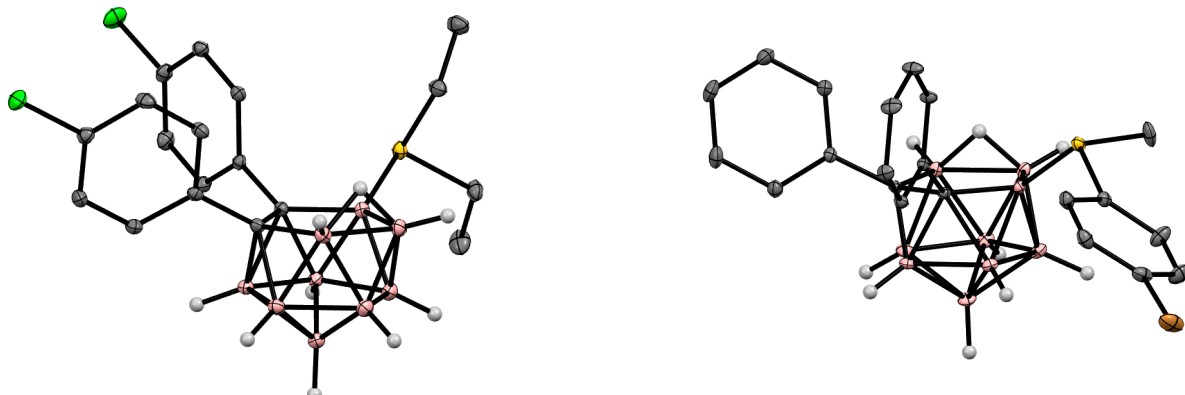

**Figure 44.** Crystal molecular structures of 9-Et$_2$S-7,8-(4'-ClC$_6$H$_4$)$_2$-7,8-C$_2$B$_9$H$_9$ (**left**) and 9-(4'-BrC$_6$H$_4$)EtS-7,8-Ph$_2$-7,8-C$_2$B$_9$H$_9$ (**right**). Hydrogen atoms of organic substituents are omitted for clarity.

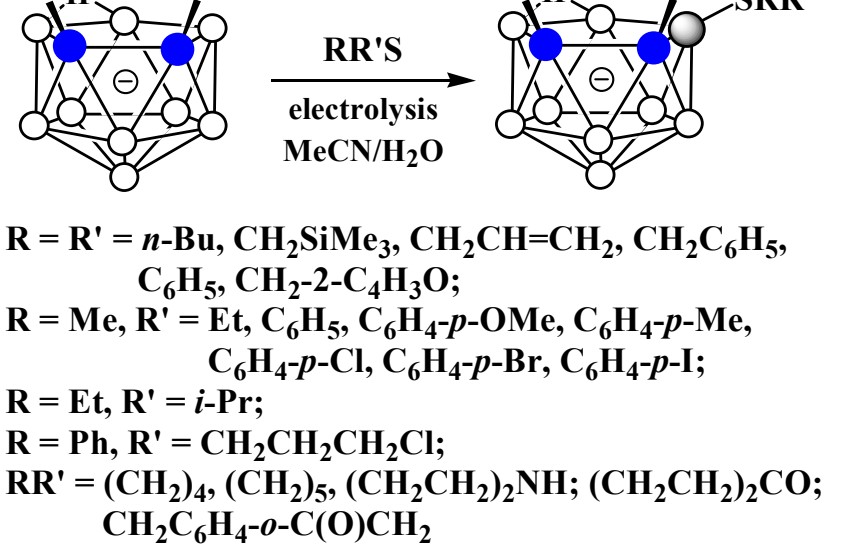

R = R' = *n*-Bu, CH$_2$SiMe$_3$, CH$_2$CH=CH$_2$, CH$_2$C$_6$H$_5$,
   C$_6$H$_5$, CH$_2$-2-C$_4$H$_3$O;
R = Me, R' = Et, C$_6$H$_5$, C$_6$H$_4$-*p*-OMe, C$_6$H$_4$-*p*-Me,
   C$_6$H$_4$-*p*-Cl, C$_6$H$_4$-*p*-Br, C$_6$H$_4$-*p*-I;
R = Et, R' = *i*-Pr;
R = Ph, R' = CH$_2$CH$_2$CH$_2$Cl;
RR' = (CH$_2$)$_4$, (CH$_2$)$_5$, (CH$_2$CH$_2$)$_2$NH; (CH$_2$CH$_2$)$_2$CO;
   CH$_2$C$_6$H$_4$-*o*-C(O)CH$_2$

**Scheme 60.** The electrocatalyzed direct B-S oxidative coupling of *nido*-carboranes with sulfides.

This approach was used for the synthesis of *nido*-carboranyl analogs of some drugs including ibuprofen, indomethacin, ciprofibrate, and probenecid [106].

The symmetrically substituted sulfonium derivatives of *nido*-carboranes 10-RR'S-7,8-C$_2$B$_9$H$_{11}$ were prepared via the reaction of the potassium salt of *nido*-carborane with various alkyl sulfides, HCl, and acetaldehyde in a mixture of water and toluene (Scheme 61, Figure 46) [120,153]. The symmetrically substituted sulfonium derivatives of *C*- and *C,C'*-substituted *nido*-carboranes were prepared in the same way (Scheme 61, Figure 46) [44,153,154].

When acetaldehyde is replaced with formaldehyde under similar conditions, a mixture of 9-R$_2$SCH$_2$-7,8-C$_2$B$_9$H$_{11}$ (as the main product) and 10-R$_2$S-7,8-C$_2$B$_9$H$_{11}$ is formed (Scheme 62) [120].

A convenient method for the synthesis of the 10-dimethylsulfonium derivative of *nido*-carborane is a two-step reaction of *nido*-carborane with dimethysulfide in toluene in the presence of a strong acid [155] (Scheme 63). However, the reaction of (Me$_3$NH)[3-Ph-7,8-C$_2$B$_9$H$_{12}$] with Me$_2$S under similar conditions leads to the sulfonium product 10-SMe$_2$-3-Ph-7,8-C$_2$B$_9$H$_{12}$ in only a 15% yield [50].

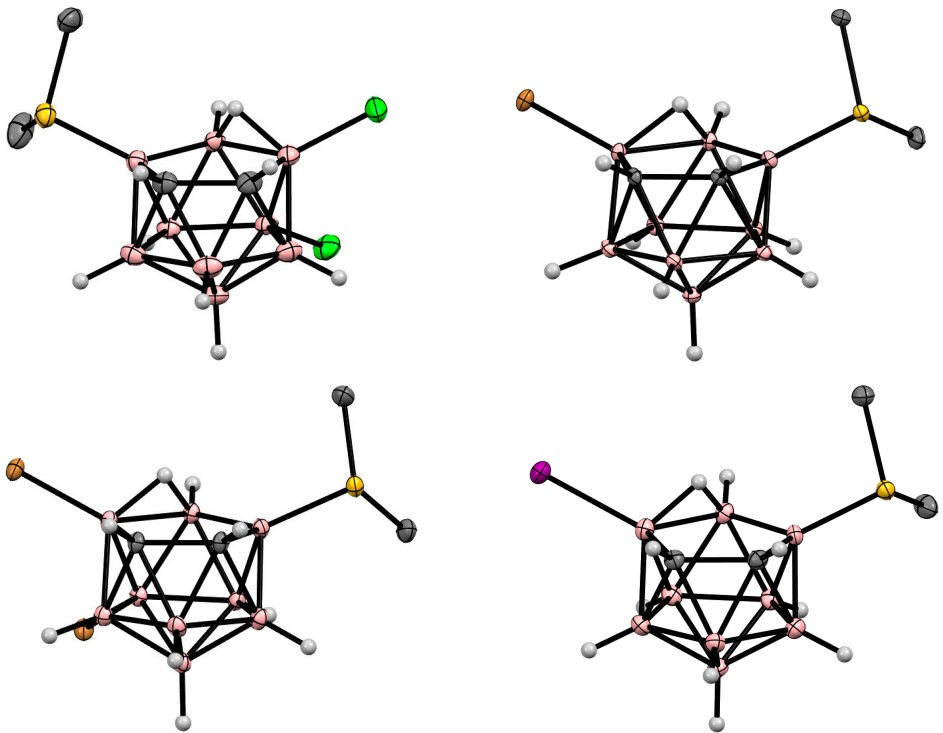

**Figure 45.** Crystal molecular structures of 6,11-Cl$_2$-9-Me$_2$S-7,8-C$_2$B$_9$H$_9$ (**top left**), 11-Br-9-Me$_2$S-7,8-C$_2$B$_9$H$_{10}$ (**top right**), 6,11-Br$_2$-9-Me$_2$S-7,8-C$_2$B$_9$H$_9$ (**bottom left**), and 11-I-9-Me$_2$S-7,8-C$_2$B$_9$H$_{10}$ (**bottom right**). Hydrogen atoms of methyl groups are omitted for clarity.

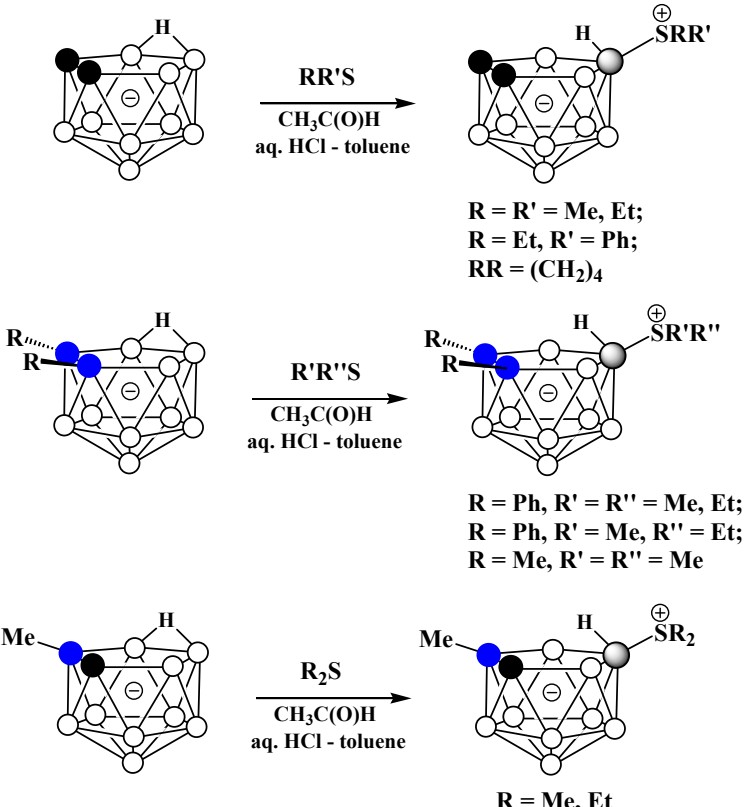

**Scheme 61.** Synthesis of symmetrically substituted sulfonium derivatives of *nido*-carborane.

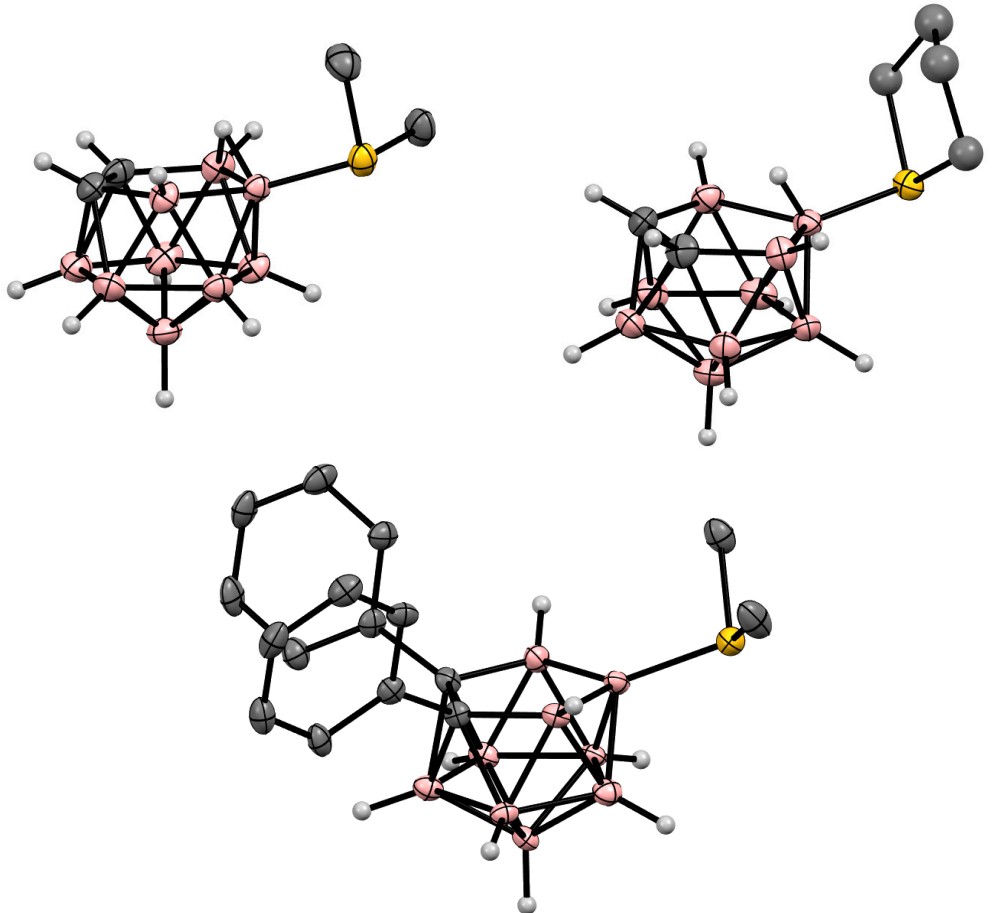

**Figure 46.** Crystal molecular structures of 10-Me$_2$S-7,8-C$_2$B$_9$H$_{11}$ (**top left**), 10-(CH$_2$)$_4$S-7,8-C$_2$B$_9$H$_{11}$ (**top right**), and 10-Me$_2$S-7,8-Ph$_2$-7,8-C$_2$B$_9$H$_9$ (**bottom**). Hydrogen atoms of organic substituents are omitted for clarity.

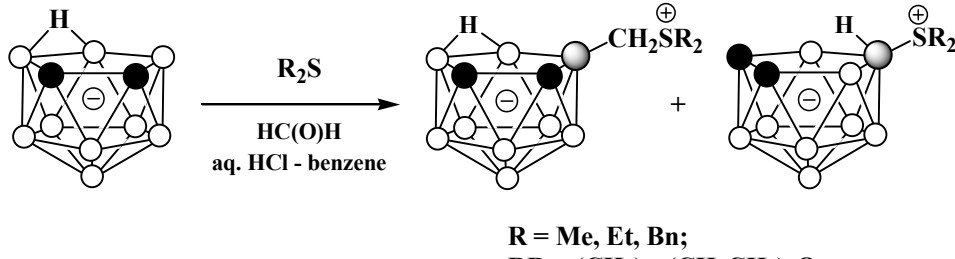

**R = Me, Et, Bn;**
**RR = (CH$_2$)$_4$, (CH$_2$CH$_2$)$_2$O**

**Scheme 62.** Synthesis of 9-R$_2$SCH$_2$-7,8-C$_2$B$_9$H$_{11}$.

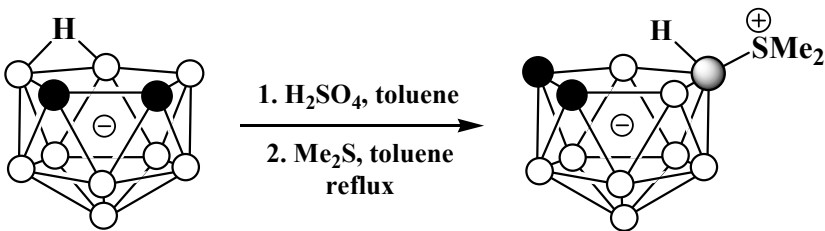

**Scheme 63.** Synthesis of 10-Me$_2$S-7,8-C$_2$B$_9$H$_{11}$.

In a similar way, the reaction of (Me$_4$N)[7,8-µ-(CH$_2$)$_3$-7,8-C$_2$B$_9$H$_{10}$] with triflic acid in a mixture of DMSO and 1,2-dichloroethane mixture results in the symmetrically substituted dimethylsulfonium derivative 10-Me$_2$S-7,8-µ-(CH$_2$)$_3$-7,8-C$_2$B$_9$H$_{10}$ (Scheme 64) [132].

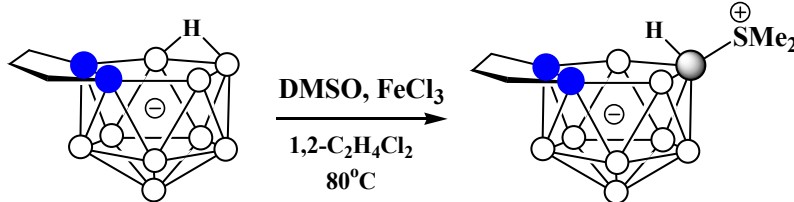

**Scheme 64.** Synthesis of 10-Me$_2$S-7,8-µ-(CH$_2$)$_3$-7,8-C$_2$B$_9$H$_9$.

Like the symmetrically substituted oxonium derivatives, the symmetrically substituted dialkylsulfonium derivatives of *nido*-carborane can be prepared by reacting the potassium or cesium salt of *nido*-carborane with alkyl sulfides in the presence of HgCl$_2$ (Scheme 65) [50].

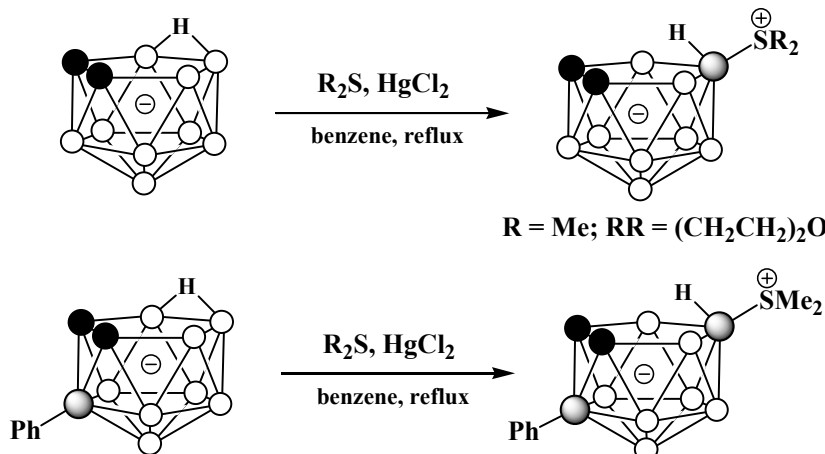

**Scheme 65.** Synthesis of symmetrically substituted dialkylsulfonium derivatives of *nido*-carborane in the presence of HgCl$_2$.

The symmetrically substituted sulfonium derivatives 10-Me$_2$S-7,8-R$_2$-7,8-C$_2$B$_9$H$_9$ were prepared via the reaction of the tetramethylammonium salts of the corresponding *nido*-carboranes with Me$_2$S in the presence of FeCl$_3$ in benzene [49] (Scheme 66).

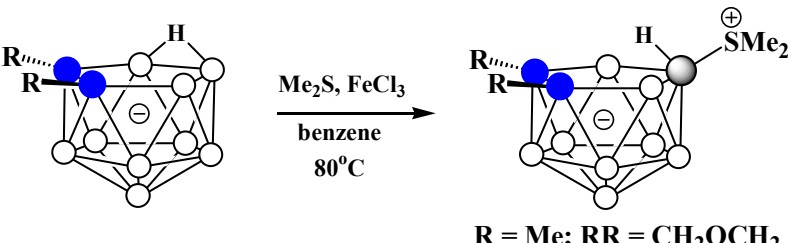

**Scheme 66.** Synthesis of symmetrically substituted dialkylsulfonium derivatives of *nido*-carborane in the presence of FeCl$_3$.

The reaction of the tetramethylammonium salt of *nido*-carborane with tetrahydrothiophene in refluxing acetone in the presence of AlCl$_3$ results in 10-(CH$_2$)$_4$S-7,8-C$_2$B$_9$H$_{11}$ (Scheme 67) [78].

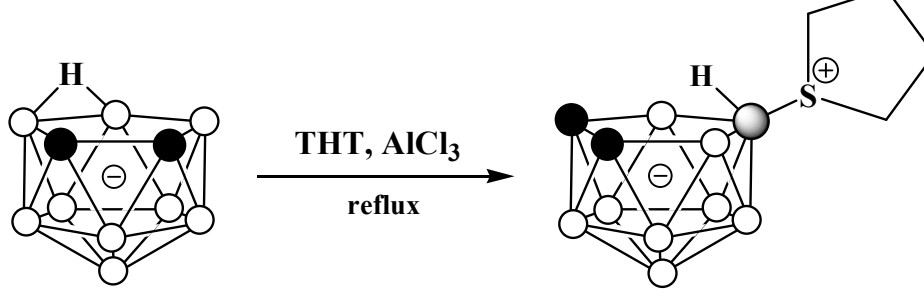

**Scheme 67.** Synthesis of 10-(CH$_2$)$_4$S-7,8-C$_2$B$_9$H$_{11}$.

The symmetrically substituted diphenylsulfonium derivative of *nido*-carborane 10-Ph$_2$S-7,8-C$_2$B$_9$H$_{11}$ was prepared via the reaction of the tetramethylammonium salt of *nido*-carborane with Ph$_3$CBF$_4$ in dichloromethane at −78 °C (Scheme 68, Figure 47) [78].

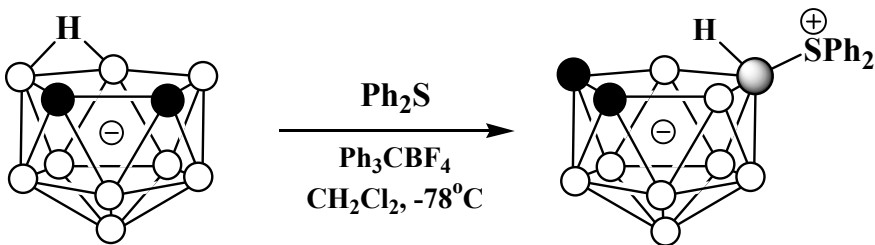

**Scheme 68.** Synthesis of 10-Ph$_2$S-7,8-C$_2$B$_9$H$_{11}$.

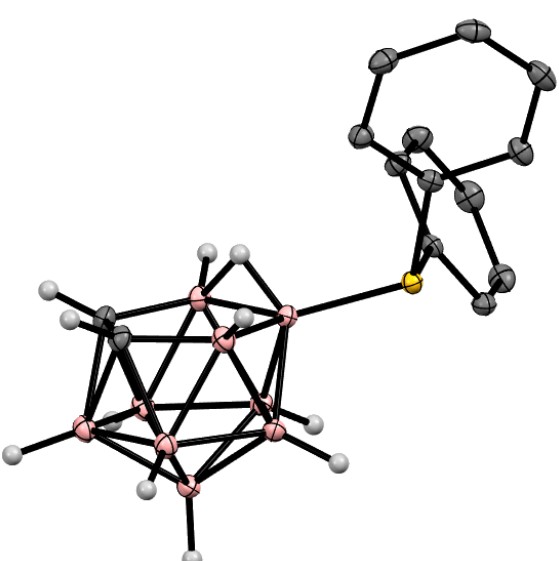

**Figure 47.** Crystal molecular structure of 10-Ph$_2$S-7,8-C$_2$B$_9$H$_{11}$. Hydrogen atoms of phenyl groups are omitted for clarity.

Similar to 9-Me$_2$S-7,8-C$_2$B$_9$H$_{11}$, the symmetrically substituted dimethylsulfonium derivative of *nido*-carborane 10-Me$_2$S-7,8-C$_2$B$_9$H$_{11}$ can be demethylated with sodium amide in boiling toluene [155] and re-alkylated with various alkylating agents in boiling chloroform or ethanol to give the corresponding sulfonium derivatives 10-R(Me)S-7,8-C$_2$B$_9$H$_{11}$ (Scheme 69, Figure 48). The resulting *nido*-carboranyl esters, nitriles, and phthalimides can be converted into corresponding carboxylic acids and amines using acid hydrolysis and deprotection with hydrazine, respectively (Scheme 68) [155,156].

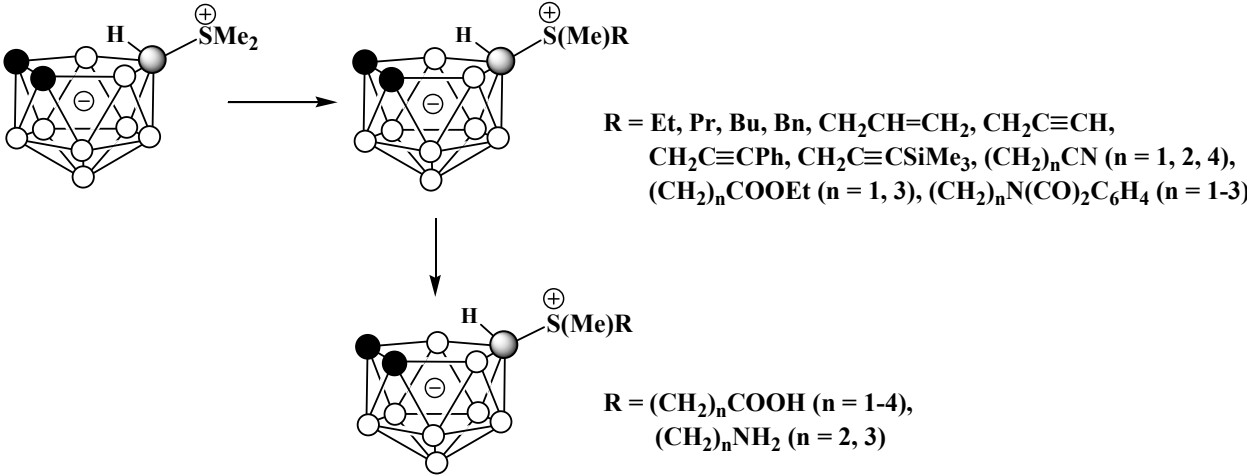

$R = Et, Pr, Bu, Bn, CH_2CH=CH_2, CH_2C\equiv CH,$
$CH_2C\equiv CPh, CH_2C\equiv CSiMe_3, (CH_2)_nCN (n = 1, 2, 4),$
$(CH_2)_nCOOEt (n = 1, 3), (CH_2)_nN(CO)_2C_6H_4 (n = 1-3)$

$R = (CH_2)_nCOOH (n = 1-4),$
$(CH_2)_nNH_2 (n = 2, 3)$

**Scheme 69.** The synthetic pathway to obtain $10\text{-R(Me)S-7,8-C}_2B_9H_{11}$ derivatives.

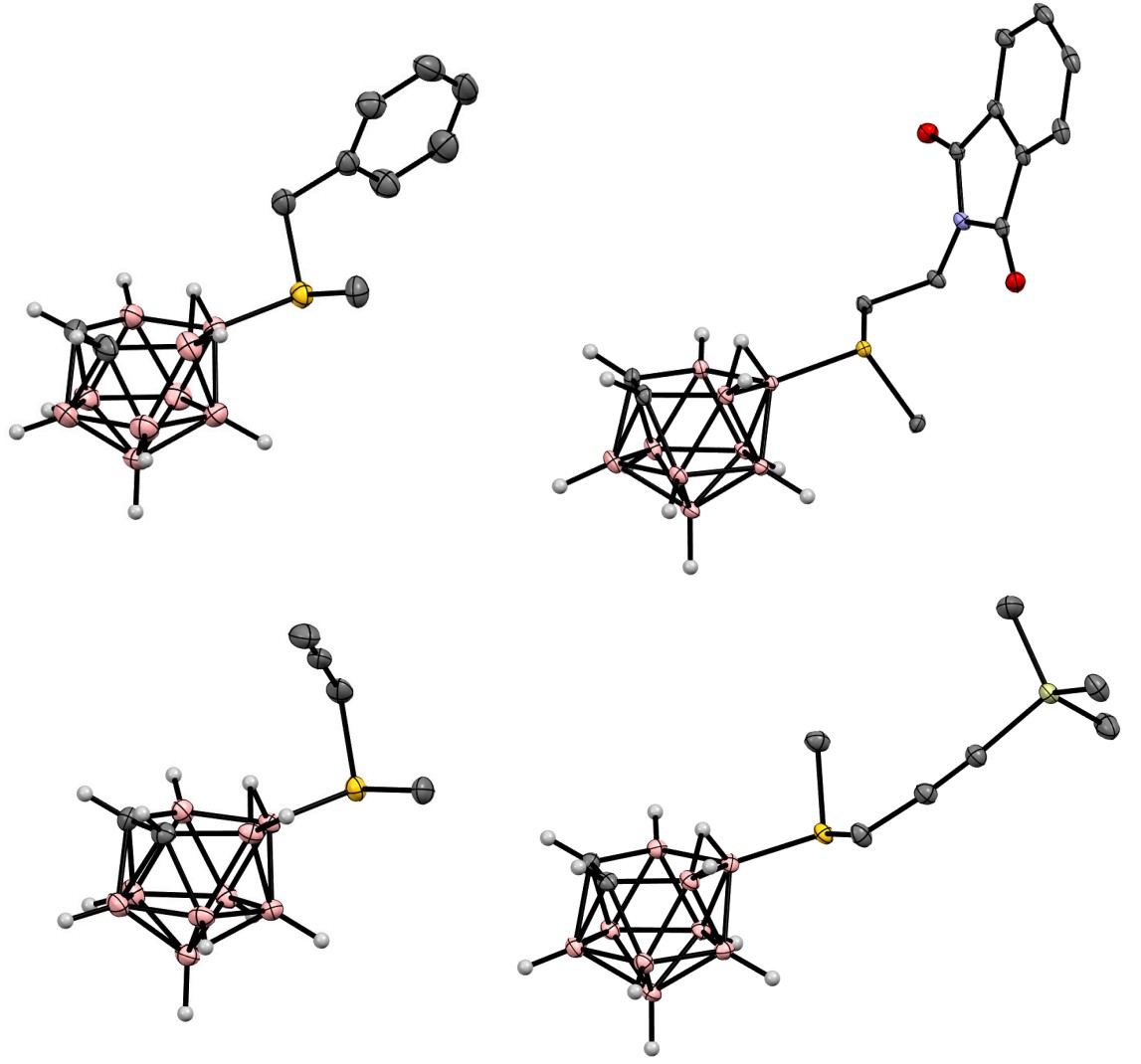

**Figure 48.** Crystal molecular structures of $10\text{-Bn(Me)S-7,8-C}_2B_9H_{11}$ (**top left**), $10\text{-}C_6H_4(CO)_2NCH_2CH_2(Me)S-7,8-C_2B_9H_{11}$ (**top right**), $10\text{-HC}\equiv CCH_2(Me)S-7,8-C_2B_9H_{11}$ (**bottom left**), and $10\text{-Me}_3SiC\equiv CCH_2(Me)S-7,8-C_2B_9H_{11}$ (**bottom right**). Hydrogen atoms of organic substituents are omitted for clarity.

A mixture of asymmetrically and symmetrically substituted dimethylsulfonium derivatives 9-Me$_2$S-7,8-C$_2$B$_9$H$_{11}$ and 10-Me$_2$S-7,8-C$_2$B$_9$H$_{11}$ was obtained via the reaction of the potassium salt of *nido*-carborane with H$_2$SO$_4$ and K$_2$Cr$_2$O$_7$ in a mixture of water and chloroform, followed by the addition of dimethylsulfide [157]. The reaction proceeds through the formation of di-*nido*-carborane C$_4$B$_{18}$H$_{22}$ followed by its splitting using dimethyl sulfide as a Lewis base.

The only example of the introduction of a dialkylsulfonium substituent into the lower belt of *nido*-carborane described in the literature is the reaction of the 9-mercapto derivative of *ortho*-carborane 9-HS-1,2-C$_2$B$_{10}$H$_{11}$ with KOH and MeI in methanol (Scheme 70) [136].

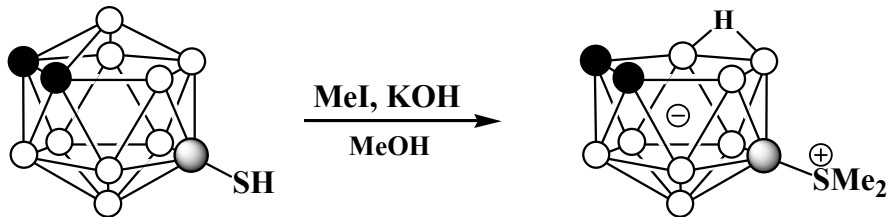

**Scheme 70.** Synthesis of 5-Me$_2$S-7,8-C$_2$B$_9$H$_{11}$.

In the chemistry of the *closo*-dodecaborate anion, an approach was previously developed for the preparation of its practically important mercapto derivative [B$_{12}$H$_{11}$SH]$^{2-}$ through the reaction of the parent *closo*-dodecaborate with thioureas or thioamides in an acidic medium followed by alkaline hydrolysis of the resulting charge-compensated *S*-thiouronium and *S*-thioimidolium derivatives [140]. The reaction of the formed in situ protonated form of *nido*-carborane C$_2$B$_9$H$_{13}$ in oluene under reflux conditions gave a mixture of the asymmetrically and symmetrically substituted thiouronium derivatives 9-(H$_2$N)$_2$CS-7,8-C$_2$B$_9$H$_{11}$ and 10-(H$_2$N)$_2$CS-7,8-C$_2$B$_9$H$_{11}$. These derivatives were hydrolyzed with NaOH in water, and the formed mercapto derivatives were alkylated with benzyl bromide in chloroform to give the corresponding dibenzylsulfonium derivatives 9-Bn$_2$S-7,8-C$_2$B$_9$H$_{11}$ and 10-Bn$_2$S-7,8-C$_2$B$_9$H$_{11}$, which were separated using column chromatography on silica (Scheme 71, Figure 49) [158,159].

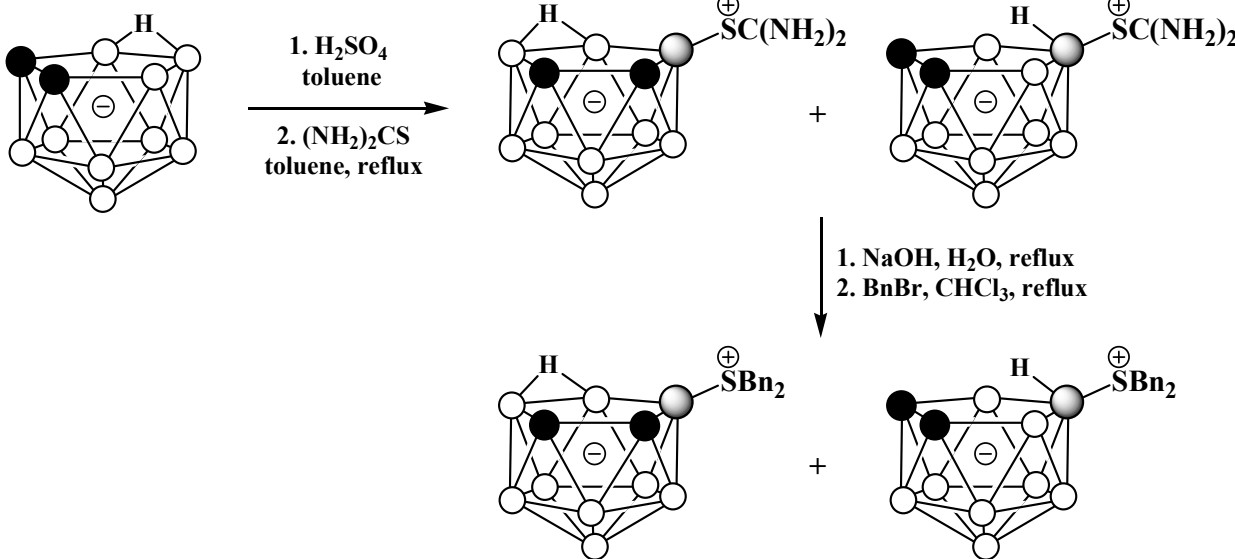

**Scheme 71.** Reaction of *nido*-carborane with thiourea under acidic conditions.

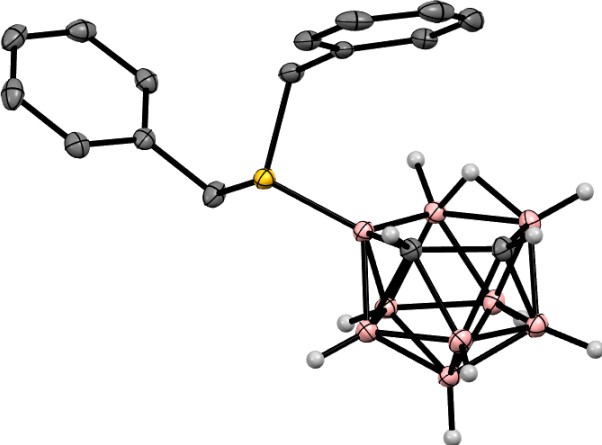

**Figure 49.** Crystal molecular structure of 9-Bn$_2$S-7,8-C$_2$B$_9$H$_{11}$. Hydrogen atoms of organic substituents are omitted for clarity.

The reactions of the tetramethylammonium salt of *nido*-carborane with thioacetamide and *N,N*-dimethylthioacetamide in refluxing acetone in the presence of AlCl$_3$ leads to the formation of the asymmetrically and symmetrically substituted *S*-thioimidolium derivatives 9-*i*-PrHNC(Me)S-7,8-C$_2$B$_9$H$_{11}$ and 10-Me$_2$NC(Me)S-7,8-C$_2$B$_9$H$_{11}$, respectively (Scheme 72, Figure 50) [98].

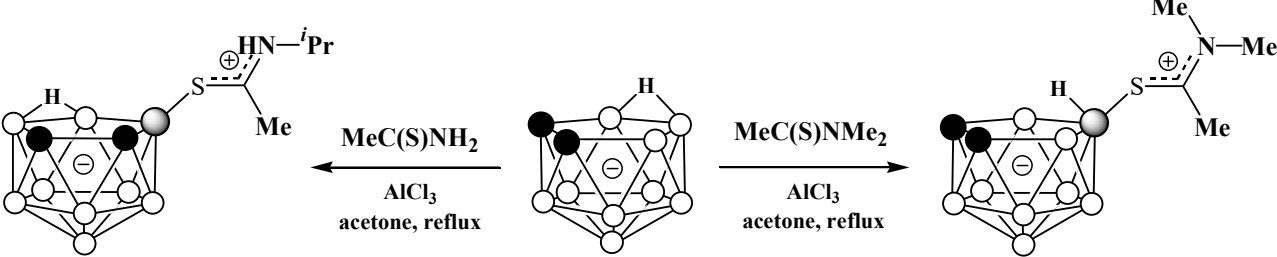

**Scheme 72.** Synthesis of *S*-thioimidolium derivatives of *nido*-carborane.

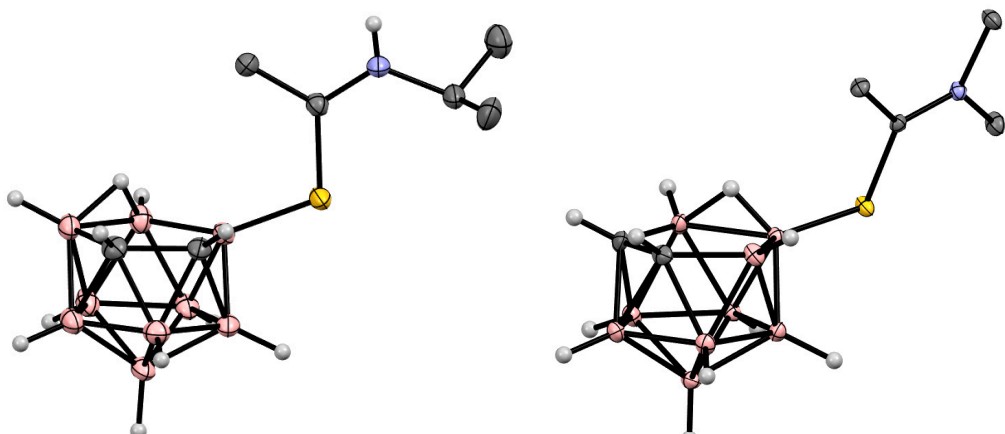

**Figure 50.** Crystal molecular structures of 9-*i*-PrHNC(Me)S-7,8-C$_2$B$_9$H$_{11}$ (**left**) and 10-Me$_2$NC(Me)S-7,8-C$_2$B$_9$H$_{11}$ (**right**). Hydrogen atoms of alkyl groups are omitted for clarity.

The 5-dimethylsulfonium derivative of *nido*-carborane was synthesized via the reaction of *orto*-carboran-9-yl(phenyl) iodonium tetrafluoroborate [9-PhI-1,2-C$_2$B$_{10}$H$_{11}$][BF$_4$] with dimethylsulfoxide (Scheme 73, Figure 51) [160].

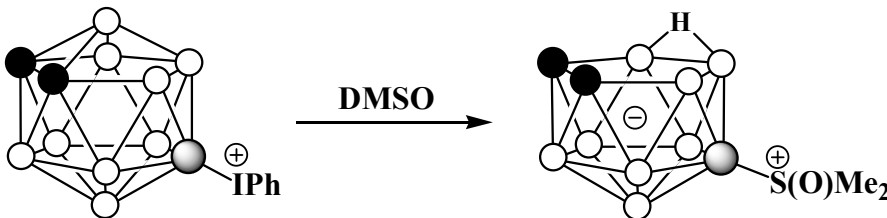

**Scheme 73.** Synthesis of the 5-dimethylsulfoxonium derivative of *nido*-carborane 5-Me$_2$(O)S-7,8-C$_2$B$_9$H$_{11}$.

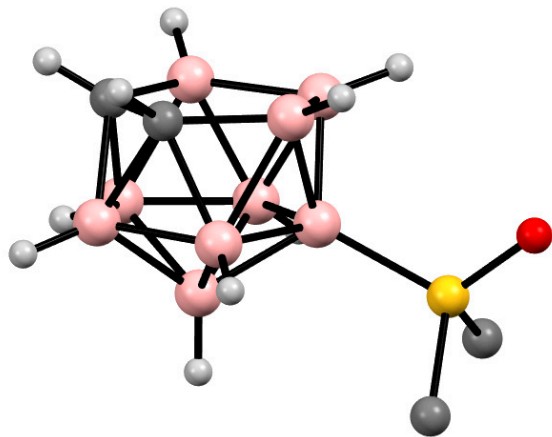

**Figure 51.** Crystal molecular structure of 5-Me$_2$(O)S-7,8-C$_2$B$_9$H$_{11}$. Hydrogen atoms of methyl groups are omitted for clarity.

## 7. Charge-Compensated Derivatives of *Nido*-Carborane with Boron–Selenium and Boron–Tellurium Bonds

The charge-compensated derivatives of *nido*-carborane with boron–selenium and boron–tellurium bonds are rather rare. Similar to the dialkyl- and diarylsulfonium derivatives, a series of asymmetrically substituted triakyl(aryl)selenium and triaryltellurium derivatives 9-RR′X-7,8-Ph$_2$-7,8-C$_2$B$_9$H$_9$ (X = Se, Te) were prepared via electrocatalyzed oxidative couplings of 7,8-diphenyl-*nido*-carborane with RR′Se and R$_2$Te, respectively (Scheme 74, Figure 52) [106].

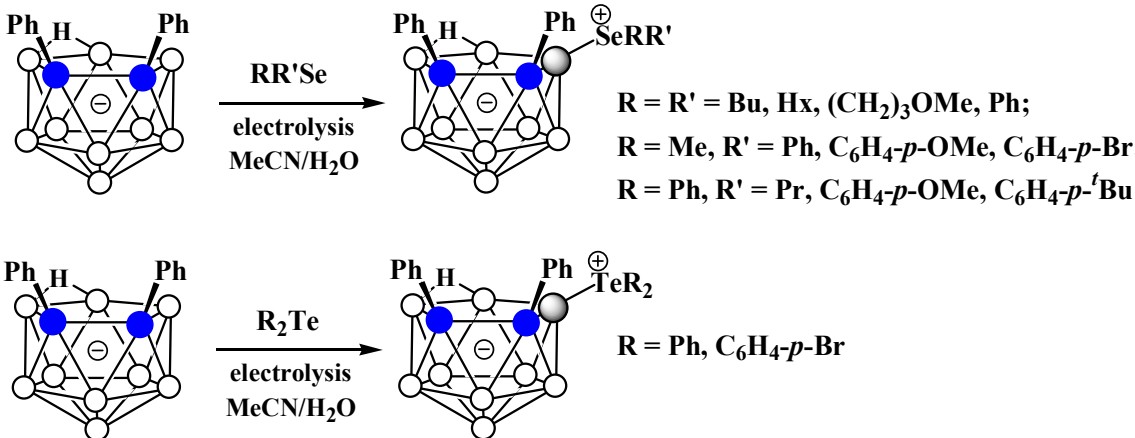

**Scheme 74.** Synthesis of 9-dialkyl(aryl)selenium and 9-diaryltellurium derivatives of *nido*-carborane 9-RR′X-7,8-Ph$_2$-7,8-C$_2$B$_9$H$_9$.

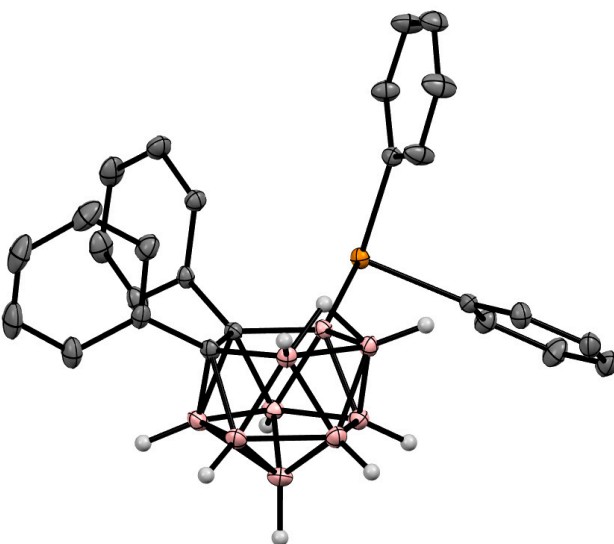

**Figure 52.** Crystal molecular structure of 9-$Ph_2Te$-7,8-$Ph_2$-7,8-$C_2B_9H_9$. Hydrogen atoms of phenyl groups are omitted for clarity.

## 8. Some Other Charge-Compensated Derivatives of *Nido*-Carborane

The asymmetrically substituted 9-carbonyl derivative of *nido*-carborane 9-O≡C-7,8-$C_2B_9H_{11}$ and the 3,3,8-$(CO)_3$-3,1,2-$CoC_2B_9H_{10}$ cobaltacarborane based on its symmetrically substituted analog as a ligand were isolated as minor products of the reaction of the parent *nido*-carborane with $[Co_2(CO)_8]$ [161].

The symmetrically substituted cobaltacenium derivative of *nido*-carborane 10-{CpCo($C_5H_4$)}-7,8-$Me_2$-7,8-$C_2B_9H_9$ (Figure 53) was prepared along with the 3-Cp-1,2-$Me_2$-3,1,2-$CoC_2B_9H_9$ cobaltacarborane in the reaction of the dithallium dicarbollide salt $Tl_2$[7,8-$Me_2$-7,8-$C_2B_9H_9$] with CpCo(CO)$I_2$ in acetonitrile [162].

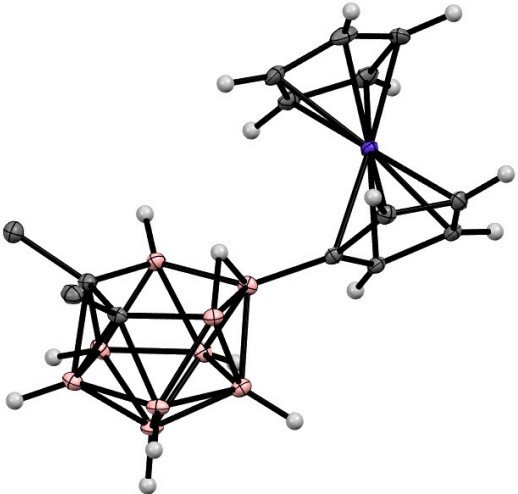

**Figure 53.** Crystal molecular structure of 10-{CpCo($C_5H_4$)}-7,8-$Me_2$-7,8-$C_2B_9H_9$. Hydrogen atoms of methyl groups are omitted for clarity.

## 9. Some Comments on Substitution Mechanisms in *Nido*-Carborane

In conclusion, we would like to touch upon the issue of substitution mechanisms in *nido*-carborane. As mentioned above, the introduction of substituents into the lower belt of the *nido*-carborane cage passes through the stage of substitution in the *closo*-carborane followed by deboronation and, in our opinion, does not require special comments. The synthesis of derivatives with substituents at the boron atoms in the upper belt of the *nido*-

carborane cage can proceed according to various reaction mechanisms and, depending on this, lead to both symmetrically and asymmetrically substituted derivatives.

The secondary substitution reaction mechanisms, such as Pd-catalyzed/promoted cross-coupling reactions of the iodo derivatives of *nido*-carborane or functional group-directed B-H activation via transition metal complexes, are quite obvious and do not require discussion.

As a rule, substitution in polyhedral boron hydrides can occur via two mechanisms [163]. The first one is analogous to well-known aromatic electrophilic substitution, while the second one involves the attack of electrophile $E^+$ followed by its elimination together with hydride (-EH). Then, the resulting electrophilic center is attacked by a nucleophile. This mechanism is called electrophilically induced nucleophilic substitution (EINS). In the simplest case, a proton can act as an electrophile; in this case, the reaction proceeds according to the mechanism of acid-assisted nucleophilic substitution (AANS). Since the first stage of the reaction in any case involves the attack of the electrophile, regardless of the mechanism, the substitution should proceed at the boron atom with the largest negative charge. The electrophilic center on a boron atom can also arise when the most-hydride hydrogen atom is removed by a Lewis acid. However, since the most-hydride hydrogen atom is bonded to the boron atom with the largest negative charge, the substitution position does not change. It is known that halogenation reactions, which are the simplest example of electrophilic substitution, occur at positions B(9) and B(11) of the *nido*-carborane cage [164–166]. Thus, if the substitution proceeded only via the aforementioned mechanisms, it would lead exclusively to asymmetrically substituted derivatives.

However, in the case of *nido*-carborane, there are a number of possibilities for substitution to occur in a different way. A characteristic feature of *nido*-carborane is the presence of the "extra"-hydrogen, which is able to migrate between the boron atoms of the open pentagonal face. Therefore, the intramolecular migration of the "extra" hydrogen to the electrophilic center formed in position B(9) can occur faster than the attack of the nucleophile. This should lead to the transfer of the electrophilic center to position B(10), the attack of which by the nucleophile will lead to symmetrically substituted derivatives.

On the other hand, strong bases can remove the "extra" hydrogen, leading to the formation of the dicarbollide anion $[7,8-C_2B_9H_{11}]^{2-}$ with a different electron density distribution than *nido*-carborane. In this case, the substitution proceeds at position B(10) with the formation of symmetrically substituted derivatives [104]. Interestingly, in the case of the protonated form of *nido*-carborane $7,8-C_2B_9H_{13}$, substitution also leads to the formation of symmetrically substituted derivatives [121,155]. This can be caused by the elimination of a hydrogen molecule with the closure of an unstable 11-vertex *closo*-polyhedron, which, before rearranging into a stable 2,3-isomer [167], is attacked by a nucleophile to form a substituted *nido*-carborane. It should be noted that in this case, as in the case of oxidative addition [79–81,84,106], we should consider these reaction pathways more like potential opportunities than established mechanisms, since there are no detailed studies on the mechanisms of these reactions. We can talk somewhat more definitely about the mercury-promoted substitution reactions, since in these cases, the formation of $\eta^1$-B(10)-mercuracarboranes, which can be considered as the initial products of the reaction, were reliably established with single-crystal X-ray diffraction studies [168–170].

Nevertheless, despite the fact that the mechanisms of formation of various charge-compensated *nido*-carborane derivatives remain largely unknown, analysis of the available literature data allows the targeted synthesis of these derivatives with high selectivity and good yields.

**Author Contributions:** Conceptualization, M.Y.S. and I.B.S.; writing—original draft preparation, M.Y.S., S.A.A. and I.B.S.; writing—review and editing, I.B.S.; supervision, I.B.S.; project administration, M.Y.S. and I.B.S. All authors have read and agreed to the published version of the manuscript.

**Funding:** This research was supported by the Russian Science Foundation (21-73-10199).

**Conflicts of Interest:** The authors declare no conflict of interest.

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
