# Peer review of "Charge-Compensated Derivatives of Nido-Carborane"

_inorganics, doi:10.3390/inorganics11020072_

Round 1

Reviewer 1 Report

This is a comprehensive review that is suitable for publication. To the reader the style becomes repetitive but it is a useful source of reliable information.  I am not overly familiar with the detail of the topic but have some corrections and some generlaized questions. 

The introduction should include a statement on previous reviews in the area, whether the review is comprehensive or not, and cut off date for recent papers. 

Abstract – is ‘data’ the correct word? Synthetic procedures? 

      nido  in itallic 

         ….. cage. 

            Delete  ‘type of’ 

L 35.  is a significantly stronger donor than the cyclopentadienide anion 

L57 What about position 1 - (see line 463 for an example). 

L100 delete ‘(see below)’ 

L140 – is there a list of abbreviations? E.g. BODIPY. TMEDA etc 

L156 scheme caption  ‘The synthesis of some pyridinium …'  But he sentence is correct if the scheme contains all. 

L601 – triaryl(alkyl)phosphines – meaning not precise. Triarylphosphines  and diarylalkylphosphines would be better.  This is a general point and should be considered throughout the manuscript. 

L616 ‘limited to’ 

L963 these should be diaryl and not triaryl derivatives but seem my point above about precision (L601). 

General point – is structure drawn with H bridging 10/11 for convenience/clarity? Surely it should bridge 9/10. 

The terms symmetrical and asymmetrical are used throughout the review – perhaps  their  definitions could be described in the introductory section.   The H atom on the open face in symmetrical derivatives need to be checked in the schemes – In many structures (e.g. scheme 25 to name one) the H atom is endo – this I believe is incorrect, and it needs to bridge.  This needs correction in all diagrams. 

Author Response

Reviewer #1

This is a comprehensive review that is suitable for publication. To the reader the style becomes repetitive but it is a useful source of reliable information.  I am not overly familiar with the detail of the topic but have some corrections and some generlaized questions. 

We thank the Reviewer for the positive feedback on this article and for valuable comments, which made it possible to improve the quality of the work.

The introduction should include a statement on previous reviews in the area, whether the review is comprehensive or not, and cut off date for recent papers. 

Previously, there were no reviews on this topic in the literature. This is the first review on charge-compensated nido-carborane derivatives.

Abstract – is ‘data’ the correct word? Synthetic procedures? 

      nido  in itallic 

         ….. cage. 

            Delete  ‘type of’ 

Some changes have been made in this part according to your comments.

L 35.  is a significantly stronger donor than the cyclopentadienide anion 

Correction done

L57 What about position 1 - (see line 463 for an example). 

Correction done

L100 delete ‘(see below)’ 

Correction done

L140 – is there a list of abbreviations? E.g. BODIPY. TMEDA etc 

Added abbreviation decoding

L156 scheme caption  ‘The synthesis of some pyridinium …'  But he sentence is correct if the scheme contains all. 

We believe that the scheme and the caption is correct.

L601 – triaryl(alkyl)phosphines – meaning not precise. Triarylphosphines  and diarylalkylphosphines would be better.  This is a general point and should be considered throughout the manuscript. 

Correction done

L616 ‘limited to’ 

Correction done

L963 these should be diaryl and not triaryl derivatives but seem my point above about precision (L601). 

Correction done

General point – is structure drawn with H bridging 10/11 for convenience/clarity? Surely it should bridge 9/10. 

In fact, the bridging hydrogen is located above all three boron atoms of the open pentagonal face of nido-carborane. But we draw it on unsubstituted boron atoms, so as not to complicate the picture

The terms ‘symmetrical’ and ‘asymmetrical’ are used throughout the review – perhaps their definitions could be described in the introductory section.   The H atom on the open face in symmetrical derivatives need to be checked in the schemes – In many structures (e.g. scheme 25 to name one) the H atom is endo – this I believe is incorrect, and it needs to bridge.  This needs correction in all diagrams. 

Initially, we believed that the concept of symmetrically and asymmetrically substituted derivatives is self-sufficient, but we agree with the Reviewer and introduced the appropriate definitions in the Introduction.

As regards the position of the "additional" hydrogen atom in the schemes, the situation is as follows. In the unsubstituted nido-carborane, the "additional" hydrogen atom migrates between the two bridge positions B(9)-H-B(10) and B(10)-H-B(11) passing through the endo-position B(10)-H. The substitution of one of the hydrogen atoms in positions 7 or 9 of the open pentagonal face of nido-carborane leads to the energy non-equivalence of the two bridge positions. In the case of electron-withdrawing substituents, which include charge-compensating substituents, the most advantageous position is remote from the substituent. Therefore, in this case, the "additional" hydrogen occupies a clearly marked position and does not migrate both in the solid state and in solution (See S. Hermanek, Chem. Rev., 1992, 92, 325-362). When the hydrogen atom is replaced in position 10, the equivalence of the two bridge positions and the migration of hydrogen between them is preserved (if the substituent does not contain an additional stereocenter). Therefore, the "extra" hydrogen atom often occupies exactly the endo position, as observed for the 10-methyldiphenylphosphonium derivative depicted in Scheme 25 (See Fig. 25 (right)). However, since the migration barrier between the two positions is low, often the position of the "additional" hydrogen is determined by various additional factors, including intermolecular interactions in the crystal. In particular, it can occupy both the bridge position (Fig. 45 (top left)), and the endo-position (Fig. 45 (top right)) or not reliably localize at all (Fig. 45 (bottom)). Therefore, in the case of symmetrically substituted nido-carborane derivatives, we prefer to depict the "additional" hydrogen atom precisely in the endo position, thus emphasizing the complete equivalence of neighboring unsubstituted positions in contrast to asymmetrically substituted derivatives.

Reviewer 2 Report

The review deals with charge-compensated nido-carboranes. In these compounds, an exo-polyhedral ligand binds a boron atom of the 11-vertex nido cage, affording zwitterionic neutral derivatives. The manuscripts is organized according to the type of B-X bond, starting from B-N, throughout B-P, B-As, B-Sb, B-O, B-S and B-Se to other minor derivatives. Overall, the work is a summary of reactions and schemes that illustrate the formation of these type of clusters; and, therefore, it is interesting from the synthetic point of view, since a scientist can learn different methods for the preparation of substituted nido-carboranes that can be used, for example, as ligands for the preparation of new metal complexes.

However, the review does not provide any mechanistic insights that could give a better understanding of the numerous reactions dealt with.

In this regard the crystallographically determined molecular structures, which are illustrated in the figures, are not discussed in any aspect, having, therefore, the same value as the chemdraw drawings in the schemes.

The lack of insights in this review is further demonstrated by the fact that the authors have not written a conclusion section.

In my opinion, the manuscript is a simple list of reactions, schemes and figures, without further chemical visions. And, in order to improve the academic value of this work, the authors should include some analysis and descriptions of the reactions, crystal structures, etc.

The authors should not forget to write the conclusions of the review. 

Author Response

Reviewer #2

The review deals with charge-compensated nido-carboranes. In these compounds, an exo-polyhedral ligand binds a boron atom of the 11-vertex nido cage, affording zwitterionic neutral derivatives. The manuscripts is organized according to the type of B-X bond, starting from B-N, throughout B-P, B-As, B-Sb, B-O, B-S and B-Se to other minor derivatives. Overall, the work is a summary of reactions and schemes that illustrate the formation of these type of clusters; and, therefore, it is interesting from the synthetic point of view, since a scientist can learn different methods for the preparation of substituted nido-carboranes that can be used, for example, as ligands for the preparation of new metal complexes.

We thank the Reviewer for the positive feedback on this article and for valuable comments, which made it possible to improve the quality of the work.

However, the review does not provide any mechanistic insights that could give a better understanding of the numerous reactions dealt with.

Unfortunately, it is difficult to assign a specific mechanism with certainty to most of the reactions for obtaining charge-compensated nido-carborane derivatives, since the nature of nido-carborane opens up various ways to obtain the same derivatives, and there is no detailed study of the reaction mechanisms in the literature. Nevertheless, we have attempted to give some summary which should briefly illuminate various possible mechanisms for substitution reactions in nido-carborane.

In this regard the crystallographically determined molecular structures, which are illustrated in the figures, are not discussed in any aspect, having, therefore, the same value as the chemdraw drawings in the schemes.

We believe that crystallographically determined molecular structures carry more information than schemes that give a very approximate idea of both the mutual arrangement of the substituents and their size. Perhaps it would be worth discussing with several examples the possibilities of intra-molecular interactions between substituents and nido-carborane cage (as it was done in Sivaev et al., Phosphorus, Sulfur, Silicon, Related Elements, 2018, 193, 104), but it seems to us that this would take us away from the subject of this review.

The lack of insights in this review is further demonstrated by the fact that the authors have not written a conclusion section.

In the revised version of the manuscript, a brief consideration of possible mechanisms of reactions plays role of Conclusion.

In my opinion, the manuscript is a simple list of reactions, schemes and figures, without further chemical visions. And, in order to improve the academic value of this work, the authors should include some analysis and descriptions of the reactions, crystal structures, etc. The authors should not forget to write the conclusions of the review. 

In Conclusion we have attempted to discuss briefly various possible mechanisms for substitution reactions in nido-carborane.

Round 2

Reviewer 2 Report

In the revised manuscript, the authors have dealt with all my comments and concerns, and I believe the manuscript has been sufficiently improved to warrant publication in Inorganics.